# Benign Overfitting in Multiclass Classification: All Roads Lead to Interpolation

**Ke Wang**
Department of Statistics and Applied Probability
University of California, Santa Barbara
Santa Barbara, CA 93106
kewang01@ucsb.edu

**Vidya Muthukumar**
School of Electrical and Computer Engineering & Industrial and Systems Engineering
Georgia Institute of Technology
Atlanta, GA 30332
vmuthukumar8@gatech.edu

**Christos Thrampoulidis**
Department of Electrical and Computer Engineering
University of British Columbia
Vancouver, BC Canada V6T 1Z4
cthrampo@ece.ubc.ca

## Abstract

The growing literature on "benign overfitting" in overparameterized models has been mostly restricted to regression or binary classification settings; however, most success stories of modern machine learning have been recorded in multiclass settings. Motivated by this discrepancy, we study benign overfitting in multiclass linear classification. Specifically, we consider the following popular training algorithms on separable data: (i) empirical risk minimization (ERM) with cross-entropy loss, which converges to the multiclass support vector machine (SVM) solution; (ii) ERM with least-squares loss, which converges to the min-norm interpolating (MNI) solution; and, (iii) the one-vs-all SVM classifier. Our first key finding is that under a simple sufficient condition, *all* three algorithms lead to classifiers that interpolate the training data and have equal accuracy. When the data is generated from Gaussian mixtures or a multinomial logistic model, this condition holds under high enough effective overparameterization. Second, we derive novel error bounds on the accuracy of the MNI classifier, thereby showing that all three training algorithms lead to benign overfitting under sufficient overparameterization. Ultimately, our analysis shows that good generalization is possible for SVM solutions beyond the realm in which typical margin-based bounds apply.

## 1 Introduction

Modern deep neural networks are *overparameterized* with respect to the amount of training data and achieve zero training error, yet generalize well on test data. Recent analysis has shown that fitting of noise in regression tasks can in fact be relatively benign for sufficiently high-dimensional linear models [BLLT20, BHX20, HMRT19, MVSS20, KLS20]. However, these analyses do not

35th Conference on Neural Information Processing Systems (NeurIPS 2021).

directly extend to classification, which requires separate treatment. In fact, very recent progress on sharp analysis of interpolating *binary* classifiers [MNS+20, CL21, WT21, CGB21] revealed high-dimensional regimes in which binary classification generalizes well, but the corresponding regression task does *not* work and/or the success *cannot* be predicted by classical margin-based bounds.

In an important separate development, these same high-dimensional regimes admit an equivalence of loss functions used at training time. The support vector machine (SVM), which arises from minimizing the logistic loss using gradient descent [SHN+18, JT19], was recently shown to satisfy a high-probability equivalence to interpolation, which arises from minimizing the squared loss [MNS+20, HMX21]. This equivalence suggests that interpolation is ubiquitous in very overparmaeterized settings, and can arise naturally as a consequence of the optimization procedure even when this is not explicitly encoded or intended. Moreover, this equivalence to interpolation and corresponding analysis implies that the SVM can generalize even in regimes where classical learning theory bounds are not predictive. In the logistic model case [MNS+20] and Gaussian binary mixture model case [CL21, WT21, CGB21], it is shown that good generalization of the SVM is possible beyond the realm in which classical margin-based bounds apply. These analyses lend theoretical grounding to the surprising hypothesis that *squared loss can be equivalent to, or possibly even superior*, to the cross-entropy loss for classification tasks. This hypothesis was supported empirically on kernel machines in Ryan Rifkin's doctoral dissertation work [Rif02, RK04], and more recently in overparameterized neural networks [HB20, PL20].

These compelling perspectives have thus far been limited to regression and *binary* classification settings. In contrast, most success stories and surprising new phenomena of modern machine learning have been recorded in *multiclass* classification settings, which appear naturally in a host of applications that demand the ability to automatically distinguish between large numbers of different classes; for example, the popular ImageNet dataset [RDS+15] contains on the order of 1000 classes. Whether a) good generalization beyond effectively low-dimensional regimes where margin-based bounds are predictive is possible, and b) equivalence of squared loss and cross-entropy loss holds in multiclass settings remained open problems.

This paper makes significant progress towards a complete understanding of the optimization and generalization properties of high-dimensional linear multiclass classification, both for unconditional Gaussian covariates (where labels are generated via a multinomial logistic model), and high-dimensional Gaussian mixture models. Our contributions are listed in more detail below.

## 1.1  Our Contributions

• We establish a *deterministic* sufficient condition under which the multiclass SVM solution has a very simple and symmetric structure: it is identical to the solution of the One-vs-All (OvA) SVM classifier that uses the one-hot encoded labels. Moreover, the constraints at both solutions are active. Geometrically, this means that *all data points are support vectors*.

• This implies a surprising equivalence between traditionally different formulations of multiclass SVM, which in turn are equivalent to the minimum-norm interpolating (MNI) classifier on one-hot label vectors. Thus, the outcomes of training with cross-entropy (CE) loss and squared loss are identical.

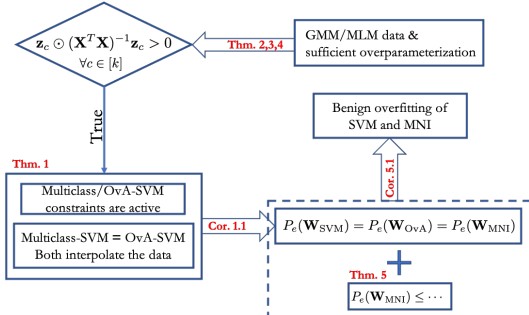

Figure 1: Contributions and organization.

• Next, for data following a Gaussian-mixtures model (GMM) or a Multinomial logistic model (MLM), we show that the above sufficient condition is satisfied with high-probability under sufficient effective overparameterization depending on the number of classes, and on quantities related to the data covariance. Our numerical results show excellent agreement with our theoretical findings.

• Subsequently, we provide novel bounds on the error of the MNI classifier for data generated either from the GMM or the MLM and characterize overparmeterization conditions under which benign overfitting occurs. A direct outcome of our results is that benign overfitting occurs under these conditions regardless of whether the cross-entropy loss or squared loss is used during training.

Figure 1 describes our contributions and their implications through a flowchart. *To the best of our knowledge, these are the first results characterizing a) equivalence of loss functions, and b) generalization of interpolating solutions in the multiclass setting.* The multiclass setting poses several challenges over and above the recently studied binary case. When presenting our results in later sections, we discuss in detail how our analysis circumvents these challenges.

## 1.2 Related Work

**Multiclass classification and the impact of training loss functions.** There is a classical body of work on algorithms for multiclass classification, e.g., [WW98, BB99, DB95, CS02, LLW04] and several empirical studies of their comparative performance [RK04, FÖ2, ASS01] (also see [HYS16, GCOZ17, KS18, BEH20, DCO20, HB20, PL20] for recent such studies in the context of deep nets). Many of these (e.g. [RK04, HB20, BEH20]) have found that least-squares minimization yields competitive test classification performance to cross-entropy minimization. *Our proof of equivalence of the SVM and MNI solutions under sufficient overparameterization provides theoretical support for this line of work.* This is a consequence of the implicit bias of gradient descent run on the CE and squared losses leading to the multiclass SVM [SHN$^+$18, JT19] and MNI [EHN96] respectively. Numerous classical works investigated consistency [Zha04, LLW04, TB07, PGS13, PS16] and finite-sample behavior, e.g., [KP02, CKMY16, LDBK15, Mau16, LDZK19] of multiclass classification algorithms in the underparameterized regime. In contrast, our focus is on the highly overparameterized regime, where the typical uniform convergence techniques cannot apply.

**Binary classification error analyses in overparameterized regime.** The recent wave of analyses of the minimum-$\ell_2$-norm interpolator (MNI) in high-dimensional linear regression (an incomplete list is [BLLT20, BHX20, HMRT19, MVSS20, KLS20]) prompted researchers to consider to what extent the phenomena of benign overfitting and double descent [BHMM19, GJS$^+$20] can be proven to occur in classification tasks. Even the binary classification setting turns out to be significantly more challenging to study owing to the discontinuity of the $0 - 1$ test loss function. Sharp asymptotic formulas for the generalization error of binary classification algorithms in the linear high-dimensional regime have been derived in several recent works [Hua17, SC19, MLC19, SAH19, TPT20, TPT21, DKT21, MRSY19, KA21, LS20, SAH20, AKLZ20, Lol20, DL20]. These formulas are solutions to complicated nonlinear systems of equations that typically do not admit closed-form expressions. A separate line of work provides non-asymptotic error bounds for both the MNI classifier and the SVM classifier [CL21, MNS$^+$20, WT21, CGB21]; in particular, [MNS$^+$20] analyzed the SVM in a Gaussian covariates model by explicitly connecting its solution to the MNI solution. Subsequently, [WT21] also took this route to analyze the SVM and MNI in mixture models, which turn out to be more technically involved. Even more recently, [CGB21] provided extensions of the result by [WT21] to sub-Gaussian mixtures. While these non-asymptotic analyses are only sharp in their dependences on $n$ and $p$, they provide closed-form generalization expressions in terms of easily interpretable summary statistics. Interestingly, these results imply good generalization of the SVM beyond the regime in which margin-based bounds are predictive. Specifically, [MNS$^+$20] identifies a separating regime for Gaussian covariates in which corresponding regression tasks would not generalize. In the Gaussian mixture model, margin-based bounds [SFBL98, BM03] (as well as corresponding recently derived mistake bounds on interpolating classifiers [LR21]) would require the intrinsic signal-to-noise-ratio (SNR) to scale at least as $\omega(p^{1/2})$ for good generalization; however, the analyses of [CL21, WT21, CGB21] show that good generalization is possible for significantly lower SNR scaling as $\omega(p^{1/4})$ The above error analyses are specialized to the binary case, where closed-form error expressions are easy to derive [MNS$^+$20]. The only related work applicable to multiclass settings is [TOS20], which also highlights the numerous challenges of obtaining a sharp error analysis in multiclass settings. Specifically, [TOS20] derived sharp generalization formulas for multiclass least-squares in underparameterized settings; extensions to the overparameterized regime and other losses beyond least-squares remained wide open. Finally, [KT21] recently derived sharp phase-transition thresholds for the feasibility of OvA-SVM on multiclass Gaussian mixture data in the linear high-dimensional regime. However, their result does not cover the more challenging multiclass-SVM that we investigate here.

**Other SVM analyses.** The number of support vectors in *binary* SVM has been characterized in low-dimensional separable and non-separable settings [DOS99, BG01, MO05] and scenarios have been

identified in which there is vanishing fraction of support vectors, as this implies good generalization[1] via PAC-Bayes sample compression bounds [Vap13]. In the highly overparameterized regime that we consider, perhaps surprisingly, the opposite behavior occurs: *all training points become support vectors with high probability* [DOS99, BG01, MO05, MNS+20, HMX21]. In particular, [HMX21] provided sharp non-asymptotic sufficient conditions for this phenomenon for both isotropic and anisotropic settings. The techniques in [MNS+20, HMX21] are highly specialized to the binary SVM and its dual, where a simple complementary slackness condition directly implies the property of interpolation. In contrast, the complementary slackness condition for the case of multiclass SVM *does not* directly imply interpolation; in fact, the operational meaning of "all training points becoming support vectors" is unclear in the multiclass SVM. *Our proof of deterministic equivalence goes beyond the complementary slackness condition and uncovers a surprising symmetric structure[2] by showing equivalence of multiclass SVM to a symmetric OvA classifier.*

**Notation** For a vector $\mathbf{v} \in \mathbb{R}^p$ , let $\|\mathbf{v}\|_2 = \sqrt{\sum_{i=1}^p v_i^2}$, $\|\mathbf{v}\|_1 = \sum_{i=1}^p |v_i|$, $\|\mathbf{v}\|_\infty = \max_i\{|v_i|\}$. $\mathbf{v} > \mathbf{0}$ is interpreted elementwise. $\mathbf{1}$ / $\mathbf{0}$ denote the all-ones / all-zeros vectors and $\mathbf{e}_i$ denotes the $i$-th standard basis vector. For a matrix $\mathbf{M}$, $\|\mathbf{M}\|_2$ denotes its $2 \to 2$ operator norm and $\|\mathbf{M}\|_F$ denotes the Frobenius norm. $\odot$ denotes the Hadamard product. $[n]$ denotes the set $\{1, 2, ..., n\}$. We also use standard "Big O" notations $\Theta(\cdot)$, $\omega(\cdot)$, e.g., see [CLRS09, Chapter 3]. Finally, we write $\mathcal{N}(\boldsymbol{\mu}, \boldsymbol{\Sigma})$ for the (multivariate) Gaussian distribution of mean $\boldsymbol{\mu}$ and covariance matrix $\boldsymbol{\Sigma}$, and, $Q(x) = \mathbb{P}(Z > x)$, $Z \sim \mathcal{N}(0, 1)$ for the Q-function of a standard normal. Throughout, constants refer to numbers that do not depend on the problem dimensions $n$ or $p$.

## 2 Problem setting

We consider the multiclass classification problem with $k$ classes. Let $\mathbf{x} \in \mathbb{R}^p$ denote the feature vector and $y \in [k]$ represent the class label associated with one of the $k$ classes. We assume that the training data has $n$ feature/label pairs $\{\mathbf{x}_i, y_i\}_{i=1}^n$. We focus on the overparameterized regime, i.e., $p > Cn$, and will frequently consider $p \gg n$. For convenience, we express the labels using the one-hot coding vector $\mathbf{y}_i \in \mathbb{R}^k$, where only the $y_i$-th entry of $\mathbf{y}_i$ is 1 and all other entries are zero, i.e., $\mathbf{y}_i = \boldsymbol{e}_{y_i}$. With this notation, the feature and label matrices are given in compact form as follows: $\mathbf{X} = [\mathbf{x}_1 \quad \mathbf{x}_2 \quad \cdots \quad \mathbf{x}_n] \in \mathbb{R}^{p \times n}$ and $\mathbf{Y} = [\mathbf{y}_1 \quad \mathbf{y}_2 \quad \cdots \quad \mathbf{y}_n] = [\mathbf{v}_1 \quad \mathbf{v}_2 \quad \cdots \mathbf{v}_k]^T \in \mathbb{R}^{k \times n}$, where we have defined $\mathbf{v}_c \in \mathbb{R}^n, c \in [k]$ to denote the $c$-th row of the matrix $\mathbf{Y}$.

### 2.1 Data models

We assume that the data pairs $\{\mathbf{x}_i, y_i\}_{i=1}^n$ are generated IID. We will consider two models for the distribution of $(\mathbf{x}, y)$. For both models, we define the mean vectors $\{\boldsymbol{\mu}_j\}_{j=1}^k \in \mathbb{R}^p$, and the mean matrix is given by $\mathbf{M} := [\boldsymbol{\mu}_1 \quad \boldsymbol{\mu}_2 \quad \cdots \quad \boldsymbol{\mu}_k] \in \mathbb{R}^{p \times k}$.

**Gaussian Mixture Model (GMM).** In this model, the mean vector $\boldsymbol{\mu}_i$ represents the conditional mean vector for the $i$-th class. Specifically, each observation $(\mathbf{x}_i, y_i)$ belongs to to class $c \in [k]$ with probability $\pi_c$ and conditional on the label $y_i$, $\mathbf{x}_i$ follows a multivariate Gaussian distribution. In summary, we have

$$\mathbb{P}(y = c) = \pi_c \text{ and } \mathbf{x} = \boldsymbol{\mu}_y + \mathbf{q}, \ \mathbf{q} \sim \mathcal{N}(\mathbf{0}, \boldsymbol{\Sigma}). \tag{1}$$

In this work, we focus on the isotropic case $\boldsymbol{\Sigma} = \mathbf{I}_p$. Our analysis can likely be extended to more general settings, but we leave this to future work.

**Multinomial Logit Model (MLM).** In this model, the feature vector $\mathbf{x} \in \mathbb{R}^p$ follows $\mathcal{N}(\mathbf{0}, \boldsymbol{\Sigma})$, and the conditional density of the class label $y$ is given by the soft-max function. Specifically, we have

$$\mathbf{x} \sim \mathcal{N}(\mathbf{0}, \boldsymbol{\Sigma}) \text{ and } \mathbb{P}(y = c|\mathbf{x}) = \exp(\boldsymbol{\mu}_c^T \mathbf{x}) \Big/ \sum_{j \in [k]} \exp(\boldsymbol{\mu}_j^T \mathbf{x}). \tag{2}$$

For this model, we analyze both the isotropic and anisotropic cases.

---

[1]In this context, the fact that [MNS+20, WT21] provide good generalization bounds in the regime where support vectors proliferate is particularly surprising. In conventional wisdom, a proliferation of support vectors was associated with overfitting but this turns out to not be the case here.

[2]This symmetric structure is somewhat reminiscent of the recently observed neural collapse phenomenon in deep neural networks [PHD20], although the details of the obtained solutions are quite different.

## 2.2 Data separability

We consider linear classifiers parameterized by $\mathbf{W} = \begin{bmatrix} \mathbf{w}_1 & \mathbf{w}_2 & \cdots & \mathbf{w}_k \end{bmatrix}^T \in \mathbb{R}^{k \times p}$. Given input feature vector $\mathbf{x}$, the classifier is a function that maps $\mathbf{x}$ into an output of $k$ via[3] $\mathbf{x} \mapsto \mathbf{W}\mathbf{x} \in \mathbb{R}^k$. We will operate in a regime where the training data are linearly separable. In multiclass settings, there exist multiple notions of separability. Here, we focus on (i) multiclass/$k$-class separability (ii) one-vs-all (OvA) separability, and, recall their definitions below.

**Definition 1** (multiclass and OvA separability). *The dataset $\{\mathbf{x}_i, y_i\}_{i \in [n]}$ is multiclass linearly separable when $\exists \mathbf{W} = [\mathbf{w}_1, \mathbf{w}_2, \ldots, \mathbf{w}_k]^T \in \mathbb{R}^{k \times p} : (\mathbf{w}_{y_i} - \mathbf{w}_c)^T \mathbf{x}_i \geq 1, \ \forall c \neq y_i, c \in [k], \ and \ \forall i \in [n]$. The dataset is one-vs-all (OvA) separable when $\exists \mathbf{W} = [\mathbf{w}_1, \mathbf{w}_2, \ldots, \mathbf{w}_k]^T \in \mathbb{R}^{k \times p} : \mathbf{w}_c^T \mathbf{x}_i \geq 1$ if $y_i = c$ and $\mathbf{w}_c^T \mathbf{x}_i \leq -1$ if $y_i \neq c$, $\forall c \in [k]$, and $\forall i \in [n]$.*

In the overparameterized regime $p > n$ with Gaussian data, we have $\mathrm{rank}(\mathbf{X}) = \mathrm{n}$ almost surely, which implies OvA separability. It turns out that OvA separability implies multiclass separability, but not vice versa (see [BM94] for a counterexample).

## 2.3 Classification error

Consider a linear classifier $\widehat{\mathbf{W}}$ and a fresh sample $(\mathbf{x}, y)$ generated following the same distribution as the training data. As is standard, we predict $\hat{y}$ by a "winner takes it all strategy", i.e., $\hat{y} = \arg\max_{j \in [k]} \widehat{\mathbf{w}}_j^T \mathbf{x}$. Then, the classification error conditioned on the true label being $c$, which we refer to as the *class-wise classification error*, is defined as $\mathbb{P}_{e|c} := \mathbb{P}(\hat{y} \neq y | y = c) = \mathbb{P}(\widehat{\mathbf{w}}_c^T \mathbf{x} \leq \max_{j \neq c} \widehat{\mathbf{w}}_j^T \mathbf{x})$. In turn, the *total classification error* is defined as $\mathbb{P}_e := \mathbb{P}(\hat{y} \neq y) = \mathbb{P}(\arg\max_{j \in [k]} \widehat{\mathbf{w}}_j^T \mathbf{x} \neq y) = \mathbb{P}(\widehat{\mathbf{w}}_y^T \mathbf{x} \leq \max_{j \neq y} \widehat{\mathbf{w}}_j^T \mathbf{x})$.

## 2.4 Classification algorithms

Next, we review several different training strategies for which we characterize the total/class-wise classification error in this paper.

**Multiclass SVM.** Consider training $\mathbf{W}$ by minimizing the popular cross-entropy (CE) loss $\mathcal{L}(\mathbf{W}) := -\log\left(e^{\mathbf{w}_{y_i}^T \mathbf{x}_i} / \sum_{c \in [k]} e^{\mathbf{w}_c^T \mathbf{x}_i}\right)$ with the gradient descent algorithm (with constant step size $\eta$). In the separable regime that we consider, the CE loss $\mathcal{L}(\mathbf{W})$ can be driven to zero. Moreover, [SHN+18, Thm. 7] showed that the normalized iterates $\{\mathbf{W}^t\}_{t \geq 1}$ converge as $\lim_{t \to \infty} \left\| \mathbf{W}^t / \log t - \mathbf{W}_{\mathrm{SVM}} \right\|_F = 0$,[4] where $\mathbf{W}_{\mathrm{SVM}}$ is the solution of the *multiclass SVM* [WW98] given by

$$\mathbf{W}_{\mathrm{SVM}} := \arg\min_{\mathbf{W}} \|\mathbf{W}\|_F \quad \text{sub. to } (\mathbf{w}_{y_i} - \mathbf{w}_c)^T \mathbf{x}_i \geq 1, \ \forall i \in [n], c \in [k] \text{ s.t. } c \neq y_i. \quad (3)$$

**One-vs-all SVM.** In contrast to Eqn. (3) that optimizes the hyperplanes $\{\mathbf{w}_c\}_{c \in [k]}$ jointly, the one-vs-all (OvA)-SVM classifier solves $k$ separable optimization problems maximizing the margin of each class with respect to all the rest. Concretely, the OvA-SVM solves for all $c \in [k]$:

$$\mathbf{w}_{\mathrm{OvA},c} := \arg\min_{\mathbf{w}} \|\mathbf{w}\|_2 \quad \text{sub. to } \mathbf{w}^T \mathbf{x}_i \geq 1, \text{ if } y_i = c; \ \mathbf{w}^T \mathbf{x}_i \leq -1 \text{ if } y_i \neq c, \ \forall i \in [n]. \quad (4)$$

In general, the solutions to Equations (3) and (4) are different. While the OvA-SVM does not have an obvious connection to any training loss function, its relevance will become clear in Section 3. Perhaps surprisingly, we will prove that in the highly overparameterized regime the multiclass SVM solution is identical to a slight variant of (4).

**Min-norm interpolating (MNI) classifier.** An alternative to the CE loss is the square loss $\mathcal{L}(\mathbf{W}) := \frac{1}{2n} \|\mathbf{Y} - \mathbf{W}\mathbf{X}\|_2^2 = \frac{1}{2n} \sum_{i=1}^n \|\mathbf{W}\mathbf{x}_i - \mathbf{y}_i\|_2^2$. While the square-loss appears to be more tailored to regression, it in fact has competitive classification accuracy to the CE loss in practice [Rif02, HB20, PL20]. Since $\mathrm{rank}(\mathbf{X}) = \mathrm{n}$ almost surely, the data can be linearly interpolated, i.e. the square-loss

---

[3]For simplicity, we ignore the bias term throughout.

[4]Note that the scaling factor $\log t$ here does *not* depend on the class label; hence, in the limit of GD iterations, the solution $\mathbf{W}^t$ decides the same label as multiclass SVM for any test sample.

can be made zero. Then, it is well-known [EHN96] that gradient descent with sufficiently small step size and appropriate initialization converges to the minimum-norm -interpolating (MNI) solution:

$$\mathbf{W}_{\text{MNI}} := \arg\min_{\mathbf{W}} \|\mathbf{W}\|_F, \ \ \text{sub. to } \mathbf{X}^T\mathbf{w}_c = \mathbf{v}_c, \forall c \in [k]. \tag{5}$$

Since $\mathbf{X}^T\mathbf{X}$ is invertible, the solution above is given in closed form as $\mathbf{W}_{\text{MNI}}^T = \mathbf{X}(\mathbf{X}^T\mathbf{X})^{-1}\mathbf{Y}^T$. From here on, we refer to (5) as the MNI classifier.

## 3 Proliferation of support vectors

In this section, we show equivalence of the solutions of the three classifiers defined above.

### 3.1 A key deterministic condition

We first establish a key deterministic property of SVM that holds for *generic* multiclass datasets $(\mathbf{X}, \mathbf{Y})$ (not necessarily generated by either the GMM or MLM), as long as $\text{rank}(\mathbf{X}) = \text{n}$. Specifically, Theorem 1 below derives a sufficient condition (cf. (8)) under which the multiclass SVM solution has a surprisingly simple structure. First, the constraints are *all* active at the optima (cf. (9)). Second, and perhaps more interestingly, the equality of the constraints is satisfied in a very symmetric way such that (cf. (10)) for all $i \in [n], c \in [k]$, we have

$$\hat{\mathbf{w}}_c^T\mathbf{x}_i = z_{ci} := \begin{cases} (k-1)/k & , \ c = y_i \\ -1/k & , \ c \neq y_i \end{cases}. \tag{6}$$

**Theorem 1.** *For a multiclass separable dataset with feature matrix $\mathbf{X} = [\mathbf{x}_1, \mathbf{x}_2, \ldots, \mathbf{x}_n] \in \mathbb{R}^{p \times n}$ and label matrix $\mathbf{Y} = [\mathbf{v}_1, \mathbf{v}_2, \ldots, \mathbf{v}_k]^T \in \mathbb{R}^{k \times n}$, let $\mathbf{W}_{SVM} = [\hat{\mathbf{w}}_1, \hat{\mathbf{w}}_2, \ldots, \hat{\mathbf{w}}_k]^T$ be the multiclass SVM solution in (3). For each class $c \in [k]$ define vectors $\mathbf{z}_c \in \mathbb{R}^n$ such that*

$$\mathbf{z}_c = \mathbf{v}_c - (1/k)\mathbf{1}_n, \ c \in [k]. \tag{7}$$

*Assume that the Gram matrix $\mathbf{X}^T\mathbf{X}$ is invertible and that the following condition holds*

$$\mathbf{z}_c \odot (\mathbf{X}^T\mathbf{X})^{-1}\mathbf{z}_c > \mathbf{0}, \quad \forall c \in [k]. \tag{8}$$

*Then, the SVM solution $\mathbf{W}_{SVM}$ is such that all the constraints in (3) are active. That is,*

$$(\hat{\mathbf{w}}_{y_i} - \hat{\mathbf{w}}_c)^T\mathbf{x}_i = 1, \ \forall c \neq y_i, c \in [k], \text{ and } \ \forall i \in [n]. \tag{9}$$

*Moreover, it holds that*

$$\mathbf{X}^T\hat{\mathbf{w}}_c = \mathbf{z}_c, \ \forall c \in [k]. \tag{10}$$

For $k = 2$ classes, it can be checked that Eqn. (8) reduces to the condition in Eqn.(22) of [MNS+20] for the binary SVM. Compared to the binary setting, the conclusion for multiclass is richer: provided that Eqn. (8) holds, not only do we show that all data points are support vectors, but also that they satisfy a set of symmetric OvA-type constraints. The proof of Eqn. (10) is particularly subtle and involved: unlike in the binary case, it does *not* follow directly from a complementary slackness condition on the dual of the multiclass SVM. We provide a short proof sketch in Section 3.1.1 and defer details to the supplementary material (SM).

We make the following additional remarks on the interpretation of Eqn. (10). First, our proof shows a somewhat stronger conclusion: when inequality (8) holds, the multiclass SVM solutions $\hat{\mathbf{w}}_c, c \in [k]$ are same as the solutions to the following *symmetric OvA-type classifier* (cf. Eqn. (4)):

$$\min_{\mathbf{w}_c} \frac{1}{2}\|\mathbf{w}_c\|_2^2 \qquad \text{sub. to } \ \mathbf{x}_i^T\mathbf{w}_c \begin{cases} \geq (k-1)/k & , y_i = c, \\ \leq -1/k & , y_i \neq c, \end{cases} \ \forall i \in [n], \tag{11}$$

for all $c \in [k]$. The OvA-type classifier above can be interpreted as a binary cost-sensitive SVM classifier [IMSV19] that enforces the margin corresponding to all other classes to be $(k-1)$ times smaller compared to the margin for class $c$.

The second remark regarding (10) is crucial for the rest of this paper. Precisely, (10) shows that when (8) holds, then the multiclass SVM solution $\mathbf{W}_{\text{SVM}}$ has the same classification error as that of the minimum-norm interpolating solution. This conclusion, stated as a corollary below, drives our classification error analysis in Section 4.

**Corollary 1** (SVM=MNI). *Under the same assumptions as in Theorem 1, and provided that the inequality in Eqn. (8) holds, it holds that $\mathbb{P}_{e|c}(\mathbf{W}_{SVM}) = \mathbb{P}_{e|c}(\mathbf{W}_{MNI})$ for all $c \in [k]$. Thus, the total classification errors of both solutions are equal: $\mathbb{P}_e(\mathbf{W}_{SVM}) = \mathbb{P}_e(\mathbf{W}_{MNI})$.*

*Proof sketch.* First, it follows from Eqn. (10) that $\hat{\mathbf{w}}_c, c \in [k]$ coincides with the unique solution of a MNI classifier on shifted labels, given by $\widetilde{\mathbf{w}}_c = \widehat{\mathbf{w}}_c := \mathbf{X}(\mathbf{X}^T\mathbf{X})^{-1}\mathbf{z}_c$. Second, using the affine relation between $\mathbf{z}_c$ and $\mathbf{v}_c$ in Eqn. (7), we get $\mathbb{P}_{e|c}(\mathbf{W}_{MNI}) = \mathbb{P}_{e|c}(\widetilde{\mathbf{W}}_{MNI})$, where we denote $\widetilde{\mathbf{W}}_{MNI} = [\widetilde{\mathbf{w}}_1, \ldots, \widetilde{\mathbf{w}}_k]$. This completes the proof of the corollary. More details given in the SM. $\square$

### 3.1.1 Proof sketch of Theorem 1

To prove Theorem 1, we constructed a new parameterization of the dual of the multiclass SVM (given in Eqn. (14)). Letting dual variables $\{\lambda_{c,i}\}$ for every $i \in [n], c \in [k] : c \neq y_i$ corresponding to the constraints on the primal form in (3), the standard form of the dual of multiclass SVM is written as

$$\max_{\lambda_{c,i} \geq 0} \sum_{i \in [n]} \left( \sum_{c \neq y_i} \lambda_{c,i} \right) - \frac{1}{2} \sum_{c \in [k]} \left\| \sum_{i \in [n]:y_i=c} \left( \sum_{c' \neq y_i} \lambda_{c',i} \right) \mathbf{x}_i - \sum_{i \in [n]:y_i \neq c} \lambda_{c,i}\mathbf{x}_i \right\|_2^2. \quad (12)$$

Let $\hat{\lambda}_{c,i}$ be the maximizers in Eqn. (12). By complementary slackness, we have

$$\hat{\lambda}_{c,i} > 0 \implies (\hat{\mathbf{w}}_{y_i} - \hat{\mathbf{w}}_c)^T \mathbf{x}_i = 1. \quad (13)$$

Thus, to prove Eqn. (9), it will suffice showing that $\hat{\lambda}_{c,i} > 0, \forall i \in [n], c \in [k] : c \neq y_i$ provided that Eqn. (8) holds. The challenge is that it is hard to work directly with (12) because the variables $\lambda_{c,i}$ are coupled within the objective. Our key idea is to re-parameterize the dual objective in terms of new variables $\beta_{c,i}$ and of coefficients involving the vectors $\mathbf{z}_c$ we introduced in Eqn. (7). Deferring the detailed derivations to the SM, we can show that (12) is equivalent to the following program:

$$\max_{\boldsymbol{\beta}_c \in \mathbb{R}^n, c \in [k]} \quad \sum_{c \in [k]} \boldsymbol{\beta}_c^T \mathbf{z}_c - \frac{1}{2}\|\mathbf{X}\boldsymbol{\beta}_c\|_2^2 \quad (14)$$

$$\text{sub. to} \quad \beta_{y_i,i} = -\sum_{c \neq y_i} \beta_{c,i}, \ \forall i \in [n] \quad \text{and} \quad \boldsymbol{\beta}_c \odot \mathbf{z}_c \geq \mathbf{0}, \forall c \in [k],$$

where, for each $c \in [k]$ we let $\boldsymbol{\beta}_c = [\beta_{c,1}, \beta_{c,2}, \ldots, \beta_{c,n}] \in \mathbb{R}^n$. Moreover, the new dual variables are related to the original ones in that

$$z_{c,i}\beta_{c,i} > 0 \iff \lambda_{c,i} > 0, \quad \text{for all } c \in [k] \text{ and } i \in [n] : y_i \neq c. \quad (15)$$

The next step is to consider the *unconstrained* maximizer in (14), that is $\hat{\boldsymbol{\beta}}_c = (\mathbf{X}^T\mathbf{X})^{-1}\mathbf{z}_c, \forall c \in [k]$, and show that $\hat{\boldsymbol{\beta}}_c, c \in [k]$ is feasible in (14). Skipping the detailed argument here, by doing so, we prove that $\hat{\boldsymbol{\beta}}_c, c \in [k]$ is in fact the unique optimal solution of (14). But now, realizing that Eqn. (8) is equivalent to $\mathbf{z}_c \odot \hat{\boldsymbol{\beta}}_c > 0$, we have found that $\hat{\boldsymbol{\beta}}_c, c \in [k]$ further satisfies the $n$ *strict* inequality constraints in (14). Thus, from Eqn. (15), the original dual variables $\{\lambda_{c,i}\}$ are also all strictly positive, which completes the proof of the first part of the theorem (Eqn. (9)).

Next, we outline the proof of Eqn. (10). We consider the OvA-classifier in (11). The proof has two steps. First, using similar arguments to what was done above, we show that when Eqn. (8) holds, then all the inequality constraints in (11) are active at the optimal. That is, the minimizers $\mathbf{w}_{\text{sym-OvA},c}$ of (11) satisfy Eqn. (10). Second, to prove that Eqn. (10) is satisfied by the minimizers $\hat{\mathbf{w}}_c$ of the multiclass SVM in Eqn. (3), we need to show that $\mathbf{w}_{\text{sym-OvA},c} = \hat{\mathbf{w}}_c$ for all $c \in [k]$. We do this by showing that, under Eqn. (8), the duals of (3) and (11) are equivalent. By strong duality, the optimal costs of the primal problems are also the same. Then, because the objective is the same for the two primals and because $\mathbf{w}_c^*$ is feasible and (3) is strongly convex, we can conclude with the desired.

## 3.2 Connection to effective overparameterization

Theorem 1 establishes a *deterministic* condition that applies to *any* multiclass separable dataset as long as the data matrix $\mathbf{X}$ is full-rank. In this subsection, we show that the inequality (8) occurs with high-probability under both the GMM and MLM models for data, with sufficient overparameterization.

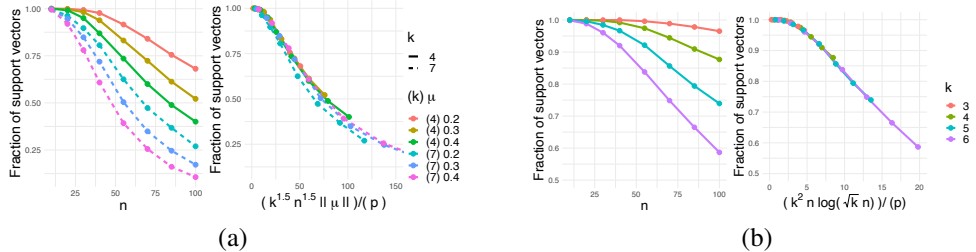

(a)                (b)

Figure 2: Fraction of support vectors satisfying Eqn. (10). (a) GMM: $k = 4$ and 7, (b) MLM: $k = 3, 4, 5, 6$. In (a), "(4) 0.2" means 4 classes and $\|\boldsymbol{\mu}\|_2/\sqrt{p} = 0.2$. The curves nearly overlap when plotted versus $k^{1.5}n^{1.5}\|\boldsymbol{\mu}\|_2/p$ as predicted by the second condition in Eqn. (16) of Theorem 2. In (b), the curves overlap when plotted versus $k^2 n \log(\sqrt{k}n)/p$ as predicted by Thm. 4.

### 3.2.1 Gaussian mixture model

We focus on an equal-energy, equal-prior setting for ease of exposition. Our proofs extend rather naturally to more general settings, but the results are more complicated to state and offer no new insights for our purpose.

**Assumption 1** (Equal energy/prior). *The mean vectors have equal energy and the priors are equal, i.e. we have $\|\boldsymbol{\mu}\|_2 := \|\boldsymbol{\mu}_c\|_2$ and $\pi_c = \pi = 1/k$, for all $c \in [k]$.*

**Theorem 2.** *Assume that the training set follows a multiclass GMM with $\boldsymbol{\Sigma} = \mathbf{I}_p$ and Assumption 1, and that the number of training samples $n$ is large enough. There exist constants $c_1, c_2, c_3 > 1$ and $C_1, C_2 > 1$ such that inequality (8) holds with probability at least $1 - \frac{c_1}{n} - c_2 k e^{-\frac{n}{c_3 k^2}}$, provided that*

$$p > C_1 k^3 n \log(kn) + n - 1 \quad \text{and} \quad p > C_2 k^{1.5} n^{1.5} \|\boldsymbol{\mu}\|_2. \tag{16}$$

Our theorem establishes a set of two conditions under which inequality (8) and the conclusions of Theorem 1 hold, i.e. $\mathbf{W}_{\text{SVM}} = \mathbf{W}_{\text{MNI}}$. The first condition requires sufficient overparameterization $p = \Omega(k^3 n \log(kn))$, while the second one requires that the signal strength is not too large. Intuitively, we can understand these conditions as follows. Note that inequality (8) is satisfied provided that the inverse Gram matrix $(\mathbf{X}^T\mathbf{X})^{-1}$ is "close" to identity, or any other positive-definite diagonal matrix. (This is the proof strategy that is also followed in [MNS+20] for the case of Gaussian features; here, we show this for the more difficult mixture-of-Gaussians case.) Recall from Eqn. (1) that $\mathbf{X} = \mathbf{M}\mathbf{Y} + \mathbf{Q} = \sum_{j=1}^{k} \boldsymbol{\mu}_j \mathbf{v}_j^T + \mathbf{Q}$ where $\mathbf{Q}$ is a $p \times n$ standard Gaussian matrix. Our theorem's first condition is sufficient for $(\mathbf{Q}^T\mathbf{Q})^{-1}$ to have the desired property; the major technical challenge is that $(\mathbf{X}^T\mathbf{X})^{-1}$ involves additional terms that intricately depend on the label matrix $\mathbf{Y}$ itself. Our key technical contribution is showing that these extra terms do *not* drastically change the desired behavior, provided that the norms of the mean vectors are well controlled. At a high-level we accomplish this with a recursive argument as follows. Denote $\mathbf{X}_0 = \mathbf{Q}$ and $\mathbf{X}_i = \sum_{j=1}^{i} \boldsymbol{\mu}_j \mathbf{v}_j^T + \mathbf{Q}$ for $i \in [k]$. Then, at each stage $i$ of the recursion, we show how to bound quadratic forms involving $(\mathbf{X}_i^T\mathbf{X}_i)^{-1}$ using bounds established previously at stage $i-1$ on quadratic forms involving $(\mathbf{X}_{i-1}^T\mathbf{X}_{i-1})^{-1}$. A critical property for the success of our proof strategy is the observation that the rows of $\mathbf{Y}$ are always orthogonal, that is, $\mathbf{v}_i^T\mathbf{v}_j = 0$, for $i \neq j$. The complete proof of the theorem is given in the SM.

Next, we present numerical results that confirm our theoretical statement. We also discuss the tightness of the two sufficient conditions in Eqn. (16). Throughout the paper, in all our figures, we show averages over 100 Monte-Carlo realizations. In Fig. 2(a), we plot the *fraction of training points in the multiclass SVM satisfying Eqn. (10)* as a function of training size $n$ for $k = 4$ and $k = 7$ classes (please see SM for other experiment details and more results). To verify the second condition in Eqn. (16), Fig. 2(a) also plots the same set of curves over a re-scaled axis $k^{1.5}n^{1.5}\|\boldsymbol{\mu}\|_2/p$. The 6 curves corresponding to different settings nearly overlap in this new scaling, which suggests the correct order of the corresponding condition. We conjecture that our second condition is tight up to an extra $\sqrt{n}$ factor which we believe is an artifact of the analysis. We also believe that the $k^3$ factor in the first condition can be relaxed slightly to $k^2$ (as is done for the MLM case; see Fig. 2(b)).

### 3.2.2 Multinomial logistic model

We now consider the MLM data and the anisotropic setting. The eigendecomposition of the covariance matrix is given by $\boldsymbol{\Sigma} = \sum_{i=1}^{p} \lambda_i \mathbf{u}_i \mathbf{u}_i^T$, where $\boldsymbol{\lambda} = [\lambda_1, \cdots, \lambda_p]$. Following [HMX21], we also define the effective dimensions $d_2 := \|\boldsymbol{\lambda}\|_1^2 / \|\boldsymbol{\lambda}\|_2^2$ and $d_\infty := \|\boldsymbol{\lambda}\|_1 / \|\boldsymbol{\lambda}\|_\infty$. The following result contains sufficient conditions for the SVM and MNI solutions to coincide.

**Theorem 3.** *Assume $n$ training samples following the MLM defined in* (2). *There exist constants $c$ and $C_1, C_2 > 1$ such that inequality* (8) *holds with probability at least $(1 - \frac{c}{n})$ provided that $d_\infty > C_1 k^2 n \log(kn)$ and $d_2 > C_2(\log(kn) + n)$. In fact, the only conditions we require on the generated labels is conditional independence.*

The sufficient conditions in Theorem 3 require that the spectral structure in the covariance matrix $\boldsymbol{\Sigma}$ has sufficiently slowly decaying eigenvalues (corresponding to sufficiently large $d_2$), and that it is not too "spiky" (corresponding to sufficiently large $d_\infty$). For the special case of $k = 2$ classes, our conditions reduce to those in [HMX21] for binary classification; in fact under the MLM model we can leverage a more sophisticated deterministic equivalence to Eqn. (8) provided in that work. The dominant dependence on $k$, given by $k^2$, is a byproduct of the "unequal" margin in (6). Fig. 2(b) empirically verifies the tightness of this factor. For the isotropic case $\boldsymbol{\Sigma} = \mathbf{I}_p$, we can prove a slightly sharper result in logarithmic factors, which we state next.

**Theorem 4.** *Assume $n$ samples from the MLM with $\boldsymbol{\Sigma} = \mathbf{I}_p$. There exist a constant $c > 1$ such that inequality* (8) *holds with probability at least $(1 - \frac{c}{n})$ provided that $p > 10k^2 n \log(\sqrt{k}n) + n - 1$.*

Our numerical results in Fig. 2(b) suggest that this sufficient condition is order-wise tight. Specifically, in Fig. 2, we fixed $p = 1000$, varied $n$ from 10 to 100 and the numbers of classes from $k = 3$ to $k = 6$. We chose orthogonal mean vectors for each class with equal energy $\|\boldsymbol{\mu}\|_2^2 = p$. Fig. 2(b) shows the fraction of training points in the multiclass SVM satisfying Eqn. (10) as a function of $n$. Clearly, smaller $k$ results in higher proportion of support vectors for the same number of measurements $n$. To verify the condition in Theorem 4, Fig. 2(b) plots the same curves over a re-scaled axis $k^2 n \log(\sqrt{k}n)/p$ (as suggested by Thm.4). These curves nearly overlap.

## 4 Generalization bounds and benign overfitting

In this section, we derive non-asymptotic bounds on the error of the MNI classifier, and discuss sufficient conditions for the multiclass SVM to satisfy benign overfitting. We focus on the case of GMM data due to space constraints, and discuss corresponding results on MLM data in the SM.

### 4.1 Generalization bounds for the MNI classifier

We present classification error bounds under the additional assumption of orthogonal means for ease of exposition — this can be relaxed with some additional work as described in the SM.

**Assumption 2** (Orthogonal means)**.** *In addition to Assumption* 1, *assume that the means are orthogonal, that is $\boldsymbol{\mu}_c^T \boldsymbol{\mu}_j = 0$, for all $c \neq j \in [k]$.*

**Theorem 5.** *Let Assumption* 2 *and condition in Eqn.*(16) *hold. Further assume constants $C_1, C_2, C_3 > 1$ such that $\left(1 - \frac{C_1}{\sqrt{n}} - \frac{C_2 n}{p}\right)\|\boldsymbol{\mu}\|_2 > C_3 \sqrt{k}$. Then, there exist additional constants $c_1, c_2, c_3$ and $C_4 > 1$ such that $\mathbb{P}_{e|c} \leq (k-1) \exp\left(- \|\boldsymbol{\mu}\|_2^2 \frac{\left((1 - \frac{C_1}{\sqrt{n}} - \frac{C_2 n}{p})\|\boldsymbol{\mu}\|_2 - C_3\sqrt{k}\right)^2}{C_4(\|\boldsymbol{\mu}\|_2^2 + \frac{kp}{n})}\right)$ with probability at least $1 - \frac{c_1}{n} - c_2 k e^{-\frac{n}{c_3 k^2}}$, for every $c \in [k]$. Moreover, the same bound holds for the total classification error $\mathbb{P}_e$.*

For large enough and finite $n$, our bound reduces to the results in [WT21, CGB21] when $k = 2$ (with slightly different constant numbers). There are two major challenges in the proof, which is presented in the SM. First, in contrast to the binary case the classification error does *not* simply reduce to bounding correlations between vector means $\boldsymbol{\mu}_c$ and their estimators $\hat{\mathbf{w}}_c$. Second, just as in the proof of Theorem 2, technical complications arise from the multiple mean components in $\mathbf{X}$. We use a variant of the recursion-based argument described in Section 3.2.1 to obtain our final bound.

## 4.2 Conditions for benign overfitting

In our results thus far, we have studied the classification error of the MNI classifier (Theorem 5), and shown equivalence of the multiclass SVM and MNI solutions (Theorems 1, 2 and Corollary 1). Combining these results, we now provide sufficient conditions under which the classification error of the multiclass SVM solution (also of the MNI) approaches 0 as model size $p$ increases.

**Corollary 2.** *Let the same assumptions as in Theorem 5 hold. Then, for finite number of classes $k$ and finite sample size $n$, there exist positive constants $c_i$'s and $C_i$'s $> 1$, such that the multiclass SVM classifier $\mathbf{W}_{SVM}$ in (3) satisfies the symmetric interpolation constraint in (10) and its total classification error approaches 0 as $p \to \infty$ with probability at least $1 - \frac{c_1}{n} - c_2 k e^{-\frac{n}{c_3 k^2}}$, provided $\|\boldsymbol{\mu}\|_2 = \Theta(p^\beta)$ for $\beta \in (1/4, 1)$.*

We compare our result with the binary case result in [CL21, WT21, CGB21]. When $k$ and $n$ are both finite, condition $\|\boldsymbol{\mu}\|_2 = \Theta(p^\beta)$ for $\beta \in (1/4, 1)$ is the same as the binary result. Note that, like in the binary case, Corollary 2 applies beyond the regime in which margin-bounds would be predictive of good generalization, which would require $\beta \in (1/2, 1)$.

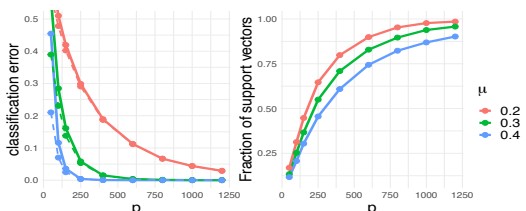

We now present numerical illustrations validating our results. We set the number of classes $k = 4$, fix $n = 40$, and vary $p = 50, \ldots, 1200$ to guarantee sufficient overparameterization. We consider the case of orthogonal and equal-norm mean vectors $\|\boldsymbol{\mu}\|_2 = \mu\sqrt{p}$, with $\mu = 0.2, 0.3$ and $0.4$. In Fig. 3, we plot the classification error as a function of $p$ for both MNI

Figure 3: The plot shows the classification error and fraction of support vectors with $k = 4$. We can see the classification errors approach 0 and the fractions of support vectors approach 1 as $p$ gets larger. Different colors correspond to different mean norms.

estimates (solid lines) and multiclass SVM solutions (dashed lines). As we now expect, the solid and dashed curves almost overlap. Further, as $p$ increases, we see that the classification error decreases towards zero. Fig. 3 shows the fraction of support vectors satisfying (10) among all the constraints in (3). We see that the classification error goes to zero very fast when $\mu$ is large, but the proportion of support vectors increases at a slow rate. In contrast, when $\mu$ is small, the proportion of support vectors increases fast, but the classification error decreases slowly.

## 5 Conclusion and future work

Our work provides, to the best of our knowledge, the first results characterizing a) equivalence of loss functions, and b) generalization of interpolating solutions in multiclass settings. Like almost all benign overfitting analysis, our techniques are tailored to high-dimensional linear models with Gaussian or independent sub-Gaussian features. Extending these results to kernel machines and other nonlinear settings is of substantial interest. It would also be interesting to explore the potential connections of the symmetric structure shown in Theorem 1 with the recently discovered neural collapse phenomenon on deep nets [PHD20].

## 6 Acknowledgements

This work is partially supported by the NSF under Grant Number CCF-2009030 and by a CRG8 award from KAUST. C. Thrampoulidis would also like to recognize his affiliation with the University of California, Santa Barbara. The authors would like to thank the anonymous reviewers for helpful discussion and suggestions.

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
