# Supplementary material for:
# "Benign Overfitting in Multiclass Classification: All Roads Lead to Interpolation"

**Ke Wang**
Department of Statistics and Applied Probability
University of California, Santa Barbara
Santa Barbara, CA 93106
kewang01@ucsb.edu

**Vidya Muthukumar**
School of Electrical and Computer Engineering & Industrial and Systems Engineering
Georgia Institute of Technology
Atlanta, GA 30332
vmuthukumar8@gatech.edu

**Christos Thrampoulidis**
Department of Electrical and Computer Engineering
University of British Columbia
Vancouver, BC Canada V6T 1Z4
cthrampo@ece.ubc.ca

## Abstract

The growing literature on "benign overfitting" in overparameterized models has been mostly restricted to regression or binary classification settings; however, most success stories of modern machine learning have been recorded in multiclass settings. Motivated by this discrepancy, we study benign overfitting in multiclass linear classification. Specifically, we consider the following popular training algorithms on separable data: (i) empirical risk minimization (ERM) with cross-entropy loss, which converges to the multiclass support vector machine (SVM) solution; (ii) ERM with least-squares loss, which converges to the min-norm interpolating (MNI) solution; and, (iii) the one-vs-all SVM classifier. Our first key finding is that under a simple sufficient condition, *all* three algorithms lead to classifiers that interpolate the training data and have equal accuracy. When the data is generated from Gaussian mixtures or a multinomial logistic model, this condition holds under high enough effective overparameterization. Second, we derive novel error bounds on the accuracy of the MNI classifier, thereby showing that all three training algorithms lead to benign overfitting under sufficient overparameterization. Ultimately, our analysis shows that good generalization is possible for SVM solutions beyond the realm in which typical margin-based bounds apply.

## Organization of the supplementary material

The supplementary material (SM) is organized as follows.

35th Conference on Neural Information Processing Systems (NeurIPS 2021).

1. **Section A:** We present additional numerical experiments validating our theoretical findings throughout the paper. For completeness, we also present error bars computed over Monte Carlo realizations.

2. **Section B:** We provide a detailed proof of our key finding in Theorem 1.

3. **Section C:** We prove Theorem 2 on multiclass SVM interpolation for GMM data.

4. **Section D:** We prove Theorems 3 and 4 on multiclass SVM interpolation for MLM data.

5. **Section E:** We prove Theorems 3 and 4 on classification error of the MNI classifier for GMM and MLM data.

6. **Section F:** Here we derive recursive formulas for computing quadratic forms, which are required for the proofs in Sections C and E for GMM data.

7. **Section G:** We derive conditions under which the OvA-classifier interpolates the data for both GMM and MLM data. Thus, all three classifiers the (i) MNI, (ii) multiclass-SVM, and (iii) OvA-SVM lead to interpolation.

To ease readability and accessibility, we also opted to keep the main manuscript. The SM starts at page 18.

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

# Contents

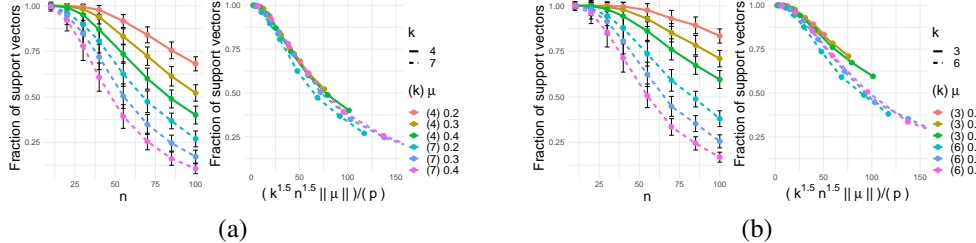

$$(a) \qquad\qquad (b)$$

Figure 4: Fraction of support vectors satisfying Equation (10). The error bars show the standard deviation. (a) $k = 4$ and 7, (b) $k = 3$ and 6. On the legend, "(4) 0.3" means 4 classes and $\|\boldsymbol{\mu}\|_2/\sqrt{p} = 0.2$. The curves nearly overlap when plotted versus $k^{1.5}n^{1.5}\|\boldsymbol{\mu}\|_2/p$ as predicted by the second condition in Equation (16) of Theorem 2.

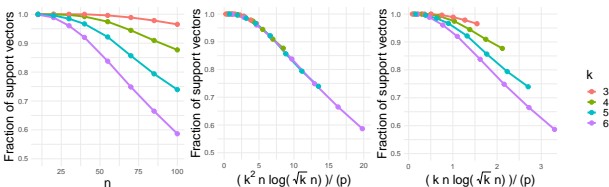

Figure 5: The plot shows the fraction of support vectors under MLM with different number of classes. The curves overlap when plotted versus $k^2 n \log(\sqrt{k}n)/p$ (Middle plot) as predicted by Theorem 4.

## A   Experiment details and additional results

We first present numerical results that confirm our conclusions in Theorem 2. We also discuss the tightness of the two sufficient conditions in Eqn.(16). Recall that throughout the paper, in all our figures, we show averages over 100 Monte-Carlo realizations. In Fig. 4(a) (same as Figure 2(a), repeated here for convenience), we solved the multiclass SVM and plotted the *fraction of support vectors satisfying Equation* (10) as a function of training size $n$. The error bars show the standard deviation at each point. We fixed dimension $p = 1000$ and class priors $\pi = \frac{1}{k}$. To study how the outcome depends on the number of classes $k$ and mean strength $\|\boldsymbol{\mu}\|_2$, we experimented with two values of class number $k = 4, 7$ and three equal-energy scenarios where $\forall c \in [k] : \|\boldsymbol{\mu}_c\|_2 = \|\boldsymbol{\mu}\|_2 = \mu\sqrt{p}$ with $\mu = 0.2, 0.3, 0.4$. Fig. 4(a)(Left) shows how the fraction of support vectors changes with $n$. Observe that smaller $\mu$ results in larger proportion of support vectors for the same $n$. To verify our theorem's second condition in Equation (16), Fig. 4(a)(Right) plots the same set of curves over a re-scaled axis $k^{1.5}n^{1.5}\|\boldsymbol{\mu}\|_2/p$. The 6 curves corresponding to different settings nearly overlap in this new scaling, which suggests the correct order of the corresponding condition. In Fig. 4(b), we repeat the experiment in Fig. 4(a) for different values of $k = 3$ and $k = 6$. Again, these curves nearly overlap when the x-axis is scaled according to the second condition in Equation (16).

We next confirm our results in Theorem 3 in Fig. 5. (The first two subplots are same as Figure 2(b), repeated here for convenience.) Here, we fixed $p = 1000$, varied $n$ from 10 to 100 and the numbers of classes from $k = 3$ to $k = 6$. We chose orthogonal mean vectors for each class with equal energy $\|\boldsymbol{\mu}\|_2^2 = p$ and solved multiclass SVM. Fig. 5(Left) shows the fraction of support vectors satisfying Equation (10) as a function of $n$. Clearly, smaller $k$ results in higher proportion of support vectors with the desired property for the same number of measurements $n$. To verify the condition in Theorem 4, Fig. 5(Middle) plots the same curves over a re-scaled axis $k^2 n \log(\sqrt{k}n)/p$ (as suggested by Theorem 4). We additionally draw the same curves over $kn \log(\sqrt{k}n)/p$ in Fig. 4(Right). Note the overlap of the curves in the middle plot.

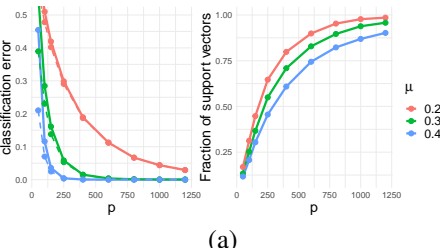 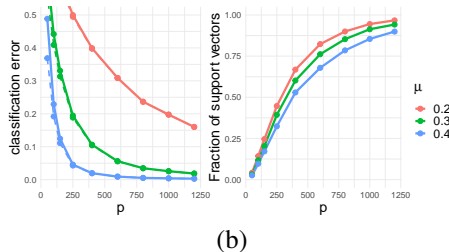

|  (a)  |  (b)  |

Figure 6: The plot shows the classification error and fraction of support vectors. (a) sets $k = 4$ and (b) sets 6. The mean vector $\|\boldsymbol{\mu}\|_2 = \mu\sqrt{p}$, where $\mu = 0.2, 0.3$ and $0.4$. We can see the classification errors approach $0$ and the fractions of support vectors approach $1$ as $p$ gets larger.

Finally, we present numerical illustrations validating our results in Corollary 2. In Fig. 6(a), we set the number of classes $k = 4$. To guarantee sufficient overparameterization, we fix $n = 40$ and vary $p$ from 50 to 1200. In Fig. 6(a)(Left), we plot the classification error as a function of $p$ for both MNI estimates (solid lines) and multiclass SVM solutions (dashed lines). Different colors correspond to different mean norms. The solid and dashed curves almost overlap as predicted from our results in Section 3. We simulated 3 different settings for the mean matrices: each has orthogonal and equal-norm mean vectors $\|\boldsymbol{\mu}\|_2 = \mu\sqrt{p}$, with $\mu = 0.2, 0.3$ and $0.4$. We verify that as $p$ increases, the classification error decreases towards zero. Fig. 6(a)(Right) shows the fraction of support vectors satisfying (10) among all the constraints in (3). The probabilities approach 1 as $p$ gets larger. Also, we see that the classification error goes to zero very fast when $\mu$ is large, but then the proportion of support vectors increases at a slow rate. In contrast, when $\mu$ is small, the proportion of support vectors increases fast, but the classification error decreases slowly. In Fig. 6(b), we use the same setting as in Fig. 6(a) except for the number of classes $k = 6$ and $n = 30$. The rate at which classification error goes to zero and the proportion of support vectors increase, both become slower as now there are more classes.

## B    Proofs for Section 3.1

In this section, we provide the proofs of deterministic equivalence between the multiclass SVM, OvA SVM and MNI classifiers. We begin with the proof of Theorem 1.

### B.1    Detailed proof of Theorem 1

We start by writing the dual of the multiclass SVM, repeated here for convenience.

$$\min_{\mathbf{W}} \frac{1}{2}\|\mathbf{W}\|_F^2 \quad \text{sub. to } (\mathbf{w}_{y_i} - \mathbf{w}_c)^\top \mathbf{x}_i \geq 1, \ \forall i \in [n], c \in [k] : c \neq y_i. \tag{17}$$

We have dual variables $\{\lambda_{c,i}\}$ for every $i \in [n], c \in [k] : c \neq y_i$ corresponding to the constraints on the primal form above. Then, the dual of the multiclass SVM takes the form

$$\max_{\lambda_{c,i} \geq 0} \sum_{i \in [n]} \Big(\sum_{\substack{c \in [k] \\ c \neq y_i}} \lambda_{c,i}\Big) - \frac{1}{2}\sum_{c \in [k]} \Big\| \sum_{i \in [n]: y_i = c} \Big(\sum_{\substack{c' \in [k] \\ c' \neq y_i}} \lambda_{c',i}\Big)\mathbf{x}_i - \sum_{i \in [n]: y_i \neq c} \lambda_{c,i}\mathbf{x}_i \Big\|_2^2. \tag{18}$$

Let $\hat{\lambda}_{c,i}, i \in [n], c \in [k] : c \neq y_i$ be the maximizers in Equation (18). By complementary slackness, we have

$$\hat{\lambda}_{c,i} > 0 \implies (\hat{\mathbf{w}}_{y_i} - \hat{\mathbf{w}}_c)^\top \mathbf{x}_i = 1. \tag{19}$$

Thus, it will suffice to prove that $\hat{\lambda}_{c,i} > 0, \forall i \in [n], c \in [k] : c \neq y_i$ provided that (8) holds.

**Key alternative parameterization of the dual.** It is challenging to work directly with Equation (18) because the variables $\lambda_{c,i}$ are coupled in the objective function. Our main idea is to re-parameterize

the dual objective in terms of new variables $\{\beta_{c,i}\}$, which we define as follows for all $c \in [k]$ and $i \in [n]$:

$$\beta_{c,i} = \begin{cases} \sum_{c' \neq y_i} \lambda_{c',i} & , y_i = c, \\ -\lambda_{c,i} & , y_i \neq c. \end{cases} \tag{20}$$

For each $c \in [k]$, we denote $\boldsymbol{\beta}_c = [\beta_{c,1}, \beta_{c,2}, \ldots, \beta_{c,n}] \in \mathbb{R}^n$. With these, we show that the dual objective becomes

$$\sum_{c \in [k]} \boldsymbol{\beta}_c^\top \mathbf{z}_c - \frac{1}{2} \sum_{c \in [k]} \Big\| \sum_{i \in [n]} \beta_{c,i} \mathbf{x}_i \Big\|_2^2 = \sum_{c \in [k]} \boldsymbol{\beta}_c^\top \mathbf{z}_c - \frac{1}{2} \|\mathbf{X}\boldsymbol{\beta}_c\|_2^2. \tag{21}$$

The equivalence of the quadratic term in $\boldsymbol{\beta}$ is straightforward. To show the equivalence of the linear term in $\boldsymbol{\beta}$, we denote $A := \sum_{i \in [n]} \Big( \sum_{c \in [k], c \neq y_i} \lambda_{c,i} \Big)$, and simultaneously get

$$A = \sum_{i \in [n]} \beta_{y_i,i} \qquad \text{and} \qquad A = \sum_{i \in [n]} \sum_{c \neq y_i} (-\beta_{c,i}),$$

by the definition of variables $\{\beta_{c,i}\}$ in Equation (20). Then, we have

$$A = \frac{k-1}{k} \cdot A + \frac{1}{k} \cdot A = \frac{k-1}{k} \sum_{i \in [n]} \beta_{y_i,i} + \frac{1}{k} \sum_{i \in [n]} \sum_{c \neq y_i} (-\beta_{c,i})$$

$$\overset{\text{(i)}}{=} \sum_{i \in [n]} \mathbf{z}_{y_i,i} \beta_{y_i,i} + \sum_{i \in [n]} \sum_{c \neq y_i} \mathbf{z}_{c,i} \beta_{c,i}$$

$$= \sum_{i \in [n]} \sum_{c \in [k]} \mathbf{z}_{c,i} \beta_{c,i} = \sum_{c \in [k]} \boldsymbol{\beta}_c^\top \mathbf{z}_c.$$

Above, inequality (i) follows from the definition of $\mathbf{z}_c$ in Equation (7), rewritten coordinate-wise as:

$$z_{c,i} = \begin{cases} \frac{k-1}{k}, & y_i = c, \\ -\frac{1}{k}, & y_i \neq c. \end{cases}$$

Thus, we have shown that the objective of the dual can be rewritten in terms of variables $\{\beta_{c,i}\}$. After rewriting the constraints in terms of $\{\beta_{c,i}\}$, the dual of the SVM (Equation (3)) can be equivalently written as:

$$\max_{\boldsymbol{\beta}_c \in \mathbb{R}^n, c \in [k]} \quad \sum_{c \in [k]} \boldsymbol{\beta}_c^\top \mathbf{z}_c - \frac{1}{2} \|\mathbf{X}\boldsymbol{\beta}_c\|_2^2 \tag{22}$$

$$\text{sub. to} \qquad \beta_{y_i,i} = -\sum_{c \neq y_i} \beta_{c,i}, \ \forall i \in [n] \quad \text{and} \quad \boldsymbol{\beta}_c \odot \mathbf{z}_c \geq \mathbf{0}, \forall c \in [k].$$

Note that the first constraint above ensures consistency with the definition of $\boldsymbol{\beta}_c$ in Eqn. (20). The second constraint guarantees the non-negativity constraint of the original dual variables in (18), because we have

$$\beta_{c,i} z_{c,i} = \frac{\lambda_{c,i}}{k} \text{ for all } i \in [n], c \in [k] : c \neq y_i.$$

Consequently, we have

$$\beta_{c,i} z_{c,i} \geq 0 \iff \lambda_{c,i} \geq 0 \tag{23}$$

for all $c \in [k]$ and $i \in [n] : y_i \neq c$. In fact, the equivalence above also holds with the inequalities replaced by strict inequalities. Also note that the second constraint for $c = y_i$ yields $\frac{k-1}{k} \sum_{c' \neq y_i} \lambda_{c',i} \geq 0$, which is automatically satisfied when Equation (23) is satisfied. Thus, these constraints are redundant.

**Proof of Equation** (9). Let $\hat{\boldsymbol{\beta}}_c, c \in [k]$ be the unconstrained maximizer in (22), i.e.

$$\hat{\boldsymbol{\beta}}_c = (\mathbf{X}^\top \mathbf{X})^{-1} \mathbf{z}_c, \ \forall c \in [k].$$

We will show that the unconstrained maximizer $\hat{\boldsymbol{\beta}}_c, c \in [k]$ is feasible in the constrained program in (22). Thus, it is its unique optimal solution.

To prove this, we will first prove that $\hat{\boldsymbol{\beta}}_c, c \in [k]$ satisfies the $n$ equality constraints in (22). For convenience, let $\mathbf{g}_i \in \mathbb{R}^n, i \in [n]$ denote the $i$-th row of $(\mathbf{X}^\top \mathbf{X})^{-1}$. Then, for the $i$-th element $\hat{\beta}_{c,i}$ of $\hat{\boldsymbol{\beta}}_c$, it holds that $\hat{\beta}_{c,i} = \mathbf{g}_i^\top \mathbf{z}_c$. Thus, for all $i \in [n]$, we have

$$\hat{\beta}_{y_i,i} + \sum_{c \neq y_i} \hat{\beta}_{c,i} = \mathbf{g}_i^\top \left( \mathbf{z}_{y_i} + \sum_{c \neq y_i} \mathbf{z}_c \right) = \mathbf{g}_i^\top \left( \sum_{c \in [k]} \mathbf{z}_c \right) = 0,$$

where in the last equality we used the definition of $\mathbf{z}_c$ in (7) and the fact that $\sum_{c \in [k]} \mathbf{v}_c = \mathbf{1}_n$, since each column of the label matrix $\mathbf{Y}$ has exactly one non-zero element equal to 1. Second, since Equation (8) holds, $\hat{\boldsymbol{\beta}}_c, c \in [k]$ further satisfies the $n$ *strict* inequality constraints in (22).

We have shown that the unconstrained maximizer is feasible in the constrained program (22). Thus, we can conclude that it is also the solution to the latter. By Equation (23), we note that the original dual variables $\{\lambda_{c,i}\}$ are all strictly positive. This completes the proof of the first part of the theorem, i.e. the proof of Equation (9).

**Proof of Equation** (10). To prove Equation (10), consider the following OvA-type classifier: for all $c \in [k]$,

$$\min_{\mathbf{w}_c} \frac{1}{2} \|\mathbf{w}_c\|_2^2 \qquad \text{sub. to} \quad \mathbf{x}_i^\top \mathbf{w}_c \begin{cases} \geq \frac{k-1}{k}, & y_i = c, \\ \leq -\frac{1}{k}, & y_i \neq c, \end{cases} \quad \forall i \in [n]. \tag{24}$$

To see the connection with Equation (10), note the condition for the constraints in (24) to be active is exactly Equation (10). Thus, it suffices to prove that the constraints of (24) are active under the theorem's assumptions. We work again with the dual of (24):

$$\max_{\boldsymbol{\nu}_c \in \mathbb{R}^k} \quad -\frac{1}{2} \|\mathbf{X}\boldsymbol{\nu}_c\|_2^2 + \mathbf{z}_c^\top \boldsymbol{\nu}_c \qquad \text{sub. to} \quad \mathbf{z}_c \odot \boldsymbol{\nu}_c \geq \mathbf{0}. \tag{25}$$

Again by complementary slackness, the desired Equation (10) holds provided that all dual constraints in (25) are strict at the optimal.

We now observe two critical similarities between (25) and (22): (i) the two dual problems have the same objectives (indeed the objective in (22) is separable over $c \in [k]$); (ii) they share the constraint $\mathbf{z}_c \odot \boldsymbol{\nu}_c \geq \mathbf{0}$ / $\mathbf{z}_c \odot \boldsymbol{\beta}_c \geq \mathbf{0}$. From this observation, we can use the same argument as for (22) to show that when Eqn. (8) holds, $\hat{\boldsymbol{\beta}}_c$ is optimal in (25).

Now, let $\text{OPT}_{(17)}$ and $\text{OPT}_{(24)}^c$ be the optimal costs of the multiclass SVM in (17) and of the symmetric OvA-SVM in (24) parameterized by $c \in [k]$. Also, denote $\text{OPT}_{(22)}$ and $\text{OPT}_{(25)}^c, c \in [k]$ the optimal costs of their respective duals in (22) and (25), respectively. We proved above that

$$\text{OPT}_{(22)} = \sum_{c \in [k]} \text{OPT}_{(25)}^c. \tag{26}$$

Further let $\mathbf{W}_{\text{sym-OvA}} = [\mathbf{w}_{\text{sym-OvA},1}, \ldots, \mathbf{w}_{\text{sym-OvA},k}]$ be the optimal solution in the symmetric OvA-SVM in (25). We have proved that under Equation (8) $\mathbf{w}_c^\star$ satisfies the constraints in (25) with equality, that is $\mathbf{X}^\top \mathbf{w}_{\text{sym-OvA},c} = \mathbf{z}_c, \forall c \in [k]$. Thus, it suffices to prove that $\mathbf{W}_{\text{sym-OvA}} = \mathbf{W}_{\text{SVM}}$. By strong duality, we get

$$\text{OPT}_{(24)}^c = \text{OPT}_{(25)}^c, \ c \in [k] \implies \sum_{c \in [k]} \text{OPT}_{(24)}^c = \sum_{c \in [k]} \text{OPT}_{(25)}^c$$

$$\overset{(26)}{\implies} \sum_{c \in [k]} \text{OPT}_{(24)}^c = \text{OPT}_{(22)}$$

$$\overset{(24)}{\implies} \sum_{c \in [k]} \frac{1}{2} \|\mathbf{w}_{\text{sym-OvA},c}\|_2^2 = \text{OPT}_{(22)}. \tag{27}$$

Again, by strong duality we get $\text{OPT}_{(22)} = \text{OPT}_{(17)}$. Thus, we have

$$\sum_{c \in [k]} \frac{1}{2} \|\mathbf{w}_{\text{sym-OvA},c}\|_2^2 = \text{OPT}_{(17)}.$$

Note also that $\mathbf{W}_{\mathrm{OvA}}$ is feasible in (17) since

$$\mathbf{X}^\top \mathbf{w}_{\mathrm{sym\text{-}OvA},c} = \mathbf{z}_c, \ \forall c \in [k] \implies (\mathbf{w}_{\mathrm{sym\text{-}OvA},y_i} - \mathbf{w}_{\mathrm{sym\text{-}OvA},c})^\top \mathbf{x}_i = 1, \ \forall c \neq y_i, c \in [k], \text{ and } \forall i \in [n].$$

Therefore, $\mathbf{W}_{\mathrm{sym\text{-}OvA}}$ is optimal in (17). Finally, note that the optimization objective in (17) is strongly convex. Thus, it has a unique minimum and therefore $\mathbf{W}_{\mathrm{SVM}} = \mathbf{W}_{\mathrm{sym\text{-}OvA}}$ as desired.

### B.2 Detailed proof of Corollary 1

The corollary follows directly by combining Theorem 1 with the following lemma.

**Lemma 1.** *Fix arbitrary constants $\alpha > 0, \beta$ and consider the MNI-solution $\mathbf{w}_c^{\alpha,\beta} = \mathbf{X}(\mathbf{X}^\top \mathbf{X})^{-1}(\alpha \mathbf{v}_c + \beta \mathbf{1}), c \in [k]$ corresponding to a target vector of labels $\alpha \mathbf{v}_c + \beta \mathbf{1}_n$. Let $\mathbb{P}_{e|c}^{\alpha,\beta}, c \in [k]$ be the class-conditional and total classification errors of the classifier $\mathbf{w}^{\alpha,\beta}$. Then, for any constants $\alpha' > 0, \beta'$, it holds that $\mathbb{P}_{e|c}^{\alpha,\beta} = \mathbb{P}_{e|c}^{\alpha',\beta'}, \forall c \in [k]$.*

*Proof.* Note that $\mathbf{w}_c^{\alpha=1,\beta=0} = \mathbf{w}_{\mathrm{MNI},c}, c \in [k]$ and for arbitrary $\alpha > 0, \beta$, we have: $\mathbf{w}_c^{\alpha,\beta} = \alpha \mathbf{w}_{\mathrm{MNI},c} + \beta \mathbf{X}(\mathbf{X}^\top \mathbf{X})^{-1}\mathbf{1}$. Moreover, it is not hard to check that $\mathbf{w}_{\mathrm{MNI},c}^\top \mathbf{x} \leq \max_{j \neq c} \mathbf{w}_{\mathrm{MNI},j}^\top \mathbf{x}$ if and only if $(\alpha \mathbf{w}_{\mathrm{MNI}c} + \mathbf{b})^\top \mathbf{x} \leq \max_{j \neq c}(\alpha \mathbf{w}_{\mathrm{MNI},j} + \mathbf{b})^\top \mathbf{x}$, for any $\mathbf{b} \in \mathbb{R}^p$. The claim then follows by choosing $\mathbf{b} = \beta \mathbf{X}(\mathbf{X}^\top \mathbf{X})^{-1}\mathbf{1}$ and noting that $\alpha > 0, \beta$ were chosen arbitrarily. $\square$

## C  Proof of Theorem 2

In this section, we provide the proof of Theorem 2, which was stated and discussed in Section 3.2.1.

### C.1  Notation and proof strategy

We begin by defining notation that is specific to this proof. For $c \in [k]$, we define

$$\mathbf{A}_c := (\mathbf{Q} + \sum_{j=1}^{c} \boldsymbol{\mu}_j \mathbf{v}_j^T)^T (\mathbf{Q} + \sum_{j=1}^{c} \boldsymbol{\mu}_j \mathbf{v}_j^T).$$

Recall that in the above, $\boldsymbol{\mu}_j$ denotes the $j^{th}$ class mean of dimension $p$, and $\mathbf{v}_j$ denotes the $n$-dimensional indicator that each training example is labeled as class $j$. Since we have made an equal-energy assumption on the class means (Assumption 2), we will denote $\|\boldsymbol{\mu}\|_2 := \|\boldsymbol{\mu}_c\|_2$ throughout this proof as shorthand. Further, recall from Eqn. (1) that the feature matrix can be expressed as $\mathbf{X} = \mathbf{M}\mathbf{Y} + \mathbf{Q}$, where $\mathbf{Q} \in \mathbb{R}^{p \times n}$ is a standard Gaussian matrix. Thus, we have

$$\mathbf{X}^T \mathbf{X} = \mathbf{A}_k \qquad \text{and} \qquad \mathbf{Q}^T \mathbf{Q} = \mathbf{A}_0.$$

Further, let $\mathbf{d}_c := \mathbf{Q}^T \boldsymbol{\mu}_c$, for $c \in [k]$. Finally, we define the following quadratic forms involving $\mathbf{A}_c^{-1}$ for $c, j, m \in [k]$ and $i \in [n]$:

$$
\begin{aligned}
s_{mj}^{(c)} &:= \mathbf{v}_m^T \mathbf{A}_c^{-1} \mathbf{v}_j, \\
t_{mj}^{(c)} &:= \mathbf{d}_m^T \mathbf{A}_c^{-1} \mathbf{d}_j, \\
h_{mj}^{(c)} &:= \mathbf{v}_m^T \mathbf{A}_c^{-1} \mathbf{d}_j, \\
g_{ji}^{(c)} &:= \mathbf{v}_j^T \mathbf{A}_c^{-1} \mathbf{e}_i, \\
f_{ji}^{(c)} &:= \mathbf{d}_j^T \mathbf{A}_c^{-1} \mathbf{e}_i.
\end{aligned}
\tag{28}
$$

For convenience, we refer to terms above as *quadratic forms of order $c$* or *the $c$-th order quadratic forms*, where $c$ indicates the corresponding superscript. Because of the sum of multiple mean components $\sum_{j=1}^{c} \boldsymbol{\mu}_j \mathbf{v}_j^T$, it is challenging to bound the quadratic forms involving the Gram matrix $\mathbf{A}_k^{-1}$ directly. Our idea is to work recursively starting from bounding quadratic forms involving $\mathbf{A}_0^{-1}$. Specifically, we denote $\mathbf{P}_1 = \mathbf{Q} + \boldsymbol{\mu}_1 \mathbf{v}_1^T$ and derive the following recursion on the $\mathbf{A}_0, \mathbf{A}_1, \ldots, \mathbf{A}_k$

matrices:

$$\mathbf{A}_1 = \mathbf{P}_1^T \mathbf{P}_1 = \mathbf{A}_0 + \begin{bmatrix} \|\boldsymbol{\mu}\|_2 \mathbf{v}_1 & \mathbf{Q}^T \boldsymbol{\mu}_1 & \mathbf{v}_1 \end{bmatrix} \begin{bmatrix} \|\boldsymbol{\mu}\|_2 \mathbf{v}_1^T \\ \mathbf{v}_1^T \\ \boldsymbol{\mu}_1^T \mathbf{Q} \end{bmatrix},$$

$$\mathbf{A}_2 = (\mathbf{P}_1 + \boldsymbol{\mu}_2 \mathbf{v}_2^T)^T (\mathbf{P}_1 + \boldsymbol{\mu}_2 \mathbf{v}_2^T) = \mathbf{A}_1 + \begin{bmatrix} \|\boldsymbol{\mu}\|_2 \mathbf{v}_2 & \mathbf{P}_1^T \boldsymbol{\mu}_2 & \mathbf{v}_2 \end{bmatrix} \begin{bmatrix} \|\boldsymbol{\mu}\|_2 \mathbf{v}_2^T \\ \mathbf{v}_2^T \\ \boldsymbol{\mu}_2^T \mathbf{P}_1 \end{bmatrix}, \tag{29}$$

and so on, until $\mathbf{A}_k$ (see Appendix F.1 for the complete expressions for the recursion). Using this trick, we can exploit bounds on forms involving $\mathbf{A}_0^{-1}$ to obtain bounds for quadratic forms involving $\mathbf{A}_1^{-1}$, and so on until $\mathbf{A}_k^{-1}$. A complementary useful observation facilitating this approach is that the class label indicators are orthogonal by definition, i.e. $\mathbf{v}_i^T \mathbf{v}_j = 0$, for $i, j \in [k]$. (This is a consequence of the obvious fact that any training data point has a unique label.) Thus, the newly added mean component $\boldsymbol{\mu}_{c+1} \mathbf{v}_{c+1}^T$ is orthogonal to the already existing mean components included in the matrix $\mathbf{A}_c$. Consequently, we will see that adding new mean components will only slightly change the magnitude of these these quadratic forms as $c$ ranges from 0 to $k$.

Following the above prescribed approach leads to a key technical Lemma 2, which is presented in Appendix C.2. In Appendix C.2, we also show how to prove Theorem 2 using that lemma. A series of auxiliary lemmas used to prove Lemma 2 are stated in Appendix C.3. The proofs of all the lemmas are given in the remaining subsections.

## C.2 Proof of Theorem 2

In our new notation, the desired inequality in Eqn. (8) becomes

$$z_{ci} \mathbf{e}_i^T \mathbf{A}_k^{-1} \mathbf{z}_c > 0, \quad \text{for all} \ \ c \in [k] \ \text{and} \ i \in [n].$$

We can equivalently write the definition of $\mathbf{z}_c$ in Eqn. (7) as

$$\mathbf{z}_c = \frac{k-1}{k} \mathbf{v}_c + \sum_{j \neq c} \left( -\frac{1}{k} \right) \mathbf{v}_j = \tilde{z}_{c(c)} \mathbf{v}_c + \sum_{j \neq c} \tilde{z}_{j(c)} \mathbf{v}_j, \tag{30}$$

where we denote

$$\tilde{z}_{j(c)} = \begin{cases} -\frac{1}{k}, & \text{if} \ j \neq c \\ \frac{k-1}{k}, & \text{if} \ j = c \end{cases}.$$

Note that by this definition, we have $\tilde{z}_{y_i(c)} := z_{ci}$. This gives us

$$z_{ci} \mathbf{e}_i^T \mathbf{A}_k^{-1} \mathbf{z}_c = z_{ci}^2 \mathbf{e}_i^T \mathbf{A}_k^{-1} \mathbf{v}_{y_i} + \sum_{j \neq y_i} z_{ci} \tilde{z}_{j(c)} \mathbf{e}_i^T \mathbf{A}_k^{-1} \mathbf{v}_j,$$

$$= z_{ci}^2 g_{y_i i}^{(k)} + \sum_{j \neq y_i} z_{ci} \tilde{z}_{j(c)} g_{ji}^{(k)}. \tag{31}$$

Recall that

$$z_{ci} = \begin{cases} \frac{k-1}{k}, & \text{if} \ c = y_i \\ -\frac{1}{k}, & \text{if} \ c \neq y_i \end{cases}.$$

For each $\mathbf{e}_i^T \mathbf{A}_k^{-1} \mathbf{v}_j$, we then use the matrix inversion lemma to leave the $j$-th mean component in $\mathbf{A}_k$ out. Using this, we can express the terms $\{g_{ji}^{(k)}\}$ in terms of the *leave-one-out* versions of quadratic forms that we defined in (28), as below:

$$g_{ji}^{(k)} = \mathbf{e}_i^T \mathbf{A}_k^{-1} \mathbf{v}_j = \frac{(1 + h_{jj}^{(-j)}) g_{ji}^{(-j)} - s_{jj}^{(-j)} f_{ji}^{(-j)}}{s_{jj}^{(-j)}(\|\boldsymbol{\mu}\|_2^2 - t_{jj}^{(-j)}) + (1 + h_{jj}^{(-j)})^2}. \tag{32}$$

Above, we define $s_{jj}^{(-j)} := \mathbf{v}_j^T \mathbf{A}_{-j}^{-1} \mathbf{v}_j$, where $\mathbf{A}_{-j}$ denotes the version of the Gram matrix $\mathbf{A}_k$ with the $j$-th mean component left out. The quadratic forms $h_{jj}^{(-j)}$, $f_{ji}^{(-j)}$, $g_{ji}^{(-j)}$ and $t_{jj}^{(-j)}$ are defined similarly. The following technical lemma bounds all of these quantities.

**Lemma 2** (Quadratic forms of high orders). *Let Assumption 1 hold and further assume that $p > Ck^3 n \log(kn) + n - 1$ for large enough constant $C > 1$ and large $n$. There exist constants $c_i$'s and $C_i$'s $> 1$ such that the following bounds hold for every $i \in [n]$ and $j \in [k]$ with probability at least $1 - \frac{c_1}{n} - c_2 k e^{-\frac{n}{c_3 k^2}}$,*

$$\frac{C_1 - 1}{C_1} \cdot \frac{n}{kp} \leq s_{jj}^{(-j)} \leq \frac{C_1 + 1}{C_1} \cdot \frac{n}{kp},$$

$$t_{jj}^{(-j)} \leq \frac{C_2 n \|\boldsymbol{\mu}\|_2^2}{p},$$

$$-\frac{C_3 n \|\boldsymbol{\mu}\|_2}{\sqrt{kp}} \leq h_{jj}^{(-j)} \leq \frac{C_3 n \|\boldsymbol{\mu}\|_2}{\sqrt{kp}},$$

$$|f_{ji}^{(-j)}| \leq \frac{C_4 \sqrt{n} \|\boldsymbol{\mu}\|_2}{p},$$

$$g_{ji}^{(-j)} \geq \left(1 - \frac{1}{C_5}\right) \frac{1}{p}, \quad \text{for } j = y_i,$$

$$|g_{ji}^{(-j)}| \leq \frac{1}{C_6 k^2 p}, \quad \text{for } j \neq y_i.$$

Observe that the bounds stated in the lemma hold for any $j \in [k]$ and the bounds themselves are independent of $j$. We now show how to use Lemma 2 to complete the proof of the theorem. Following the second condition in the statement of Theorem 2, we define

$$\epsilon := \frac{k^{1.5} n^{1.5} \|\boldsymbol{\mu}\|_2}{p} \leq \tau, \tag{33}$$

where $\tau$ is a sufficiently small positive constant, the value of which will be specified later in the proof. First, we will show that the denominator of Eqn. (32) is strictly positive on the event where Lemma 2 holds. We define

$$\det_{-j} := s_{jj}^{(-j)}(\|\boldsymbol{\mu}\|_2^2 - t_{jj}^{(-j)}) + (1 + h_{jj}^{(-j)})^2.$$

By Lemma 2, the quadratic forms $s_{jj}^{(-j)}$ are of the same order $\Theta\left(\frac{n}{kp}\right)$ for every $j \in [k]$. Similarly, we have $t_{jj}^{(-j)} = \mathcal{O}\left(\frac{n}{p} \|\boldsymbol{\mu}\|_2^2\right)$ and $|h_{jj}^{(-j)}| = \mathcal{O}\left(\frac{\epsilon}{k^2 \sqrt{n}}\right)$ for $j \in [k]$. Thus, we have

$$\frac{n \|\boldsymbol{\mu}\|_2^2}{C_1 kp}\left(1 - \frac{C_2 n}{p}\right) + \left(1 - \frac{C_3 \epsilon}{k^2 \sqrt{n}}\right)^2 \leq \det_{-j} \leq \frac{C_1 n \|\boldsymbol{\mu}\|_2^2}{kp} + \left(1 + \frac{C_3 \epsilon}{k^2 \sqrt{n}}\right)^2, \tag{34}$$

with probability at least $1 - \frac{c_1}{n} - c_2 k e^{-\frac{n}{c_3 k^2}}$, for every $j \in [k]$. Here, we use the fact that $t_{jj}^{-j} \geq 0$ by the positive semidefinite property of the leave-one-out Gram matrix $\mathbf{A}_{-j}^{-1}$. Next, we choose $\tau$ in Eqn. (33) to be sufficiently small so that $C_3 \tau \leq 1/2$. Provided that $p$ is sufficiently large compared to $n$, there then exist constants $C_1', C_2' > 0$ such that we have

$$C_1' \leq \frac{\det_{-m}}{\det_{-j}} \leq C_2', \quad \text{for all } j, m \in [k].$$

with probability at least $1 - \frac{c_1}{n} - c_2 k e^{-\frac{n}{c_3 k^2}}$. Now, assume without loss of generality that $y_i = k$. Eqn. (34) shows that there exists constant $c > 0$ such that $\det_{-j} > c$ for all $j \in [k]$ with high probability provided that $p/n$ is large enough (guaranteed by the first condition of the theorem). Hence, to make the right-hand-side of Eqn. (31) positive, it suffices to show that the numerator will be positive. Accordingly, we will show that

$$z_{ci}^2\left((1 + h_{kk}^{(-k)})g_{ki}^{(-k)} - s_{kk}^{(-k)} f_{ki}^{(-k)}\right) + Cz_{ci} \sum_{j \neq k} \tilde{z}_j\left((1 + h_{jj}^{(-j)})g_{ji}^{(-j)} - s_{jj}^{(-j)} f_{ji}^{(-j)}\right) > 0, \tag{35}$$

for some $C > 1$.

We can show by simple algebra that it suffices to consider the worst case of $z_{ci} = -1/k$. To see why this is true, we consider the simpler term $z_{ci}^2 g_{y_i i}^{(-y_i)} - |\sum_{j \neq y_i} z_{ci} \tilde{z}_{j(c)} g_{ji}^{(-j)}|$. Clearly, Eqn. (35) is

positive only if the above quantity is also positive. Lemma 2 shows that when $z_{ci} = -1/k$, then $z_{ci}^2 g_{y_i i}^{(-y_i)} \geq \left(1 - \frac{1}{C_1}\right) \frac{1}{k^2 p}$ and $|z_{ci} \tilde{z}_{j(c)} g_{ji}^{(-j)}| \leq \frac{1}{C_2 k^3 p}$, for $j \neq y_i$. Hence

$$z_{ci}^2 g_{y_i i}^{(-y_i)} - |\sum_{j \neq y_i} z_{ci} \tilde{z}_{j(c)} g_{ji}^{(-j)}| \geq \left(1 - \frac{1}{C_3}\right) \frac{1}{k^2 p}.$$

Here, $z_{ci} = -1/k$ minimizes the lower bound $z_{ci}^2 g_{y_i i}^{(-y_i)} - |\sum_{j \neq y_i} z_{ci} \tilde{z}_{j(c)} g_{ji}^{(-j)}|$. To see this, we first drop the positive common factor $|z_{ci}|$ in the equation above and get $|z_{ci} g_{y_i i}^{(-y_i)} - |\sum_{j \neq y_i} \tilde{z}_{j(c)} g_{ji}^{(-j)}||$. If we had $z_{ci} = -1/k$, then $|\tilde{z}_{j(c)}|$ is either $(k-1)/k$ or $1/k$. In contrast, if we consider $z_{ci} = (k-1)/k$, then we have $|\tilde{z}_{j(c)}| = 1/k$ for all $j \neq y_i$ and so the term $|z_{ci} g_{y_i i}^{(-y_i)} - |\sum_{j \neq y_i} \tilde{z}_{j(c)} g_{ji}^{(-j)}||$ is strictly larger.

Using this worst case, i.e. $z_{ci} = -1/k$, and the trivial inequality $|\tilde{z}_{j(c)}| < 1$ for $j \neq y_i$ together with the bounds for the terms $s_{jj}^{(-j)}, t_{jj}^{(-j)}, h_{jj}^{(-j)}$ and $f_{ji}^{(-j)}$ derived in Lemma 2 gives us

$$(35) \geq \frac{1}{k^2} \left( \left(1 - \frac{C_1 \epsilon}{k^2 \sqrt{n}}\right) \left(1 - \frac{1}{C_2}\right) \frac{1}{p} - \frac{C_3 \epsilon}{k^{1.5} n} \cdot \frac{n}{kp} \right) - \frac{k}{C_4 k} \left( \left(1 + \frac{C_5 \epsilon}{k^2 \sqrt{n}}\right) \frac{1}{k^2 p} - \frac{C_6 \epsilon}{k^{1.5} n} \frac{n}{kp} \right)$$

$$\geq \frac{1}{k^2} \left(1 - \frac{1}{C_9} - \frac{C_{10} \epsilon}{k^2 \sqrt{n}} - \frac{C_{11} \epsilon}{k^2} - C_{12} \epsilon \right) \frac{1}{p}$$

$$\geq \frac{1}{k^2 p} \left(1 - \frac{1}{C_9} - C_{10} \tau \right), \tag{36}$$

with probability at least $1 - \frac{c_1}{n} - c_2 k e^{-\frac{n}{c_3 k^2}}$ for some constants $C_i$'s $> 1$. Above, we recalled the definition of $\epsilon$ and used from Lemma 2 that $h_{jj}^{(-j)} \leq \frac{C_{11} \epsilon}{k^2 \sqrt{n}}$ and $|f_{ji}^{(-j)}| \leq \frac{C_{12} \epsilon}{k^{1.5} n}$ with high probability. To complete the proof, we choose $\tau$ to be a small enough constant to guarantee $C_{10} \tau < 1 - 1/C_9$, and substitute this in Eqn. (36) to get the desired condition of Eqn. (35).

## C.3 Auxiliary Lemmas

In this section, we state a series of auxiliary lemmas that we use to prove Lemma 2. The following result shows concentration of the norms of the label indicators $\mathbf{v}_c, c \in [k]$ under the equal-priors assumption (Assumption 1). Intuitively, in this balanced setting there are $\Theta(n/k)$ samples for each class; hence, $\Theta(n/k)$ non-zeros (in fact, 1's) in each label indicator vector $\mathbf{v}_c$.

**Lemma 3.** *Under the setting of Assumption 1, there exist large constants $C_1, C_2 > 0$ such that the event*

$$\mathcal{E}_v := \left\{ \left(1 - \frac{1}{C_1}\right) \frac{n}{k} \leq \|\mathbf{v}_c\|_2^2 \leq \left(1 + \frac{1}{C_1}\right) \frac{n}{k}, \ \forall c \in [k] \right\}, \tag{37}$$

*holds with probability at least $1 - 2k e^{-\frac{n}{C_2 k^2}}$.*

Next, we provide bounds on the "base case" 0-th order quadratic forms that involve the Gram matrix $\mathbf{A}_0^{-1}$. We do this in three lemmas presented below. The first Lemma 4 follows by a direct application of [WT21, Lemma 4 and 5], the only difference being the scalings of $\mathcal{O}(1/k)$ arising from the multiclass case in the $\mathbf{v}_j$'s. The other two Lemmas 5 and 6 are proved in Section C.5.

**Lemma 4** (0-th order Quadratic forms, Part I). *Under the event $\mathcal{E}_v$, there exist constants $c_i$'s and $C_i$'s $> 1$ such that the following bounds hold with probability at least $1 - c_1 k e^{-\frac{n}{c_2}}$.*

$$t_{jj}^{(0)} \leq \frac{C_1 n \|\boldsymbol{\mu}\|_2^2}{p} \text{ for all } j \in [k],$$

$$|h_{mj}^{(0)}| \leq \frac{C_2 n \|\boldsymbol{\mu}\|_2}{\sqrt{k}p} \text{ for all } m, j \in [k],$$

$$|t_{mj}^{(0)}| \leq \frac{C_3 n \|\boldsymbol{\mu}\|_2^2}{p} \text{ for all } m \neq j \in [k],$$

$$\|\mathbf{d}_j\|_2^2 \leq C_4 n \|\boldsymbol{\mu}\|_2^2 \text{ for all } j \in [k],$$

$$\max_{i \in [n]} |f_{ji}^{(0)}| \leq \frac{C_5 \sqrt{\log(2n)} \|\boldsymbol{\mu}\|_2}{p} \text{ for all } j \in [k].$$

To sharply characterize the forms $s_{ij}^{(0)}$ we need additional work, particularly for the cross-terms where $i \neq j$. We will make use of fundamental concentration inequalities on quadratic forms of inverse Wishart matrices. The following lemma controls these quadratic forms, and shows in particular that the $s_{ij}^{(0)}$ terms for $i \neq j$ are much smaller than the corresponding terms $s_{jj}^{(0)}$. This sharp control of the cross-terms is essential for several subsequent proof steps.

**Lemma 5** (0-th order Quadratic forms, Part II). *Working on the event $\mathcal{E}_v$ defined in Eqn. (37), assume that $p > Cn\log(kn) + n - 1$ for large enough constant $C > 1$ and large $n$. There exist constants $C_i$'s $> 1$ such that with probability at least $1 - \frac{C_0}{n}$, the following bound holds:*

$$\frac{C_1 - 1}{C_1} \cdot \frac{n}{kp} \leq s_{jj}^{(0)} \leq \frac{C_1 + 1}{C_1} \cdot \frac{n}{kp}, \text{ for } j \in [k],$$

$$-\frac{C_2 + 1}{C_2} \cdot \frac{\sqrt{n}}{kp} \leq s_{ij}^{(0)} \leq \frac{C_2 + 1}{C_2} \cdot \frac{\sqrt{n}}{kp}, \text{ for } i \neq j \in [k].$$

The proof of Lemma 5 for the cross terms with $i \neq j$ critically uses the in-built orthogonality of the label indicator vectors $\{\mathbf{v}_c\}_{c \in [k]}$. Finally, the following lemma controls the quadratic forms $g_{ji}^{(0)}$.

**Lemma 6** (0-th order Quadratic forms, Part III). *Working on the event $\mathcal{E}_v$ defined in Eqn. (37), given $p > Ck^3 n\log(kn) + n - 1$ for a large constant $C$, there exist large enough constants $C_1, C_2$, such that with probability at least $1 - \frac{2}{kn}$, we have for every $i \in [n]$:*

$$\left(1 - \frac{1}{C_1}\right) \frac{1}{p} \leq g_{(y_i)i}^{(0)} \leq \left(1 + \frac{1}{C_1}\right) \frac{1}{p},$$

$$-\frac{1}{C_2} \cdot \frac{1}{k^2 p} \leq g_{ji}^{(0)} \leq \frac{1}{C_2} \cdot \frac{1}{k^2 p}, \text{ for } j \neq y_i.$$

### C.4   Proof of Lemma 2

In this section, we provide the full proof of Lemma 2. We begin with a proof outline.

#### C.4.1   Proof outline

It suffices to consider the case where $j = k$. To see this, note that when $j \neq k$ we can simply change the order of adding mean components, described in Eqn. (29), so that the $j$-th mean component is added last. For concreteness, we will also fix $i \in [n]$, $y_i = k$ and $m = k - 1$. These fixes are without loss of generality.

For the case $j = k$, the leave-one-out quadratic forms in Lemma 2 are equal to the quadratic forms of order $k - 1$, given by $s_{kk}^{(k-1)}, t_{kk}^{(k-1)}, h_{kk}^{(k-1)}, g_{ki}^{(k-1)}$ and $f_{ki}^{(k-1)}$. We will proceed recursively starting from the quadratic forms of order 1 building up all the way to the quadratic forms of order $k - 1$. Specifically, starting from order 1, we will work on the event

$$\mathcal{E}_q := \{\text{all the inequalities in Lemmas 4, 5 and 6 hold}\}, \tag{38}$$

Further, we note that Lemma 6 shows that the bound for $g_{y_i i}^{(0)}$ is different from the bound for $g_{ji}^{(0)}$ when $j \neq y_i$. We will show the following set of upper and lower bounds:

$$\left(\frac{C_{11}-1}{C_{11}}\right)\frac{n}{kp} \leq s_{kk}^{(1)} \leq \left(\frac{C_{11}+1}{C_{11}}\right)\frac{n}{kp},$$

$$-\left(\frac{C_{12}+1}{C_{12}}\right)\frac{\sqrt{n}}{kp} \leq s_{mk}^{(1)} \leq \left(\frac{C_{12}+1}{C_{12}}\right)\frac{\sqrt{n}}{kp},$$

$$t_{kk}^{(1)} \leq \frac{C_{13}n\|\boldsymbol{\mu}\|_2^2}{p},$$

$$|h_{mk}^{(1)}| \leq \frac{C_{14}n\|\boldsymbol{\mu}\|_2}{\sqrt{k}p},$$

$$|t_{mk}^{(1)}| \leq \frac{C_{15}n\|\boldsymbol{\mu}\|_2^2}{p}, \tag{39}$$

$$\|\mathbf{d}_k\|_2^2 \leq C_{16}n\|\boldsymbol{\mu}\|_2^2,$$

$$|f_{ki}^{(1)}| \leq \frac{C_{17}\sqrt{n}\|\boldsymbol{\mu}\|_2}{p},$$

$$\left(1-\frac{1}{C_{18}}\right)\frac{1}{p} \leq g_{(y_i)i}^{(1)} \leq \left(1+\frac{1}{C_{18}}\right)\frac{1}{p}, \text{ and}$$

$$-\frac{1}{C_{19}k^2 p} \leq g_{mi}^{(1)} \leq \frac{1}{C_{19}k^2 p}$$

with probability at least $1 - \frac{c}{kn^2}$. Comparing the bounds on the terms of order 1 in Eqn. (39) with the terms in Lemmas 4, 5 and 6 of order 0, the key observation is that they are all at the same order. The only exception is the term $f_{ki}^{(1)}$, which has a higher upper bound than $f_{ki}^{(0)}$. For this case, the subsequent $c$-th order terms $f_{ki}^{(c)}$ will be of the same order as $f_{ki}^{(1)}$. This allows us to repeat the same argument to now bound corresponding terms of order 2, and so on until order $k-1$. Note that for each $j \in [k]$, we have $n$ terms of the form $g_{ji}^{(1)}$, corresponding to each value of $i \in [n]$. Thus, we will adjust the final probabilities by applying a union bound over the $n$ training examples.

### C.4.2 Proofs for 1-st order quadratic forms in Eqn. (39)

The proof makes repeated use of Lemmas 4, 5 and 6. In fact, we will throughout condition on the event $\mathcal{E}_q$, defined in Equation (38), which holds with probability at least $1 - \frac{c_1}{n} - c_2 e^{-\frac{n}{c_0 k^2}}$. Specifically, by Lemma 4 we have

$$h_{mj}^{(0)} \leq \frac{C_1 \epsilon}{k^2 \sqrt{n}}, \qquad \max_{i \in [n]}|f_{mi}^{(0)}| \leq \frac{C_2 \epsilon}{k^{1.5}n}, \quad \text{and} \quad \frac{s_{mj}^{(0)}}{s_{kk}^{(0)}} \leq \frac{C}{\sqrt{n}} \text{ for } m, j \neq k, \tag{40}$$

where we recall from Eqn. (33) the notation $\epsilon := \frac{k^{1.5}n^{1.5}\|\boldsymbol{\mu}\|_2}{p}$. Also, recall that we choose $\epsilon \leq \tau$ for a sufficiently small constant $\tau$. In Eqn. (40), note that we used a loose upper bound $C\sqrt{n}\|\boldsymbol{\mu}\|_2/p$ for $\max_{i \in [n]}|f_{mi}^{(0)}|$ compared to the bound given in Lemma 4. This looser bound will suffice for the proof of Equation (33). Moreover, it actually coincides with the bound on $\max_{i \in [n]}|f_{mi}^{(1)}|$ given in Equation (33). This remark is important as it will allow us to use Equation (33) to bound 2-nd order quadratic forms in the same way as shown below.

In order to make use of Lemmas 4, 5 and 6, we need to relate the quantities of interest to corresponding quadratic forms involving $\mathbf{A}_0$. We do this recursively and make repeated use of the Woodbury identity. The recursions are proved in Appendix F.1. We now provide the proofs for the bounds on the terms in Eqn. (39) one-by-one.

**Bounds on $s_{mk}^{(1)}$.** By Eqn. (71) in Appendix F.1, we have

$$s_{mk}^{(1)} = s_{mk}^{(0)} - \frac{1}{\det_0}(\star)_s^{(0)}, \tag{41}$$

where we define

$$(\star)_s^{(0)} := (\|\boldsymbol{\mu}\|_2^2 - t_{11}^{(0)})s_{1k}^{(0)}s_{1m}^{(0)} + s_{1m}^{(0)}h_{k1}^{(0)}h_{11}^{(0)} + s_{1k}^{(0)}h_{m1}^{(0)}h_{11}^{(0)} - s_{11}^{(0)}h_{k1}^{(0)}h_{m1}^{(0)} + s_{1m}^{(0)}h_{k1}^{(0)} + s_{1k}^{(0)}h_{m1}^{(0)}$$

and $\det_0 := s_{11}^{(0)}(\|\boldsymbol{\mu}\|_2^2 - t_{11}^{(0)}) + (1 + h_{11}^{(0)})^2$. $\qquad(42)$

The essential idea is to show that $\left|\frac{(\star)_s^{(0)}}{\det_0}\right|$ is sufficiently small compared to $|s_{mk}^{(0)}|$. We first look at the first term given by $\left((\|\boldsymbol{\mu}\|_2^2 - t_{11}^{(0)})s_{1k}^{(0)}s_{1m}^{(0)}\right)/\det_0$. By Lemmas 4, 5 and the definition of $\det_0$, we have

$$\left|\frac{1}{\det_0}\left((\|\boldsymbol{\mu}\|_2^2 - t_{11}^{(0)})s_{1k}^{(0)}s_{1m}^{(0)}\right)\right| \le \frac{(\|\boldsymbol{\mu}\|_2^2 - t_{11}^{(0)})|s_{1k}^{(0)}s_{1m}^{(0)}|}{s_{11}^{(0)}(\|\boldsymbol{\mu}\|_2^2 - t_{11}^{(0)})} = \left|\frac{s_{1k}^{(0)}s_{1m}^{(0)}}{s_{11}^{(0)}}\right| \le \frac{C_1}{\sqrt{n}} \cdot \frac{C_2 + 1}{C_2} \cdot \frac{\sqrt{n}}{kp},$$

where we use $\det_0 \ge s_{11}^{(0)}(\|\boldsymbol{\mu}\|_2^2 - t_{11}^{(0)})$ and $s_{mj}^{(0)}/s_{kk}^{(0)} \le C/\sqrt{n}$ for all $m, j \ne k$. Now, we upper bound the other two dominant terms $|s_{1m}^{(0)}h_{k1}^{(0)}/\det_0|$ and $|s_{1k}^{(0)}h_{m1}^{(0)}/\det_0|$. Note that the same bound will apply to the remaining terms in Eqn. (42) because we trivially have $|h_{ij}^{(0)}| = \mathcal{O}(1)$ for all $(i,j) \in [k]$. Again, Lemmas 4 and 5 give us

$$\left|\frac{s_{1m}^{(0)}h_{k1}^{(0)}}{\det_0}\right| \le \frac{|s_{1m}^{(0)}h_{k1}^{(0)}|}{(1 + h_{11}^{(0)})^2} \le \frac{C_3\epsilon}{\left(1 - \frac{C_5\epsilon}{k^2\sqrt{n}}\right)^2 k^2\sqrt{n}} \cdot \frac{C_2 + 1}{C_2} \cdot \frac{\sqrt{n}}{kp}.$$

The identical bound holds for $|s_{1k}^{(0)}h_{m1}^{(0)}|$. Noting that $|s_{mk}^{(0)}| \le \frac{C_2+1}{C_2} \cdot \frac{\sqrt{n}}{kp}$, we then have

$$|s_{mk}^{(1)}| \le |s_{mk}^{(0)}| + \left|\frac{(\star)_s^{(0)}}{\det_0}\right|$$

$$\le \left(1 + \frac{C_6}{\sqrt{n}} + \frac{C_7\epsilon}{\left(1 - \frac{C_5\epsilon}{k^2\sqrt{n}}\right)^2 k^2\sqrt{n}}\right)\frac{C_2 + 1}{C_2} \cdot \frac{\sqrt{n}}{kp}$$

$$\le (1 + \alpha) \cdot \frac{C_2 + 1}{C_2} \cdot \frac{\sqrt{n}}{kp}, \qquad(43)$$

where in the last inequality, we use that $\epsilon \le \tau$ for sufficiently small constant $\tau > 0$, and defined

$$\alpha := \frac{C_6}{\sqrt{n}} + \frac{C_7\tau}{\left(1 - \frac{C_5\tau}{k^2\sqrt{n}}\right)^2 k^2\sqrt{n}}.$$

Now, we pick $\tau$ to be sufficiently small and $n$ to be sufficiently large such that $(1 + \alpha)\frac{C_2+1}{C_2} \le \frac{C_8+1}{C_8}$ for some constant $C_8 > 0$. Then, we conclude with the following upper bound:

$$|s_{mk}^{(1)}| \le \frac{C_8 + 1}{C_8} \cdot \frac{\sqrt{n}}{kp}.$$

**Bounds on $s_{kk}^{(1)}$.** Eqn. (72) in Appendix F.1 gives us

$$s_{kk}^{(1)} = s_{kk}^{(0)} - \frac{1}{\det_0}\left((\|\boldsymbol{\mu}\|_2^2 - t_{11}^{(0)})s_{1k}^{(0)^2} + 2s_{1k}^{(0)}h_{k1}^{(0)}h_{11}^{(0)} - s_{11}^{(0)}h_{k1}^{(0)^2} + 2s_{1k}^{(0)}h_{k1}^{(0)}\right).$$

First, we lower bound $s_{kk}^{(1)}$ by upper bounding $\frac{1}{\det_0}\left((\|\boldsymbol{\mu}\|_2^2 - t_{11}^{(0)})s_{1k}^{(0)^2}\right)$. Lemmas 4 and 5 yield

$$\frac{1}{\det_0}\left((\|\boldsymbol{\mu}\|_2^2 - t_{11}^{(0)})s_{1k}^{(0)^2}\right) \le \frac{(\|\boldsymbol{\mu}\|_2^2 - t_{11}^{(0)})s_{1k}^{(0)^2}}{s_{11}^{(0)}(\|\boldsymbol{\mu}\|_2^2 - t_{11}^{(0)}) + (1 + h_{11}^{(0)})^2} \le \frac{(\|\boldsymbol{\mu}\|_2^2 - t_{11}^{(0)})s_{1k}^{(0)^2}}{s_{11}^{(0)}(\|\boldsymbol{\mu}\|_2^2 - t_{11}^{(0)})} \le \frac{C_1}{n} \cdot \frac{n}{kp}.$$

It suffices to upper bound the other dominant term $|s_{1k}^{(0)}h_{k1}^{(0)}|/\det_0$. For this term, we have

$$\left|\frac{s_{1k}^{(0)}h_{k1}^{(0)}}{\det_0}\right| \le \frac{|s_{1k}^{(0)}h_{k1}^{(0)}|}{(1 + h_{11}^{(0)})^2} \le \frac{C_3\epsilon}{\left(1 - \frac{C_4\epsilon}{k^2\sqrt{n}}\right)^2 k^2\sqrt{n}} \cdot \frac{C_2 + 1}{C_2} \cdot \frac{\sqrt{n}}{kp}.$$

Thus, we get

$$s_{kk}^{(1)} \geq \left(1 - \frac{C_1}{n} - \frac{C_5\epsilon}{\left(1 - \frac{C_4\epsilon}{k^2\sqrt{n}}\right)^2 k^2\sqrt{n}}\right) \frac{C_6 - 1}{C_6} \cdot \frac{n}{kp} \geq (1 - \alpha) \cdot \frac{C_6 - 1}{C_6} \cdot \frac{n}{kp}.$$

Next, we upper bound $s_{kk}^{(1)}$ by a similar argument, and get

$$s_{kk}^{(1)} \leq |s_{kk}^{(0)}| + \frac{1}{\det_0} \left|2s_{1k}^{(0)}h_{k1}^{(0)}h_{11}^{(0)} + s_{11}^{(0)}h_{k1}^{(0)^2} + 2s_{1k}^{(0)}h_{k1}^{(0)}\right|$$

$$\leq \left(1 + \frac{C_7\epsilon}{\left(1 - \frac{C_4\epsilon}{k^2\sqrt{n}}\right)^2 k^2\sqrt{n}}\right) \frac{C_8 + 1}{C_8} \cdot \frac{n}{kp} \leq (1 + \alpha')\frac{C_8 + 1}{C_8} \cdot \frac{n}{kp},$$

where we used $\frac{1}{\det_0}\left((\|\boldsymbol{\mu}\|_2^2 - t_{11}^{(0)})s_{1k}^{(0)^2}\right) > 0$ in the first step. As above, we can tune $\epsilon$ and $n$ such that $(1 + \alpha')\frac{C_8+1}{C_8} \leq \frac{C_9+1}{C_9}$ and $(1 - \alpha)\frac{C_6-1}{C_6} \geq \frac{C_9-1}{C_9}$ for sufficiently large constant $C_9 > 0$.

**Bounds on $h_{mk}^{(1)}$.** Eqn. (73) in Appendix F.1 gives us

$$h_{mk}^{(1)} = h_{mk}^{(0)} - \frac{1}{\det_0}(\star)_h^{(0)},$$

where we define

$$(\star)_h^{(0)} = (\|\boldsymbol{\mu}\|_2^2 - t_{11}^{(0)})s_{1m}^{(0)}h_{1k}^{(0)} + h_{m1}^{(0)}h_{1k}^{(0)}h_{11}^{(0)} + h_{m1}^{(0)}h_{1k}^{(0)} + s_{1m}^{(0)}t_{k1}^{(0)} + s_{1m}^{(0)}t_{k1}^{(0)}h_{11}^{(0)} - s_{11}^{(0)}t_{k1}^{(0)}h_{m1}^{(0)}.$$

We focus on the two dominant terms $((\|\boldsymbol{\mu}\|_2^2 - t_{11}^{(0)})s_{1m}^{(0)}h_{1k}^{(0)})/\det_0$ and $s_{1m}^{(0)}t_{k1}^{(0)}/\det_0$. For the first dominant term $((\|\boldsymbol{\mu}\|_2^2 - t_{11}^{(0)})s_{1m}^{(0)}h_{1k}^{(0)})/\det_0$, Lemmas 4 and 5 yield

$$\left|\frac{1}{\det_0}\left((\|\boldsymbol{\mu}\|_2^2 - t_{11}^{(0)})s_{1m}^{(0)}h_{1k}^{(0)}\right)\right| \leq \frac{(\|\boldsymbol{\mu}\|_2^2 - t_{11}^{(0)})|s_{1m}^{(0)}h_{1k}^{(0)}|}{s_{11}^{(0)}(\|\boldsymbol{\mu}\|_2^2 - t_{11}^{(0)})} \leq \left|\frac{s_{1m}^{(0)}h_{1k}^{(0)}}{s_{11}^{(0)}}\right| \leq \frac{C_1}{\sqrt{n}}|h_{1k}^{(0)}| \leq \frac{C_2\epsilon}{k^2 n}.$$

For the second dominant term $s_{1m}^{(0)}t_{k1}^{(0)}/\det_0$, we have

$$\frac{1}{\det_0}s_{1m}^{(0)}t_{k1}^{(0)} \leq \frac{|s_{1m}^{(0)}t_{k1}^{(0)}|}{(1 + h_{11}^{(0)})^2} \leq \frac{C_3 n\sqrt{n}\|\boldsymbol{\mu}\|_2^2}{\left(1 - \frac{C_4\epsilon}{k^2\sqrt{n}}\right)^2 kp^2} \leq \frac{C_5\epsilon}{\left(1 - \frac{C_4\epsilon}{k^2\sqrt{n}}\right)^2 k^2 n} \cdot \frac{\epsilon}{k^2\sqrt{n}}.$$

Thus, we get

$$|h_{mk}^{(1)}| \leq |h_{mk}^{(0)}| + \left|\frac{1}{\det_0}(\star)_h^{(0)}\right| \leq \left(1 + \frac{C_1}{\sqrt{n}} + \frac{C_5\epsilon}{\left(1 - \frac{C_4\epsilon}{k^2\sqrt{n}}\right)^2 k^2 n}\right) \frac{C_6\epsilon}{k^2\sqrt{n}} \leq (1 + \alpha)\frac{C_7\epsilon}{k^2\sqrt{n}},$$

and there exists constant $C_8$ such that $(1 + \alpha)C_7 \leq C_8$, which shows the desired upper bound.

**Bounds on $t_{kk}^{(1)}$.** Eqn. (75) in Appendix F.1 gives us

$$t_{kk}^{(1)} = t_{kk}^{(0)} - \frac{1}{\det_0}\left(\left(\|\boldsymbol{\mu}\|_2^2 - t_{11}^{(0)}\right)h_{1k}^{(0)^2} + 2t_{1k}^{(0)}h_{1k}^{(0)}h_{11}^{(0)} - s_{11}^{(0)}t_{1k}^{(0)^2} + 2t_{1k}^{(0)}h_{1k}^{(0)}\right).$$

We only need an upper bound on $t_{kk}^{(1)}$. The first dominant term $s_{11}^{(0)}t_{1k}^{(0)^2}/\det_0$ is upper bounded as follows:

$$\frac{s_{11}^{(0)}t_{1k}^{(0)^2}}{\det_0} \leq \frac{s_{11}^{(0)}t_{1k}^{(0)^2}}{(1 + h_{11}^{(0)})^2} \leq \frac{C_6 n^3\|\boldsymbol{\mu}\|_2^4}{\left(1 - \frac{C_3\epsilon}{k^2\sqrt{n}}\right)^2 kp^3} \leq \frac{C_7\epsilon^2}{\left(1 - \frac{C_3\epsilon}{k^2\sqrt{n}}\right)^2 pk^4 n} \cdot \frac{n\|\boldsymbol{\mu}\|_2^2}{p}.$$

Next, the second dominant term, $t_{1k}^{(0)} h_{1k}^{(0)} / \det_0$, is upper bounded as

$$\frac{t_{1k}^{(0)} h_{1k}^{(0)}}{\det_0} \leq \frac{|t_{1k}^{(0)} h_{1k}^{(0)}|}{(1 + h_{11}^{(0)})^2} \leq \frac{C_8 \epsilon}{k^2 n \left(1 - \frac{C_3 \epsilon}{k^2 \sqrt{n}}\right)^2} \cdot \frac{n \|\boldsymbol{\mu}\|_2^2}{p}.$$

Combining the results above gives us

$$t_{kk}^{(1)} \leq t_{kk}^{(0)} + \frac{1}{\det_0} \left| 2 t_{1k}^{(0)} h_{1k}^{(0)} h_{11}^{(0)} + s_{11}^{(0)} t_{1k}^{(0)^2} + 2 t_{1k}^{(0)} h_{1k}^{(0)} \right|$$

$$\leq \left(1 + \frac{C_9 \epsilon}{\left(1 - \frac{C_3 \epsilon}{k^2 \sqrt{n}}\right)^2 k^2 \sqrt{n}}\right) \frac{n \|\boldsymbol{\mu}\|_2^2}{p} \leq \frac{C_5 n \|\boldsymbol{\mu}\|_2^2}{p}.$$

This shows the desired upper bound.

**Bounds on $t_{mk}^{(1)}$.** Eqn. (74) in Appendix F.1 gives us

$$t_{mk}^{(1)} = t_{mk}^{(0)} - \frac{1}{\det_0} (\star)_t^{(0)},$$

where we define

$$(\star)_t^{(0)} = (\|\boldsymbol{\mu}\|_2^2 - t_{11}^{(0)}) h_{1m}^{(0)} h_{1k}^{(0)} + t_{m1}^{(0)} h_{1k}^{(0)} h_{11}^{(0)} + t_{k1}^{(0)} h_{1m}^{(0)} h_{11}^{(0)} + t_{1m}^{(0)} h_{1k}^{(0)} + t_{1k}^{(0)} h_{1m}^{(0)} - s_{11}^{(0)} t_{1m}^{(0)} t_{1k}^{(0)}.$$

Again, we only need an upper bound on $t_{mk}^{(1)}$. As in the previously derived bounds, we have

$$\frac{1}{\det_0} (\|\boldsymbol{\mu}\|_2^2 - t_{11}^{(0)}) h_{1m}^{(0)} h_{1k}^{(0)} \leq \frac{(\|\boldsymbol{\mu}\|_2^2 - t_{11}^{(0)}) |h_{1m}^{(0)} h_{1k}^{(0)}|}{s_{11}^{(0)} (\|\boldsymbol{\mu}\|_2^2 - t_{11}^{(0)})} \leq \frac{C_1 n^2 \|\boldsymbol{\mu}\|_2^2}{kp^2} \cdot \frac{kp}{n} \leq \frac{C_1 n \|\boldsymbol{\mu}\|_2^2}{p}.$$

The other dominant term $t_{1m}^{(0)} h_{1m}^{(0)} / \det_0$ is upper bounded as:

$$\frac{t_{1m}^{(0)} h_{1m}^{(0)}}{\det_0} \leq \frac{|t_{1m}^{(0)} h_{1m}^{(0)}|}{(1 + h_{11}^{(0)})^2} \leq \frac{C_2 \epsilon}{k^2 \sqrt{n} \left(\left(1 - \frac{C_3 \epsilon}{k^2 \sqrt{n}}\right)^2\right)} \cdot \frac{n \|\boldsymbol{\mu}\|_2^2}{p}.$$

Combining the results above yields

$$|t_{mk}^{(1)}| \leq |t_{mk}^{(0)}| + \frac{1}{\det_0} \left| (\star)_t^{(0)} \right|$$

$$\leq \left(C_1 + \frac{C_2 \epsilon}{\left(1 - \frac{C_3 \epsilon}{k^2 \sqrt{n}}\right)^2 k^2 \sqrt{n}}\right) \frac{n \|\boldsymbol{\mu}\|_2^2}{p} \leq \frac{C_4 n \|\boldsymbol{\mu}\|_2^2}{p}.$$

Note that both $t_{kk}^{(0)}$ and $t_{mk}^{(0)}$ are much smaller than $\|\boldsymbol{\mu}\|_2^2$. The above upper bound shows that this continues to hold for $t_{kk}^{(1)}$ and $t_{mk}^{(1)}$ since $p \gg n$.

**Bounds on $f_{ki}^{(1)}$.** Consider $i \in [n]$ and fix $y_i = k$ without loss of generality. Eqn. (76) in Appendix F.1 gives us

$$f_{ki}^{(1)} = f_{ki}^{(0)} - \frac{1}{\det_0} (\star)_f^{(0)}, \tag{44}$$

where we define

$$(\star)_f^{(0)} = (\|\boldsymbol{\mu}\|_2^2 - t_{11}^{(0)}) h_{1k}^{(0)} g_{1i}^{(0)} + t_{1k}^{(0)} g_{1i}^{(0)} + t_{1k}^{(0)} h_{11}^{(0)} g_{1i}^{(0)} + h_{1k}^{(0)} f_{1i}^{(0)} + h_{1k}^{(0)} h_{11}^{(0)} f_{1i}^{(0)} - s_{11}^{(0)} t_{1k}^{(0)} f_{1i}^{(0)}. \tag{45}$$

We only need an upper bound on $f_{ki}^{(1)}$. We consider the dominant terms $(\|\boldsymbol{\mu}\|_2^2 - t_{11}^{(0)})h_{1k}^{(0)}g_{1i}^{(0)}/\det_0$, $t_{1k}^{(0)}g_{1i}^{(0)}/\det_0$, $h_{1k}^{(0)}f_{1i}^{(0)}/\det_0$ and $s_{11}^{(0)}t_{1k}^{(0)}f_{1i}^{(0)}/\det_0$. Lemmas 4, 5 and 6 give us

$$\frac{(\|\boldsymbol{\mu}\|_2^2 - t_{11}^{(0)})h_{1k}^{(0)}g_{1i}^{(0)}}{\det_0} \leq \frac{(\|\boldsymbol{\mu}\|_2^2 - t_{11}^{(0)})|h_{1k}^{(0)}g_{1i}^{(0)}|}{(\|\boldsymbol{\mu}\|_2^2 - t_{11}^{(0)})s_{11}^{(0)}} \leq \frac{C_1 n\|\boldsymbol{\mu}\|_2}{\sqrt{kp}} \cdot \frac{1}{C_2 k^2 p} \cdot \frac{kp}{n} \leq \frac{C_3}{k^{1.5}\sqrt{n}} \cdot \frac{\sqrt{n}\|\boldsymbol{\mu}\|_2}{p},$$

$$\frac{t_{1k}^{(0)}g_{1i}^{(0)}}{\det_0} \leq \frac{|t_{1k}^{(0)}g_{1i}^{(0)}|}{(1 + h_{11}^{(0)})^2} \leq \frac{C_4 n\|\boldsymbol{\mu}\|_2^2}{\left(1 - \frac{C_5 \epsilon}{k^2\sqrt{n}}\right)^2 k^2 p^2} \leq \frac{C_7 \epsilon}{\left(1 - \frac{C_5 \epsilon}{k^2\sqrt{n}}\right)^2 k^{3.5}n} \cdot \frac{\sqrt{n}\|\boldsymbol{\mu}\|_2}{p},$$

$$\frac{h_{1k}^{(0)}f_{1i}^{(0)}}{\det_0} \leq \frac{|h_{1k}^{(0)}f_{1i}^{(0)}|}{(1 + h_{11}^{(0)})^2} \leq \frac{C_6 \epsilon}{\left(1 - \frac{C_5 \epsilon}{k^2\sqrt{n}}\right)^2 k^2\sqrt{n}} \cdot \frac{\sqrt{n}\|\boldsymbol{\mu}\|_2}{p}, \text{ and}$$

$$\frac{s_{11}^{(0)}t_{1k}^{(0)}f_{1i}^{(0)}}{\det_0} \leq \frac{|s_{11}^{(0)}t_{1k}^{(0)}f_{1i}^{(0)}|}{(1 + h_{11}^{(0)})^2} \leq \frac{C_7 \epsilon^2}{k^4 n\left(1 - \frac{C_5 \epsilon}{k^2\sqrt{n}}\right)^2} \cdot \frac{\sqrt{n}\|\boldsymbol{\mu}\|_2}{p},$$

where, in the last two steps, we used the (loose) upper bound $C\sqrt{n}\|\boldsymbol{\mu}\|_2/p$ for $|f_{ji}^{(0)}|$ and previously derived bounds on $|h_{1k}^{(0)}|$ and $|s_{11}^{(0)}t_{1k}^{(0)}|$. Thus, we have

$$|f_{ki}^{(1)}| \leq |f_{ki}^{(0)}| + \left|\frac{1}{\det_0}(\star)_f^{(0)}\right|$$

$$\leq \left(1 + \frac{C_3}{k^{1.5}\sqrt{n}} + \frac{C_8 \epsilon}{\left(1 - \frac{C_5 \epsilon}{k^2\sqrt{n}}\right)^2 k^2\sqrt{n}}\right)\frac{C_9\sqrt{n}\|\boldsymbol{\mu}\|_2}{p}$$

$$\leq (1 + \alpha)\frac{C_{10}\epsilon}{k^{1.5}n},$$

and we have $(1 + \alpha)C_{10} \leq C_{11}$ for a large enough positive constant $C_{11}$. This shows the desired upper bound.

**Bounds on $g_{ki}^{(1)}$ and $g_{mi}^{(1)}$.** Eqn. (77) in Appendix F.1 gives

$$z_{ci}\mathbf{e}_i^T \mathbf{A}_1^{-1}\mathbf{u}_k = |z_{ci}|^2 \left(\mathbf{e}_i^T \mathbf{A}_0^{-1}\mathbf{v}_k - \frac{1}{\det_0}(\star)_{gk}^{(0)}\right) = |z_{ci}|^2 \left(g_{ki}^{(0)} - \frac{1}{\det_0}(\star)_{gk}^{(0)}\right), \qquad (46)$$

where we define
$$(\star)_{gk}^{(0)} = (\|\boldsymbol{\mu}_1\|_2^2 - t_{11}^{(0)})s_{1k}^{(0)}g_{1i}^{(0)} + g_{1i}^{(0)}h_{11}^{(0)}h_{k1}^{(0)} + g_{1i}^{(0)}h_{k1}^{(0)} + s_{1k}^{(0)}f_{1i}^{(0)} + s_{1k}^{(0)}h_{11}^{(0)}f_{1i}^{(0)} - s_{11}^{(0)}h_{k1}^{(0)}f_{1i}^{(0)}.$$
Lemmas 4, 5 and 6 give us

$$\frac{(\|\boldsymbol{\mu}\|_2^2 - t_{11}^{(0)})|s_{1k}^{(0)}g_{1i}^{(0)}|}{\det_0} \leq \frac{(\|\boldsymbol{\mu}\|_2^2 - t_{11}^{(0)})|s_{1k}^{(0)}g_{1i}^{(0)}|}{(\|\boldsymbol{\mu}\|_2^2 - t_{11}^{(0)})s_{11}^{(0)}} \leq \frac{C_1}{\sqrt{n}} \cdot \frac{1}{C_2 k^2 p},$$

$$\frac{|h_{k1}^{(0)}g_{1i}^{(0)}|}{\det_0} \leq \frac{|h_{k1}^{(0)}g_{1i}^{(0)}|}{(1 + h_{11}^{(0)})^2} \leq \frac{C_3 \epsilon}{\left(1 - \frac{C_4 \epsilon}{k^2\sqrt{n}}\right)^2 k^2\sqrt{n}} \cdot \frac{1}{C_2 k^2 p}, \text{ and}$$

$$\frac{|s_{1k}^{(0)}f_{1i}^{(0)}|}{\det_0} \leq \frac{|s_{1k}^{(0)}f_{1i}^{(0)}|}{(1 + h_{11}^{(0)})^2} \leq \frac{C_5 \epsilon}{\left(1 - \frac{C_4 \epsilon}{k^2\sqrt{n}}\right)^2 \sqrt{k}\sqrt{n}} \cdot \frac{1}{C_2 k^2 p}.$$

We then have

$$g_{ki}^{(1)} \geq g_{ki}^{(0)} - \frac{1}{\det_0}|(\star)_{gk}^{(0)}| \geq \left(1 - \frac{1}{C}\right)\left(1 - \frac{C_1}{k^2\sqrt{n}} - \frac{C_6 \epsilon}{\left(1 - \frac{C_4 \epsilon}{k^2\sqrt{n}}\right)^2 k^{2.5}\sqrt{n}}\right)\frac{1}{p} \geq \left(1 - \frac{1}{C}\right)\frac{1 - \alpha}{p}$$

$$g_{ki}^{(1)} \leq g_{ki}^{(0)} + \frac{1}{\det_0}|(\star)_{gk}^{(0)}| \leq \left(1 + \frac{1}{C}\right)\left(1 + \frac{C_1}{k^2\sqrt{n}} + \frac{C_7 \epsilon}{\left(1 - \frac{C_4 \epsilon}{k^2\sqrt{n}}\right)^2 k^{2.5}\sqrt{n}}\right)\frac{1}{p} \leq \left(1 + \frac{1}{C}\right)\frac{1 + \alpha}{p},$$

where for large enough $n$ and positive constant $C_9$, we have $(1+\alpha)\frac{C+1}{C} \le \frac{C_9+1}{C_9}$ and $(1-\alpha)\frac{C-1}{C} \ge \frac{C_9-1}{C_9}$. Similarly, for the case $m \ne k$, we have

$$z_{ci}\mathbf{e}_i^T\mathbf{A}_1^{-1}\mathbf{u}_m = |z_{ci}|^2\left(\mathbf{e}_i^T\mathbf{A}_0^{-1}\mathbf{v}_m - \frac{1}{\det_0}(\star)_{gm}^{(0)}\right) = |z_{ci}|^2\left(g_{mi}^{(0)} - \frac{1}{\det_0}(\star)_{gm}^{(0)}\right), \qquad (47)$$

where we define
$(\star)_{gm}^{(0)} = (\|\boldsymbol{\mu}\|_2^2 - t_{11}^{(0)})s_{1m}^{(0)}g_{1i}^{(0)} + g_{1i}^{(0)}h_{11}^{(0)}h_{m1}^{(0)} + g_{1i}^{(0)}h_{m1}^{(0)} + s_{1m}^{(0)}f_{1i}^{(0)} + s_{1m}^{(0)}h_{11}^{(0)}f_{1i}^{(0)} - s_{11}^{(0)}h_{m1}^{(0)}f_{1i}^{(0)}$.
As a consequence of our equal energy and priors assumption (Assumption 1), we can directly use the bounds of the terms in $(\star)_{gk}^{(0)}$ to bound terms in $(\star)_{gm}^{(0)}$. We get

$$|g_{mi}^{(1)}| \le \frac{1}{C}\left(1 + \frac{C_1}{\sqrt{n}} + \frac{C_8\epsilon}{(1 - (C_4\epsilon/(k^2\sqrt{n})))^2\sqrt{k}\sqrt{n}}\right)\frac{1}{k^2p} \le \frac{1}{C}\cdot\frac{1+\alpha}{k^2p}.$$

Finally, there exists a sufficiently large constant $C_{10}$ such that $(1+\alpha)/C \le 1/C_{10}$. This shows the desired bounds.

### C.4.3 Completing the proof for $k$-th order quadratic forms

Notice from the above analysis that the 1-st order quadratic forms exhibit the same order-wise dependence on $n, k$ and $p$ as the 0-th order quadratic forms, e.g. both $s_{mk}^{(0)}$ and $s_{mk}^{(1)}$ are of order $\Theta(\frac{\sqrt{n}}{kp})$. Thus, the higher-order quadratic forms that arise by including more mean components will not change too much[5]. By Eqn. (29), we can see that we can bound the 2-nd order quadratic forms by bounding quadratic forms with order 1. We consider $s_{mk}^{(2)}$ as an example:

$$s_{mk}^{(2)} = s_{mk}^{(1)} - \frac{1}{\det_1}(\star)_s^{(1)},$$

where
$$(\star)_s^{(1)} := (\|\boldsymbol{\mu}\|_2^2 - t_{22}^{(1)})s_{2k}^{(1)}s_{2m}^{(1)} + s_{2m}^{(1)}h_{k2}^{(1)}h_{22}^{(1)} + s_{2k}^{(1)}h_{m2}^{(1)}h_{22}^{(1)} - s_{22}^{(1)}h_{k2}^{(1)}h_{m2}^{(1)} + s_{2m}^{(1)}h_{k2}^{(1)} + s_{2k}^{(1)}h_{m2}^{(1)},$$
$$\det_1 := s_{22}^{(1)}(\|\boldsymbol{\mu}\|_2^2 - t_{22}^{(1)}) + (1 + h_{22}^{(1)})^2.$$

We additionally show how $f_{ki}^{(2)}$ relates to the 1-st order quadratic forms:

$$f_{ki}^{(2)} = f_{ki}^{(1)} - \frac{1}{\det_1}(\star)_f^{(1)},$$

where we define
$(\star)_f^{(1)} = (\|\boldsymbol{\mu}\|_2^2 - t_{22}^{(1)})h_{2k}^{(1)}g_{2i}^{(1)} + t_{2k}^{(1)}g_{2i}^{(1)} + t_{2k}^{(1)}h_{22}^{(1)}g_{2i}^{(1)} + h_{2k}^{(1)}f_{2i}^{(1)} + h_{2k}^{(1)}h_{22}^{(1)}f_{2i}^{(1)} - s_{22}^{(1)}t_{2k}^{(1)}f_{2i}^{(1)}$.
Observe that the equations above are very similar to Equations (41) and (42) (for $s$), and Equations (44) and (45) (for $f$), except that the quadratic forms are in terms of Gram matrix $\mathbf{A}_1$. We have shown that the quadratic forms with order 1 will not be drastically different different from the quadratic forms with order 0. Hence, we repeat the above procedures of bounding these quadratic forms $k-1$ times to obtain the desired bounds in Lemma 2. The only quantity that will change in each iteration is $\alpha$, which nevertheless remains negligible[6].

Our analysis so far is conditioned on event $\mathcal{E}_q$. We define the *unconditional* event $\mathcal{E}_u := \{$all the inequalities in Lemma 2 hold$\}$. Then, we have
$$\mathbb{P}(\mathcal{E}_u^c) \le \mathbb{P}(\mathcal{E}_u^c|\mathcal{E}_q) + \mathbb{P}(\mathcal{E}_q^c) \le \mathbb{P}(\mathcal{E}_u^c|\mathcal{E}_q) + \mathbb{P}(\mathcal{E}_q^c|\mathcal{E}_v) + \mathbb{P}(\mathcal{E}_v^c)$$
$$\le \frac{c_1}{kn} + \frac{c_2}{n} + c_3k(e^{-\frac{n}{c_4}} + e^{-\frac{n}{c_5k^2}})$$
$$\le \frac{c_6}{n} + c_7ke^{-\frac{n}{c_5k^2}},$$
for constants $c_i$'s $> 1$. This completes the proof.

---

[5]There are several low-level reasons for this. One critical reason is the aforementioned orthogonality of the label indicator vectors $\{\mathbf{v}_c\}_{c\in[k]}$, which ensures by Lemma 5 that the cross-terms $|s_{mk}^{(j)}|$ are always dominated by the larger terms $|s_{kk}^{(j)}|$. Another reason is that $h_{mk}^{(0)}$, which can be seen as the "noise" term in our analysis, is small and thus does not affect other terms.

[6]To see this, recall that in the first iteration we had $\alpha_1 := \alpha = \frac{C_1}{\sqrt{n}} + \frac{C_2\tau}{(1-(C_5\tau/(k^2\sqrt{n})))^2k^2\sqrt{n}}$ for the first-order terms. Thus, even if we repeat the procedure $k-1$ times, then we have $\alpha_k \le Ck\alpha_1$, which remains small since we consider $n \gg k$.

## C.5   Proofs of Auxiliary lemmas

We complete this section by proving the auxiliary Lemmas 3, 5 and 6, which were used in the proof of Lemma 2.

### C.5.1   Proof of Lemma 3

Our goal is to upper and lower bound $\|\mathbf{v}_c\|_2^2$, for $c \in [k]$. Note that every entry of $\mathbf{v}_c$ is either 1 or 0, hence these entries are independent sub-Gaussian random variables with sub-Gaussian parameter 1 [Wai19, Chapter 2]. Under the equal-prior Assumption 1, we have $\mathbb{E}[\|\mathbf{v}_c\|_2^2] = n/k$ when we assume equal priors. Thus, a straightforward application of Hoeffding's concentration inequality on bounded random variables [Wai19, Chapter 2] gives us

$$\mathbb{P}\left(\left|\|\mathbf{v}_c\|_2^2 - \frac{n}{k}\right| \geq t\right) \leq 2\exp\left(-\frac{t^2}{2n}\right).$$

We complete the proof by setting $t = \frac{n}{C_1 k}$ for a large enough constant $C_1$ and applying the union bound over all $c \in [k]$.

### C.5.2   Proof of Lemma 5

We use the following lemma adapted from [MNS$^+$20, Lemma 2] to bound quadratic forms of inverse Wishart matrices.

**Lemma 7.** *Define* $p'(n) := (p - n + 1)$, *and consider matrix* $\mathbf{M} \sim \text{Wishart}(p, \mathbf{I}_n)$. *For any unit Euclidean norm vector* $\mathbf{v}$ *and any* $t > 0$, *we have*

$$\mathbb{P}\left(\frac{1}{\mathbf{v}^T \mathbf{M}^{-1} \mathbf{v}} > p'(n) + \sqrt{2tp'(n)} + 2t\right) \leq e^{-t} \quad and \quad \mathbb{P}\left(\frac{1}{\mathbf{v}^T \mathbf{M}^{-1} \mathbf{v}} < p'(n) - \sqrt{2tp'(n)}\right) \leq e^{-t},$$

*provided that* $p'(n) > 2\max\{t, 1\}$.

We first upper and lower bound $s_{cc}^{(0)}$ for a fixed $c \in [k]$. Recall that we assume $p > Cn\log(kn) + n - 1$ for sufficiently large constant $C > 1$ and this can be obtained by assuming $p'(n) > Cn\log(kn)$. Let $t = 2\log(kn)$. Working on the event $\mathcal{E}_v$ defined in (37), Lemma 7 gives us

$$s_{cc}^{(0)} \leq \frac{\|\mathbf{v}_c\|_2^2}{p'(n) - \sqrt{4\log(kn)p'(n)}} \leq \frac{C_1 + 1}{C_1} \cdot \frac{n/k}{p'(n)\left(1 - \frac{2}{\sqrt{Cn}}\right)} \leq \frac{C_2 + 1}{C_2} \cdot \frac{n}{kp}$$

with probability at least $1 - \frac{2}{k^2 n^2}$. Here, the last inequality comes from the fact that $p$ is sufficiently large compared to $n$ and $C$ is large enough. Similarly, for the lower bound, we have

$$s_{cc}^{(0)} \geq \frac{\|\mathbf{v}_c\|_2^2}{p'(n) + \sqrt{4\log(kn)p'(n)} + 2\log(kn)} \geq \frac{C_1 - 1}{C_1} \cdot \frac{n/k}{p'(n)\left(1 + \frac{4}{\sqrt{Cn}}\right)} \geq \frac{C_2 - 1}{C_2} \cdot \frac{n}{kp}$$

with probability $1 - \frac{2}{k^2 n^2}$.

Now we upper and lower bound $s_{cj}^{(0)}$ for a fixed choice $j \neq c \in [k]$. We use the parallelogram law to get

$$\mathbf{v}_c^T \mathbf{A}_0^{-1} \mathbf{v}_j = \frac{1}{4}\left((\mathbf{v}_c + \mathbf{v}_j)^T \mathbf{A}_0^{-1}(\mathbf{v}_c + \mathbf{v}_j) - (\mathbf{v}_c - \mathbf{v}_j)^T \mathbf{A}_0^{-1}(\mathbf{v}_c - \mathbf{v}_j)\right).$$

Because of the orthogonality of the label indicator vectors ($\mathbf{v}_c^T \mathbf{v}_j = 0$ for any $j \neq c$), we have $\|\mathbf{v}_c + \mathbf{v}_j\|_2^2 = \|\mathbf{v}_c - \mathbf{v}_j\|_2^2$, which we denote by $\tilde{n}$ as shorthand. Then, we have

$$\mathbf{v}_c^T \mathbf{A}_0^{-1} \mathbf{v}_j \leq \frac{1}{4}\left(\frac{\tilde{n}}{p'(n) - \sqrt{4\log(kn)p'(n)}} - \frac{\tilde{n}}{p'(n) + \sqrt{4\log(kn)p'(n)} + 4\log(kn)}\right)$$

$$\leq \frac{1}{4} \cdot \frac{2\tilde{n}\sqrt{4\log(kn)p'(n)} + 4\tilde{n}\log(kn)}{(p'(n) - \sqrt{4\log(kn)p'(n)})(p'(n) + \sqrt{4\log(kn)p'(n)})}$$

$$\leq \frac{C_1 + 1}{2C_1 k} \cdot \frac{2n\sqrt{4\log(kn)p'(n)} + 4n\log(kn)}{(p'(n) - \sqrt{4\log(kn)p'(n)})(p'(n) + \sqrt{4\log(kn)p'(n)})}$$

with probability at least $1 - \frac{2}{k^2 n^2}$ Here, the last inequality follows because we have $\tilde{n} \leq \frac{2(C_1+1)}{C_1} \cdot \frac{n}{k}$ on $\mathcal{E}_v$. Because $p'(n) > Cn \log(kn)$, we have

$$\mathbf{v}_c^T \mathbf{A}_0^{-1} \mathbf{v}_j \leq \frac{C_1+1}{2C_1 k} \cdot \frac{2\sqrt{n}p'(n) \cdot \sqrt{4/C} + 4/C \cdot p'(n)}{\left(1 - \sqrt{4/(Cn)}\right) p'(n)^2}$$

$$\leq \frac{C_1+1}{2C_1} \cdot \frac{\sqrt{n}}{k} \cdot \frac{2\sqrt{4/C} + \sqrt{4/(Cn)}}{p'(n)(1 - \sqrt{4/(Cn)})}$$

$$\leq \frac{C_2+1}{C_2} \cdot \frac{\sqrt{n}}{kp},$$

where in the last step we use the fact that $C > 1$ is large enough. To lower bound $s_{cj}^{(0)}$, we get

$$\mathbf{v}_c^T \mathbf{A}_0^{-1} \mathbf{v}_j \geq \frac{1}{4} \left( \frac{\tilde{n}}{(p'(n) + \sqrt{4\log(kn)p'(n)} + 4\log(kn))} - \frac{\tilde{n}}{(p'(n) - \sqrt{4\log(kn)p'(n)})} \right)$$

$$\geq \frac{1}{4} \cdot \frac{-2\tilde{n}\sqrt{4\log(kn)p'(n)} - 4\tilde{n}\log(kn)}{(p'(n) - \sqrt{4\log(kn)p'(n)})(p'(n) + \sqrt{4\log(kn)p'(n)})}$$

$$\geq -\frac{C_1+1}{2C_1 k} \cdot \frac{2n\sqrt{4\log(kn)p'(n)} + 4n\log(kn)}{(p'(n) - \sqrt{4\log(kn)p'(n)})(p'(n) + \sqrt{4\log(kn)p'(n)})}$$

with probability at least $1 - \frac{2}{k^2 n^2}$. Then following similar steps to the upper bound of $\mathbf{v}_c^T \mathbf{A}_0^{-1} \mathbf{v}_j$ gives us

$$\mathbf{v}_c^T \mathbf{A}_0^{-1} \mathbf{v}_j \geq -\frac{C_1+1}{2C_1 k} \cdot \frac{2\sqrt{n}p'(n)\sqrt{4/C} + (4/C)p'(n)}{(p'(n) - \sqrt{4/(Cn)}p'(n))p'(n)}$$

$$\geq -\frac{C_1+1}{2C_1} \cdot \frac{\sqrt{n}}{k} \cdot \frac{2\sqrt{4/C} + (4/C\sqrt{n})}{p'(n)(1 - \sqrt{4/(Cn)})}$$

$$\geq -\frac{C_2+1}{C_2} \cdot \frac{\sqrt{n}}{kp}.$$

We finally apply the union bound on all pairs of $c, j \in [k]$ and complete the proof.

### C.5.3   Proof of Lemma 6

We first lower and upper bound $g_{(y_i)i}^{(0)}$. Recall that we assumed $y_i = k$ without loss of generality. With a little abuse of notation, we define $\|\mathbf{v}_k\|_2^2 = \tilde{n}$ and $\mathbf{u} := \sqrt{\tilde{n}}\mathbf{e}_i$. We use the parallelogram law to get

$$\mathbf{e}_i^T \mathbf{A}_0^{-1} \mathbf{v}_k = \frac{1}{4\sqrt{\tilde{n}}} \left( (\mathbf{u} + \mathbf{v}_k)^T \mathbf{A}_0^{-1} (\mathbf{u} + \mathbf{v}_k) - (\mathbf{u} - \mathbf{v}_k)^T \mathbf{A}_0^{-1} (\mathbf{u} - \mathbf{v}_k) \right).$$

Note that $\|\mathbf{u} + \mathbf{v}_k\|_2^2 = 2(\tilde{n} + \sqrt{\tilde{n}})$ and $\|\mathbf{u} - \mathbf{v}_k\|_2^2 = 2(\tilde{n} - \sqrt{\tilde{n}})$. As before, we apply Lemma 7 with $t = 2\log(kn)$ to get with probability at least $1 - \frac{2}{k^2 n^2}$,

$$\mathbf{e}_i^T \mathbf{A}_0^{-1} \mathbf{v}_k \geq \frac{1}{4\sqrt{\tilde{n}}} \left( \frac{2(\tilde{n} + \sqrt{\tilde{n}})}{(p'(n) + \sqrt{4\log(kn)p'(n)} + 4\log(kn))} - \frac{2(\tilde{n} - \sqrt{\tilde{n}})}{(p'(n) - \sqrt{4\log(kn)p'(n)})} \right)$$

$$\geq \frac{1}{4\sqrt{\tilde{n}}} \cdot \frac{4\sqrt{\tilde{n}}p'(n) - 4\tilde{n}\sqrt{4\log(kn)p'(n)} - 8\tilde{n}\log(kn)}{(p'(n) + \sqrt{4\log(kn)p'(n)} + 4\log(kn))p'(n)}$$

$$\geq \frac{p'(n) - \sqrt{\tilde{n}}\sqrt{4\log(kn)p'(n)} - 2\sqrt{\tilde{n}}\log(kn)}{(p'(n) + \sqrt{4\log(kn)p'(n)} + 4\log(kn))p'(n)},$$

$$\geq \frac{p'(n) - \sqrt{(1 + 1/C_1)n/k}\sqrt{4\log(kn)p'(n)} - 2\sqrt{(1 + 1/C_1)n/k}\log(kn)}{(p'(n) + \sqrt{4\log(kn)p'(n)} + 4\log(kn))p'(n)}.$$

The last inequality works on event $\mathcal{E}_v$, by which we have $\tilde{n} \leq \frac{2(C_1+1)n}{C_1 k}$. Then, $p'(n) > Ck^3 n \log(kn)$ gives us

$$\mathbf{e}_i^T \mathbf{A}_0^{-1} \mathbf{v}_k \geq \frac{p'(n) - \sqrt{(1+1/C_1)n/k}\sqrt{4/(Ck^3n)}p'(n) - \sqrt{(1+1/C_1)n/k}(2/Ck^3n)p'(n)}{(p'(n) + \sqrt{4\log(kn)p'(n)} + 4\log(kn))p'(n)}$$

$$\geq \frac{1 - (1/(C_2\sqrt{k^4})) - (1/(C_3 k^{3.5}\sqrt{n}))}{p'(n)(1 + 2\sqrt{4/(Ck^3n)})}$$

$$\geq \frac{C_4 - 1}{C_4} \cdot \frac{1}{p},$$

where in the last step we use the fact that $C, C_2, C_3 > 1$ are large enough. To upper bound $g^{(0)}_{(y_i)i}$, we have with probability at least $1 - \frac{2}{k^2 n^2}$,

$$\mathbf{e}_i^T \mathbf{A}_0^{-1} \mathbf{v}_k \leq \frac{1}{4\sqrt{\tilde{n}}} \left( \frac{2(\tilde{n} + \sqrt{\tilde{n}})}{(p'(n) - \sqrt{4\log(kn)p'(n)})} - \frac{2(\tilde{n} - \sqrt{\tilde{n}})}{(p'(n) + \sqrt{4\log(kn)p'(n)} + 4\log(kn))} \right)$$

$$\leq \frac{1}{4\sqrt{\tilde{n}}} \cdot \frac{4\sqrt{\tilde{n}}p'(n) + 4\tilde{n}\sqrt{4\log(kn)p'(n)} + 8\tilde{n}\log(kn)}{(p'(n) - \sqrt{4\log(kn)p'(n)})p'(n)}$$

$$\leq \frac{p'(n) + \sqrt{\tilde{n}}\sqrt{4\log(kn)p'(n)} + 2\sqrt{\tilde{n}}\log(kn)}{(p'(n) - \sqrt{4\log(kn)p'(n)})p'(n)},$$

$$\leq \frac{p'(n) + \sqrt{(1+1/C_1)n/k}\sqrt{4\log(kn)p'(n)} + 2\sqrt{(1+1/C_1)n/k}\log(kn)}{(p'(n) - \sqrt{4\log(kn)p'(n)})p'(n)}.$$

Then $p'(n) > Ck^3 n \log(kn)$ gives us

$$\mathbf{e}_i^T \mathbf{A}_0^{-1} \mathbf{v}_k \leq \frac{p'(n) + \sqrt{(1+1/C_1)n/k}\sqrt{4/(Ck^3n)}p'(n) + 2\sqrt{(1+1/C_1)n/k}(4/Ck^3n)p'(n)}{(p'(n) - \sqrt{4\log(kn)p'(n)})p'(n)}$$

$$\leq \frac{1 + (1/(C_2\sqrt{k^4})) + (1/(C_3 k^{3.5}\sqrt{n}))}{p'(n)(1 - 2\sqrt{4/(Ck^3n)})}$$

$$\leq \frac{C_4 + 1}{C_4} \cdot \frac{1}{p}.$$

We now upper and lower bound $g^{(0)}_{ji}$ for a fixed $j \neq y_i$. As before, we have

$$\mathbf{e}_i^T \mathbf{A}_0^{-1} \mathbf{v}_j = \frac{1}{4\sqrt{\tilde{n}}} \left( (\mathbf{u} + \mathbf{v}_j)^T \mathbf{A}_0^{-1}(\mathbf{u} + \mathbf{v}_j) - (\mathbf{u} - \mathbf{v}_j)^T \mathbf{A}_0^{-1}(\mathbf{u} - \mathbf{v}_j) \right).$$

Since $\mathbf{e}_i^T \mathbf{v}_j = 0$, we now have $\|\mathbf{u} + \mathbf{v}_j\|_2^2 = \|\mathbf{u} - \mathbf{v}_j\|_2^2 = 2\tilde{n}$. We apply Lemma 7 with $t = 2\log(kn)$ to get, with probability at least $1 - \frac{2}{k^2 n^2}$,

$$\mathbf{e}_i^T \mathbf{A}_0^{-1} \mathbf{v}_j \leq \frac{1}{4\sqrt{\tilde{n}}} \left( \frac{2\tilde{n}}{(p'(n) - \sqrt{4\log(kn)p'(n)})} - \frac{2\tilde{n}}{(p'(n) + \sqrt{4\log(kn)p'(n)} + 4\log(kn))} \right)$$

$$\leq \frac{1}{4\sqrt{\tilde{n}}} \cdot \frac{4\tilde{n}\sqrt{4\log(kn)p'(n)} + 8\tilde{n}\log(kn)}{(p'(n) - \sqrt{4\log(kn)p'(n)})p'(n)}$$

$$\leq \frac{\sqrt{\tilde{n}}\sqrt{4\log(kn)p'(n)} + 2\sqrt{\tilde{n}}\log(kn)}{(p'(n) - \sqrt{4\log(kn)p'(n)})p'(n)},$$

$$\leq \frac{\sqrt{(1+1/C_1)n/k}\sqrt{4\log(kn)p'(n)} + 2\sqrt{(1+1/C_1)n/k}\log(kn)}{(p'(n) - \sqrt{4\log(kn)p'(n)})p'(n)}.$$

The last inequality works on event $\mathcal{E}_v$, by which we have $\tilde{n} \leq \frac{2(C_1+1)n}{C_1 k}$. Then, $p'(n) > Ck^3 n \log(kn)$ gives us

$$\mathbf{e}_i^T \mathbf{A}_0^{-1} \mathbf{v}_j \leq \frac{\sqrt{(1+1/C_1)n/k}\sqrt{4/(Ck^3 n)}p'(n) + \sqrt{(1+1/C_1)n/k}(2/Ck^3 n)p'(n)}{(p'(n) - \sqrt{4\log(kn)p'(n)})p'(n)}$$

$$\leq \frac{(1/(C_2\sqrt{k^4})) + (1/(C_3 k^{3.5}\sqrt{n}))}{p'(n)(1 - \sqrt{4/(Ck^3 n)})}$$

$$\leq \frac{C_4+1}{C_4} \cdot \frac{1}{k^2 p},$$

where in the last step we use the fact that $C, C_2, C_3 > 1$ are large enough. To lower bound $g_{ij}^{(0)}$, we have with probability at least $1 - \frac{2}{k^2 n^2}$,

$$\mathbf{e}_i^T \mathbf{A}_0^{-1} \mathbf{v}_j \geq \frac{1}{4\sqrt{\tilde{n}}} \left( \frac{2\tilde{n}}{(p'(n) + \sqrt{4\log(kn)p'(n)} + 4\log(kn))} - \frac{2\tilde{n}}{(p'(n) - \sqrt{4\log(kn)p'(n)})} \right)$$

$$\geq \frac{1}{4\sqrt{\tilde{n}}} \cdot \frac{-4\tilde{n}\sqrt{4\log(kn)p'(n)} - 8\tilde{n}\log(kn)}{(p'(n) - \sqrt{4\log(kn)p'(n)})p'(n)}$$

$$\geq -\frac{\sqrt{\tilde{n}}\sqrt{4\log(kn)p'(n)} + 2\sqrt{\tilde{n}}\log(kn)}{(p'(n) - \sqrt{4\log(kn)p'(n)})p'(n)},$$

$$\geq -\frac{\sqrt{(1+1/C_1)n/k}\sqrt{4\log(kn)p'(n)} + 2\sqrt{(1+1/C_1)n/k}\log(kn)}{(p'(n) - \sqrt{4\log(kn)p'(n)})p'(n)}.$$

Because $p'(n) > Ck^3 n \log(kn)$, we get

$$\mathbf{e}_i^T \mathbf{A}_0^{-1} \mathbf{v}_j \geq -\frac{\sqrt{(1+1/C_1)n/k}\sqrt{4/(Ck^3 n)}p'(n) + \sqrt{(1+1/C_1)n/k}(2/Ck^3 n)p'(n)}{(p'(n) - \sqrt{4\log(kn)p'(n)})p'(n)}$$

$$\geq -\frac{(1/(C_2\sqrt{k^4})) + (1/(C_3 k^{3.5}\sqrt{n}))}{p'(n)(1 - \sqrt{4/(Ck^3 n)})}$$

$$\geq -\frac{C_4+1}{C_4} \cdot \frac{1}{k^2 p},$$

where in the last step we use the fact that $C, C_2, C_3 > 1$ are large enough. We complete the proof by applying a union bounds over all $k$ classes and $n$ training examples.

## D  Proofs for Section 3.2.2

In this section, we provide the proofs of Theorems 3 and 4, which were discussed in Section 3.2.2. After having derived the interpolation condition in Equation (8) for multiclass SVM, these proofs are in fact rather simple extensions of the arguments provided in [MNS+20, HMX21] to the multiclass case. This is unlike the GMM case that we considered in Appendix C, which required substantial additional effort over and above the binary case [WT21].

### D.1  Proof of Theorem 3

For this section, we define $\mathbf{A} = \mathbf{X}^T \mathbf{X}$ as shorthand (we denoted the same quantity as $\mathbf{A}_k$ in Appendix C). Recall that the eigendecomposition of the covariance matrix is given by $\boldsymbol{\Sigma} = \sum_{i=1}^p \lambda_i \mathbf{v}_i \mathbf{v}_i^T = \boldsymbol{V} \boldsymbol{\Lambda} \boldsymbol{V}^T$. By rotation invariance of the standard normal variable, we can write $\mathbf{A} = \mathbf{Q}^T \boldsymbol{\Lambda} \mathbf{Q}$, where the entries of $\mathbf{Q} \in \mathbb{R}^{p \times n}$ are IID $\mathcal{N}(0,1)$ random variables. Finally, recall that we denoted $\boldsymbol{\lambda} = [\lambda_1 \quad \cdots \quad \lambda_p]$ and defined the effective dimensions $d_2 = \frac{\|\boldsymbol{\lambda}\|_1^2}{\|\boldsymbol{\lambda}\|_2^2}$ and $d_\infty = \frac{\|\boldsymbol{\lambda}\|_1}{\|\boldsymbol{\lambda}\|_\infty}$. Observe that Eqn. (8) in Thm. 1 is equivalent to the condition

$$z_{ci} \mathbf{e}_i^T \mathbf{A}^{-1} \mathbf{z}_c > 0, \quad \text{for all} \ c \in [k] \ \text{and} \ i \in [n]. \tag{48}$$

We fix $c \in [k]$ and drop the subscript $c$, using $\bar{\mathbf{z}}$ to denote the vector $\mathbf{z}_c$. We first provide a deterministic equivalence to Eqn. (8) that resembles the condition provided in [HMX21, Lemma 1]. Our proof is slightly modified compared to [HMX21, Lemma 1] and relies on elementary use of block matrix inversion identity.

**Lemma 8.** *Let* $\mathbf{Q} \in \mathbb{R}^{p \times n} = [\mathbf{q}_1, \cdots, \mathbf{q}_n]$. *In our notation, Eqn.* (8) *holds for a fixed $c$ if and only if:*

$$\frac{1}{z_i}\bar{\mathbf{z}}_{\backslash i}^T \left(\mathbf{Q}_{\backslash i}^T \mathbf{\Lambda} \mathbf{Q}_{\backslash i}\right)^{-1} \mathbf{Q}_{\backslash i}^T \mathbf{\Lambda} \mathbf{q}_i < 1, \ \text{for all} \ i = 1, \cdots, n. \tag{49}$$

*Above,* $\bar{\mathbf{z}}_{\backslash i} \in \mathbb{R}^{(n-1) \times 1}$ *is obtained by removing the $i$-th entry from vector $\bar{\mathbf{z}}$ and* $\mathbf{Q}_{\backslash i} \in \mathbb{R}^{d \times (n-1)}$ *is obtained by removing the $i$-th column from* $\mathbf{Q}$.

*Proof.* By symmetry, it suffices to consider the case $i = 1$. We first write

$$\mathbf{A} = \begin{bmatrix} \mathbf{q}_1^T \mathbf{\Lambda} \mathbf{q}_1 & \mathbf{q}_1^T \mathbf{\Lambda} \mathbf{Q}_{\backslash 1} \\ \mathbf{Q}_{\backslash 1}^T \mathbf{\Lambda} \mathbf{q}_1 & \mathbf{Q}_{\backslash 1}^T \mathbf{\Lambda} \mathbf{Q}_{\backslash 1} \end{bmatrix} \triangleq \begin{bmatrix} \alpha & \mathbf{b}^T \\ \mathbf{b} & \mathbf{D} \end{bmatrix}.$$

By Schur complement [Ber09], we have

$$\mathbf{A} \succ \mathbf{0} \ \textit{iff} \ \text{either} \ \left\{ \alpha > 0 \ \text{and} \ \mathbf{D} - \frac{\mathbf{b}\mathbf{b}^T}{\alpha} \succ \mathbf{0} \right\} \ \text{or} \ \left\{ \mathbf{D} \succ \mathbf{0} \ \text{and} \ \alpha - \mathbf{b}^T \mathbf{D}^{-1} \mathbf{b} > 0 \right\}.$$

Since the entries of $\mathbf{Q}$ are drawn from a continuous distribution (IID standard Gaussian), both $\mathbf{A}$ and $\mathbf{D} = \mathbf{Q}_{\backslash 1}^T \mathbf{\Lambda} \mathbf{Q}_{\backslash 1}$ are positive definite almost surely. Therefore, $\alpha - \mathbf{b}^T \mathbf{D}^{-1} \mathbf{b} > 0$ almost surely.

Thus, by block matrix inversion identity [Ber09], we have

$$\mathbf{A}^{-1} = \begin{bmatrix} (\alpha - \mathbf{b}^T \mathbf{D}^{-1} \mathbf{b})^{-1} & -(\alpha - \mathbf{b}^T \mathbf{D}^{-1} \mathbf{b})^{-1} \mathbf{b}^T \mathbf{D}^{-1} \\ -\mathbf{D}^{-1} \mathbf{b}(\alpha - \mathbf{b}^T \mathbf{D}^{-1} \mathbf{b})^{-1} & \mathbf{D}^{-1} + \mathbf{D}^{-1} \mathbf{b}(\alpha - \mathbf{b}^T \mathbf{D}^{-1} \mathbf{b})^{-1} \mathbf{b}^T \mathbf{D}^{-1} \end{bmatrix}.$$

Therefore, $\mathbf{e}_1^T \mathbf{A}^{-1} = (\alpha - \mathbf{b}^T \mathbf{D}^{-1} \mathbf{b})^{-1} \begin{bmatrix} 1 & -\mathbf{b}^T \mathbf{D}^{-1} \end{bmatrix}$. Hence we have

$$z_1 \mathbf{e}_1^T \mathbf{A}^{-1} \bar{\mathbf{z}} = (\alpha - \mathbf{b}^T \mathbf{D}^{-1} \mathbf{b})^{-1} (z_1^2 - \mathbf{b}^T \mathbf{D}^{-1}(z_1 \bar{\mathbf{z}}_{\backslash 1})),$$

where we use the fact that $\bar{\mathbf{z}}_1 = z_1$. Since $\alpha - \mathbf{b}^T \mathbf{D}^{-1} \mathbf{b} > 0$ almost surely, we have

$$z_1 \mathbf{e}_1^T \mathbf{A}^{-1} \bar{\mathbf{z}} > 0 \iff (\alpha - \mathbf{b}^T \mathbf{D}^{-1} \mathbf{b})^{-1} (z_1^2 - \mathbf{b}^T \mathbf{D}^{-1}(z_1 \bar{\mathbf{z}}_{\backslash 1})) > 0$$

$$\iff \frac{1}{z_1} \mathbf{b}^T \mathbf{D}^{-1} \bar{\mathbf{z}}_{\backslash 1} < 1.$$

Recall that $\mathbf{b}^T = \mathbf{q}_1^T \mathbf{\Lambda} \mathbf{Q}_{\backslash 1}$ and $\mathbf{D} = \mathbf{Q}_{\backslash 1}^T \mathbf{\Lambda} \mathbf{Q}_{\backslash 1}$. This completes the proof. $\qquad \square$

Next, we define the following events:

1. For $i \in [n]$, $\mathcal{B}_i := \left\{ \frac{1}{z_i} \bar{\mathbf{z}}_{\backslash i}^T \mathbf{A}_{\backslash i}^{-1} \mathbf{Q}_{\backslash i}^T \mathbf{\Lambda} \mathbf{q}_i \geq 1 \right\}$.

2. For $i \in [n]$, given $t > 0$, $\mathcal{E}_i(t) := \left\{ \|(\bar{\mathbf{z}}_{\backslash i}^T \mathbf{A}_{\backslash i}^{-1} \mathbf{Q}_{\backslash i}^T \mathbf{\Lambda})^T\|_2^2 \geq \frac{1}{t} \right\}$.

3. $\mathcal{B} := \cup_{i=1}^n \mathcal{B}_i$.

We know all the data points are support vectors i.e., Eqn. (48) holds, if none of the events $\mathcal{B}_i$ happens; hence, $\mathcal{B}$ is the undesired event. We want to upper bound the probability of event $\mathcal{B}$. As in the argument provided in [HMX21], we have

$$\mathbb{P}(\mathcal{B}) \leq \sum_{i=1}^n \left( \mathbb{P}(\mathcal{B}_i | \mathcal{E}_i(t)^c) + \mathbb{P}(\mathcal{E}_i(t)) \right). \tag{50}$$

The lemma below gives an upper bound on $\mathbb{P}(\mathcal{B}_i | \mathcal{E}_i(t)^c)$.

**Lemma 9.** *For any $t > 0$,* $\mathbb{P}(\mathcal{B}_i | \mathcal{E}_i(t)^c) \leq 2 \exp\left(-\frac{t}{2ck^2}\right)$.

*Proof.* On the event $\mathcal{E}_i(t)^c$, we have $\|(\overline{\mathbf{z}}_{\backslash i}^T \mathbf{A}_{\backslash i}^{-1} \mathbf{Q}_{\backslash i}^T \mathbf{\Lambda})^T\|_2^2 \leq \frac{1}{t}$. Since, by its definition, $|\frac{1}{z_i}| \leq k$, we have $\frac{1}{z_i}\overline{\mathbf{z}}_{\backslash i}^T \mathbf{A}_{\backslash i}^{-1} \mathbf{Q}_{\backslash i}^T \mathbf{\Lambda} \mathbf{q}_i$ is conditionally sub-Gaussian [Wai19, Chapter 2] with parameter at most $ck^2\|(\overline{\mathbf{z}}_{\backslash i}^T \mathbf{A}_{\backslash i}^{-1} \mathbf{Q}_{\backslash i}^T \mathbf{\Lambda})^T\|_2^2 \leq ck^2/t$. Then the sub-Gaussian tail bound gives

$$\mathbb{P}(\mathcal{B}_i | \mathcal{E}_i(t)^c) \leq 2\exp\left(-\frac{t}{2ck^2}\right), \tag{51}$$

which completes the prof. $\square$

Next we upper bound $\mathbb{P}(\mathcal{E}_i(t))$ with $t = d_\infty/(2n)$. Since $\|\mathbf{z}_{\backslash i}\|_2 \leq \|\mathbf{y}_{\backslash i}\|_2$, we can directly use [HMX21, Lemma 4].

**Lemma 10** (Lemma 4, [HMX21]). $\mathbb{P}\left(\mathcal{E}_i\left(\frac{d_\infty}{2n}\right)\right) \leq 2 \cdot 9^{n-1} \cdot \exp\left(-c_1 \min\left\{\frac{d_2}{4c^2}, \frac{d_\infty}{c}\right\}\right)$.

The results above are proved for fixed choices of $i \in [n]$ and $c \in [k]$. We combine Lemmas 9 and 10 with a union bound over all $n$ training examples and $k$ classes to upper bound the probability of the undesirable event $\mathcal{B}$ over all $k$ classes by:

$$kn9^{n-1} \cdot \exp\left(-c_1 \min\left\{\frac{d_2}{4c^2}, \frac{d_\infty}{c}\right\}\right) \leq \exp\left(-c_1 \min\left\{\frac{d_2}{4c^2}, \frac{d_\infty}{c}\right\} + C_1\log(kn) + C_2 n\right)$$

$$\text{and } 2kn \cdot \exp\left(-\frac{d_\infty}{2ck^2 n}\right) \leq \exp\left(-\frac{c_2 d_\infty}{ck^2 n} + C_3\log(kn)\right).$$

Thus, the probability that every data point is a support vector is at least

$$1 - \exp\left(-c_1 \min\left\{\frac{d_2}{4c^2}, \frac{d_\infty}{c}\right\} + C_1\log(kn) + C_2 n\right) - \exp\left(-\frac{c_2 d_\infty}{ck^2 n} + C_3\log(kn)\right).$$

To ensure that $\exp\left(-c_1 \min\left\{\frac{d_2}{4c^2}, \frac{d_\infty}{c}\right\} + C_1\log(kn) + C_2 n\right) + \exp\left(-\frac{c_2 d_\infty}{ck^2 n} + C_3\log(kn)\right) \leq \frac{c_4}{n}$, we consider the conditions $c_1 \min\left\{\frac{d_2}{4c^2}, \frac{d_\infty}{c}\right\} - C_1\log(kn) - C_2 n \geq \log(n)$ and $\frac{c_2 d_\infty}{ck^2 n} - C_3\log(kn) \geq \log(n)$ to be satisfied. These are equivalent to the conditions provided in Thm. 3. This completes the proof. Note that throughout the proof, we did not use any generative model assumptions on the labels given the covariates, so in fact our proof applies to scenarios beyond the MLM. $\square$

### D.2 Proof of Theorem 4

Now, we prove the sharper statement for the isotropic case. This proof is an extension of the proof argument for the second statement in [MNS+20, Theorem 1]. As before, we seek sufficient conditions under which

$$z_{ci}\mathbf{e}_i^T \mathbf{A}^{-1}\mathbf{z}_c > 0 \text{ for all } c \in [k] \text{ and } i \in [n]. \tag{52}$$

We again fix $i \in [n]$ and $c \in [k]$, and we use $\overline{\mathbf{z}}$ to denote the vector $\mathbf{z}_c$ as shorthand. We define $\mathbf{u}_i = \frac{1}{|z_i|}\|\overline{\mathbf{z}}\|_2 z_i \mathbf{e}_i$. We use the parallelogram law to get:

$$z_i \mathbf{e}_i^T \mathbf{A}^{-1}\overline{\mathbf{z}} = \frac{|z_i|}{\|\overline{\mathbf{z}}\|_2}\mathbf{u}_i^T \mathbf{A}^{-1}\overline{\mathbf{z}} = \frac{1}{4}\frac{|z_i|}{\|\overline{\mathbf{z}}\|_2}\left((\mathbf{u}_i + \overline{\mathbf{z}})^T \mathbf{A}^{-1}(\mathbf{u}_i + \overline{\mathbf{z}}) - (\mathbf{u}_i - \overline{\mathbf{z}})^T \mathbf{A}^{-1}(\mathbf{u}_i - \overline{\mathbf{z}})\right). \tag{53}$$

In the isotropic case, $\mathbf{A}^{-1}$ follows the inverse Wishart distribution. Thus, we can apply Lemma 7 with $t = 2\log(\sqrt{k}n)$. We know that $\|\mathbf{u}_i + \overline{\mathbf{z}}\|_2^2 = 2(\|\overline{\mathbf{z}}\|_2^2 + |z_i|\|\overline{\mathbf{z}}\|_2)$ and $\|\mathbf{u}_i - \overline{\mathbf{z}}\|_2^2 = 2(\|\overline{\mathbf{z}}\|_2^2 - |z_i|\|\overline{\mathbf{z}}\|_2)$. We consider the worst case $z_i = -\frac{1}{k}$. Under the condition $p'(n) = p - n + 1 > 10k^2 n\log(\sqrt{k}n)$, we have $\sqrt{4n\log(\sqrt{k}n)} < \frac{2}{3k}\sqrt{p'(n)}$. Thus, we have for all $i \in [n]$ and with probability at least

$1 - \frac{2}{kn^2}$:

$$z_i \boldsymbol{e}_i^T \mathbf{A}_0^{-1} \overline{\mathbf{z}} \geq \frac{|z_i|}{4\|\overline{\mathbf{z}}\|_2} \left( \frac{\|\overline{\mathbf{z}}\|_2^2 + |z_i|\|\overline{\mathbf{z}}\|_2}{(p'(n) + \sqrt{4\log(\sqrt{k}n)p'(n)} + 4\log(\sqrt{k}n))} - \frac{\|\overline{\mathbf{z}}\|_2^2 - |z_i|\|\overline{\mathbf{z}}\|_2}{(p'(n) - \sqrt{4\log(\sqrt{k}n)p'(n)})} \right)$$

$$\geq \frac{|z_i|}{8\|\overline{\mathbf{z}}\|_2} \cdot \frac{2|z_i|\|\overline{\mathbf{z}}\|_2 p'(n) - 2\|\overline{\mathbf{z}}\|_2^2\sqrt{4\log(\sqrt{k}n)p'(n)} - 4\|\overline{\mathbf{z}}\|_2^2\log(\sqrt{k}n)}{(p'(n) + \sqrt{4\log(\sqrt{k}n)p'(n)})(p'(n) - \sqrt{4\log(\sqrt{k}n)p'(n)})}$$

$$\geq \frac{|z_i|}{8} \cdot \frac{2|z_i|p'(n) - 2\sqrt{4n\log(\sqrt{k}n)p'(n)} - 4n\log(\sqrt{k}n)}{(p'(n) + \sqrt{4\log(\sqrt{k}n)p'(n)})(p'(n) - \sqrt{4\log(\sqrt{k}n)p'(n)})}, \tag{54}$$

where the last inequality uses the fact that $\|\overline{\mathbf{z}}\|_2 \leq \sqrt{n}$. When $|z_i| = 1/k$, we would like $2\sqrt{4n\log(\sqrt{k}n)p'(n)}$ in the numerator of (54) to be smaller than $p'(n)/(Ck)$, thus we would like $\sqrt{4n\log(\sqrt{k}n)} \leq \sqrt{p'(n)}/k$, which holds if $p'(n) > 9k^2 n\log(\sqrt{k}n)$. Under this condition, we get

$$(54) > \frac{|z_i|}{8} \cdot \frac{2|z_i|p'(n) - \frac{4}{3k}p'(n) - \frac{4}{9k^2}p'(n)}{(p'(n) + \sqrt{4/(9kn)}p'(n))p'(n)}$$

$$\geq \frac{|z_i|}{8} \cdot \frac{\frac{1}{k}(2p'(n) - \frac{4}{3}p'(n) - \frac{4}{9}p'(n))}{2p'(n)^2}$$

$$\geq \frac{|z_i|}{8} \cdot \frac{(2 - \frac{4}{3} - \frac{4}{9})}{2kp}$$

$$\geq \frac{|z_i|}{72kp},$$

where we again use the relation $4n\log(\sqrt{k}n) < \frac{4}{9k^2}p'(n)$, for all $i \in [n]$. Thus we have shown that under this condition, $z_{ci}\boldsymbol{e}_i^T \mathbf{A}^{-1}\mathbf{z}_c > 0$ for a fixed $i \in [n]$ and $c \in [k]$ with probability at least $1 - \frac{c_0}{kn^2}$. Finally, we apply the union bound over all $n$ examples and all $k$ classes, which gives $z_{ci}\boldsymbol{e}_i^T \mathbf{A}_0^{-1}\mathbf{z}_c > 0$ for all $i \in [n]$ and $c \in [k]$ with probability at least $1 - \frac{c_0}{n}$. This completes the proof. As in the proof of Thm. 3, we did not use any generative model assumptions on the labels given the covariates. $\qquad\square$

# E   Classification error proofs

In this section, we provide the proofs of classification error of the MNI under both GMM and MLM models. We begin with the proof for the GMM case (Thm. 5).

## E.1   Proof of Theorem 5

### E.1.1   Proof strategy and notations

The notation and main arguments of this proof follow closely the content of Section C.

Our starting point here is the lemma below (adapted from [TOS20, D.10]) that provides a simpler upper bound on the class-wise error $\mathbb{P}_{e|c}$.

**Lemma 11.** *Under GMM,* $\mathbb{P}_{e|c} \leq \sum_{j \neq c} Q\left( \frac{(\widehat{\mathbf{w}}_c - \widehat{\mathbf{w}}_j)^T \boldsymbol{\mu}_c}{\|\widehat{\mathbf{w}}_c - \widehat{\mathbf{w}}_j\|_2} \right)$. *In particular, if* $(\widehat{\mathbf{w}}_c - \widehat{\mathbf{w}}_j)^T \boldsymbol{\mu}_c > 0$, *then* $\mathbb{P}_{e|c} \leq \sum_{j \neq c} \exp\left( -\frac{((\widehat{\mathbf{w}}_c - \widehat{\mathbf{w}}_j)^T \boldsymbol{\mu}_c)^2}{4(\widehat{\mathbf{w}}_c^T \widehat{\mathbf{w}}_c + \widehat{\mathbf{w}}_j^T \widehat{\mathbf{w}}_j)} \right)$.

*Proof.* [TOS20, D.10] shows $\mathbb{P}_{e|c}$ is upper bounded by $\sum_{j\neq c} Q\left(\frac{(\widehat{\mathbf{w}}_c - \widehat{\mathbf{w}}_j)^T \boldsymbol{\mu}_c}{\|\widehat{\mathbf{w}}_c - \widehat{\mathbf{w}}_j\|_2}\right)$. Then if $(\widehat{\mathbf{w}}_c - \widehat{\mathbf{w}}_j)^T \boldsymbol{\mu}_c > 0$, the Chernoff bound [Wai19, Ch. 2] gives

$$\mathbb{P}_{e|c} \leq \sum_{j\neq c} \exp\left(-\frac{((\widehat{\mathbf{w}}_c - \widehat{\mathbf{w}}_j)^T \boldsymbol{\mu}_c)^2}{2\|\widehat{\mathbf{w}}_c - \widehat{\mathbf{w}}_j\|_2^2}\right) \leq \sum_{j\neq c} \exp\left(-\frac{((\widehat{\mathbf{w}}_c - \widehat{\mathbf{w}}_j)^T \boldsymbol{\mu}_c)^2}{4(\widehat{\mathbf{w}}_c^T \widehat{\mathbf{w}}_c + \widehat{\mathbf{w}}_j^T \widehat{\mathbf{w}}_j)}\right),$$

where the last inequality uses the identity $\mathbf{a}^T \mathbf{b} \leq 2(\mathbf{a}^T \mathbf{a} + \mathbf{b}^T \mathbf{b})$. $\qquad\square$

Thanks to Lemma 11, we can upper bound $P_{e|c}$ by lower bounding the terms

$$\frac{((\widehat{\mathbf{w}}_c - \widehat{\mathbf{w}}_j)^T \boldsymbol{\mu}_c)^2}{(\widehat{\mathbf{w}}_c^T \widehat{\mathbf{w}}_c + \widehat{\mathbf{w}}_j^T \widehat{\mathbf{w}}_j)}, \quad \text{for all } c \neq j \in [k]. \tag{55}$$

Our key observation is that this can be accomplished without the need to control the more intricate cross-correlation terms $\widehat{\mathbf{w}}_c^T \widehat{\mathbf{w}}_j$ for $c \neq j \in [k]$.

Without loss of generality, we assume onwards that $c = k$ and $j = k - 1$ (as in Appendix C). Similar to Appendix C, the quadratic forms introduced in Eqn. (28) play key role here, as well. For convenience, we recall the definitions of the *c-th order quadratic forms* for $c, j, m \in [k]$ and $i \in [n]$:

$$s_{mj}^{(c)} := \mathbf{v}_m^T \mathbf{A}_c^{-1} \mathbf{v}_j,$$
$$t_{mj}^{(c)} := \mathbf{d}_m^T \mathbf{A}_c^{-1} \mathbf{d}_j,$$
$$h_{mj}^{(c)} := \mathbf{v}_m^T \mathbf{A}_c^{-1} \mathbf{d}_j,$$
$$g_{ji}^{(c)} := \mathbf{v}_j^T \mathbf{A}_c^{-1} \mathbf{e}_i,$$
$$f_{ji}^{(c)} := \mathbf{d}_j^T \mathbf{A}_c^{-1} \mathbf{e}_i.$$

Further, recall that $\widehat{\mathbf{w}}_c = \mathbf{X}(\mathbf{X}^T \mathbf{X})^{-1} \mathbf{v}_c$ and $\mathbf{X} = \sum_{j=1}^{k} \boldsymbol{\mu}_j \mathbf{v}_j^T + \mathbf{Q}$. Also, from orthogonality of the class mean vectors (Assumption 2), we have $\boldsymbol{\mu}_c^T \mathbf{X} = \|\boldsymbol{\mu}\|_2^2 \mathbf{v}_c^T + \mathbf{d}_c^T$. Thus,

$$\widehat{\mathbf{w}}_c^T \boldsymbol{\mu}_c - \widehat{\mathbf{w}}_j^T \boldsymbol{\mu}_c = \|\boldsymbol{\mu}\|_2^2 \mathbf{v}_c^T (\mathbf{X}^T \mathbf{X})^{-1} \mathbf{v}_c + \mathbf{v}_c^T (\mathbf{X}^T \mathbf{X})^{-1} \mathbf{d}_c - \|\boldsymbol{\mu}\|_2^2 \mathbf{v}_c^T (\mathbf{X}^T \mathbf{X})^{-1} \mathbf{v}_j - \mathbf{v}_j^T (\mathbf{X}^T \mathbf{X})^{-1} \mathbf{d}_c. \tag{56}$$

Additionally,

$$\widehat{\mathbf{w}}_c^T \widehat{\mathbf{w}}_c = \mathbf{v}_c^T (\mathbf{X}^T \mathbf{X})^{-1} \mathbf{v}_c, \quad \text{and} \quad \widehat{\mathbf{w}}_j^T \widehat{\mathbf{w}}_j = \mathbf{v}_j^T (\mathbf{X}^T \mathbf{X})^{-1} \mathbf{v}_j.$$

Using the leave-one-out trick in Appendix C.1 and the matrix-inversion lemma, we show in Appendix E.1.5 that (55) is

$$\left(\frac{\|\boldsymbol{\mu}\|_2^2 s_{cc}^{(j)} - s_{cc}^{(j)} t_{cc}^{(j)} + {h_{cc}^{(j)}}^2 + h_{cc}^{(j)} - \|\boldsymbol{\mu}\|_2^2 s_{jc}^{(j)} - h_{jc}^{(j)} - h_{jc}^{(j)} h_{cc}^{(j)} + s_{jc}^{(j)} t_{cc}^{(j)}}{\det_j}\right)^2 \Big/ \left(\frac{s_{cc}^{(j)}}{\det_j} + \frac{s_{jj}^{(-j)}}{\det_{-j}}\right), \tag{57}$$

where $\det_j = (\|\boldsymbol{\mu}\|_2^2 - t_{cc}^{(j)}) s_{cc}^{(j)} + (h_{cc}^{(j)} + 1)^2$. Note that $\det_j = \det_{-c}$ when $c = k$ and $j = k - 1$.

Next, we will prove that

$$(57) \geq \|\boldsymbol{\mu}\|_2^2 \frac{\left(\left(1 - \frac{C_1}{\sqrt{n}} - \frac{C_2 n}{p}\right) \|\boldsymbol{\mu}\|_2 - C_3 \sqrt{k}\right)^2}{C_6 \left(\|\boldsymbol{\mu}\|_2^2 + \frac{kp}{n}\right)}. \tag{58}$$

### E.1.2 Proof of Equation (58)

We will lower bound the numerator and upper bound the denominator of Eqn. (57). We will work on the high-probability event $\mathcal{E}_v$ defined in Eqn. (37) in Appendix C.3. For quadratic forms such as

$s_{cc}^{(j)}, t_{cc}^{(j)}$ and $h_{cc}^{(j)}$, the Gram matrix $\mathbf{A}_j^{-1}$ does not "include" the $c$-th mean component because we have fixed $c = k, j = k-1$. Thus, we can directly apply Lemma 2 to get

$$\frac{C_1 - 1}{C_1} \cdot \frac{n}{kp} \le s_{cc}^{(j)} \le \frac{C_1 + 1}{C_1} \cdot \frac{n}{kp},$$

$$t_{cc}^{(j)} \le \frac{C_2 n \|\boldsymbol{\mu}\|_2^2}{p},$$

$$-\frac{C_3 n \|\boldsymbol{\mu}\|_2}{\sqrt{kp}} \le h_{cc}^{(j)} \le \frac{C_3 n \|\boldsymbol{\mu}\|_2}{\sqrt{kp}},$$

on the event $\mathcal{E}_v$. We need some additional work to bound $s_{jc}^{(j)} = \mathbf{v}_j \mathbf{A}_j^{-1} \mathbf{v}_c$ and $h_{jc}^{(j)} = \mathbf{v}_j \mathbf{A}_j^{-1} \mathbf{d}_c$, since the Gram matrix $\mathbf{A}_j^{-1}$ "includes" $\mathbf{v}_j$. The proof here follows the machinery introduced in Appendix C.4 for proving Lemma 2. We provide the core argument and refer the reader therein for additional justifications. By Eqn. (71) in Appendix F.1 (with the index $j-1$ replacing the index 0), we first have

$$s_{jc}^{(j)} = s_{jc}^{(j-1)} - \frac{1}{\det_{j-1}}(\star)_s^{(j-1)},$$

where we define

$$(\star)_s^{(j-1)} = (\|\boldsymbol{\mu}\|_2^2 - t_{jj}^{(j-1)})s_{jj}^{(j-1)}s_{jc}^{(j-1)} + s_{jc}^{(j-1)}h_{jj}^{(j-1)2} + s_{jc}^{(j-1)}h_{jj}^{(j-1)} + s_{jj}^{(j-1)}h_{jc}^{(j-1)},$$

and $\det_{j-1} = (\|\boldsymbol{\mu}\|_2^2 - t_{jj}^{(j-1)})s_{jj}^{(j-1)} + (h_{jj}^{(j-1)} + 1)^2$. Further, we have

$$|s_{jc}^{(j)}| = \left| \left( 1 - \frac{(\|\boldsymbol{\mu}\|_2^2 - t_{jj}^{(j-1)})s_{jj}^{(j-1)} + h_{jj}^{(j-1)2}}{\det_{j-1}} \right) s_{jc}^{(j-1)} - \frac{1}{\det_{j-1}}(s_{jc}^{(j-1)}h_{jj}^{(j-1)} + s_{jj}^{(j-1)}h_{jc}^{(j-1)}) \right|$$

$$\le \frac{1}{C}|s_{jc}^{(j-1)}| + \frac{1}{\det_{j-1}}|(s_{jc}^{(j-1)}h_{jj}^{(j-1)} + s_{jj}^{(j-1)}h_{jc}^{(j-1)})|.$$

We focus on the dominant term $|s_{jj}^{(j-1)}h_{jc}^{(j-1)}|$. Using a similar argument to that provided in Appendix C.4, we get

$$\frac{|s_{jj}^{(j-1)}h_{jc}^{(j-1)}|}{\det_{j-1}} \le \frac{|s_{jj}^{(j-1)}h_{jc}^{(j-1)}|}{(1 + h_{jj}^{(j-1)})^2} \le \frac{C_1}{\left(1 - \frac{C_2\epsilon}{k^2\sqrt{n}}\right)^2} \cdot \frac{n}{kp} \cdot \frac{\epsilon}{k^2\sqrt{n}} \le \frac{C_3\epsilon}{\left(1 - \frac{C_2\epsilon}{k^2\sqrt{n}}\right)^2 k^2} \cdot \frac{\sqrt{n}}{kp}.$$

Thus, we have

$$|s_{jc}^{(j-1)}| \le \frac{C_4 + 1}{C_4} \cdot \frac{\sqrt{n}}{kp}.$$

Similarly, we bound the remaining term $h_{jc}^{(j)}$. Specifically, by Eqn. (73) in Section F.1, we have

$$h_{jc}^{(j)} = h_{jc}^{(j-1)} - \frac{1}{\det_{j-1}}(\star)_h^{(j-1)},$$

where we define

$$(\star)_h^{(j-1)} = (\|\boldsymbol{\mu}\|_2^2 - t_{jj}^{(j-1)})s_{jj}^{(j-1)}h_{jc}^{(j-1)} + h_{jc}^{(j-1)}h_{jj}^{(j-1)2} + h_{jc}^{(j-1)}h_{jj}^{(j-1)} + s_{jj}^{(j-1)}t_{jc}^{(j-1)}.$$

Furthermore,

$$|h_{jc}^{(j)}| = \left| \left( 1 - \frac{(\|\boldsymbol{\mu}\|_2^2 - t_{jj}^{(j-1)})s_{jj}^{(j-1)} + h_{jj}^{(j-1)2}}{\det_{j-1}} \right) h_{jc}^{(j-1)} - \frac{1}{\det_{j-1}}(h_{jc}^{(j-1)}h_{jj}^{(j-1)} + s_{jj}^{(j-1)}t_{jc}^{(j-1)}) \right|$$

$$\le \frac{1}{C}|h_{jc}^{(j-1)}| + \frac{1}{\det_{j-1}}|(h_{jc}^{(j-1)}h_{jj}^{(j-1)} + s_{jj}^{(j-1)}t_{jc}^{(j-1)})|.$$

We again consider the dominant term $|s_{jj}^{(j-1)}t_{jc}^{(j-1)}|/\det_{j-1}$ and get

$$\frac{|s_{jj}^{(j-1)}t_{jc}^{(j-1)}|}{\det_{j-1}} \leq \frac{|s_{jj}^{(j-1)}t_{jc}^{(j-1)}|}{(1+h_{jj}^{(j-1)})^2} \leq \frac{C_1}{\left(1 - \frac{C_2\epsilon}{k^2\sqrt{n}}\right)^2} \cdot \frac{n}{kp} \cdot \frac{n\|\boldsymbol{\mu}\|_2^2}{p} \leq \frac{C_3\epsilon}{\left(1 - \frac{C_2\epsilon}{k^2\sqrt{n}}\right)^2 k^2\sqrt{n}} \cdot \frac{n\|\boldsymbol{\mu}\|_2}{\sqrt{kp}}.$$

Thus, we find that

$$|h_{jc}^{(j-1)}| \leq \frac{C_4 n\|\boldsymbol{\mu}\|_2}{\sqrt{kp}}.$$

We are now ready to lower bound the RHS in Eqn. (57) by lower bounding its numerator and upper bounding its denominator.

First, for the numerator we have the following sequence of inequalities:

$$\|\boldsymbol{\mu}\|_2^2 s_{cc}^{(j)} - s_{cc}^{(j)} t_{cc}^{(j)} + {h_{cc}^{(j)}}^2 + h_{cc}^{(j)} - \|\boldsymbol{\mu}\|_2^2 s_{jc}^{(j)} - h_{jc}^{(j)} - h_{jc}^{(j)} h_{cc}^{(j)} + s_{jc}^{(j)} t_{cc}^{(j)}$$

$$\geq \|\boldsymbol{\mu}\|_2^2 s_{cc}^{(j)} - \|\boldsymbol{\mu}\|_2^2 s_{jc}^{(j)} - s_{cc}^{(j)} t_{cc}^{(j)} + s_{jc}^{(j)} t_{cc}^{(j)} + h_{cc}^{(j)} - h_{jc}^{(j)} - h_{jc}^{(j)} h_{cc}^{(j)}$$

$$\geq \frac{C_1 - 1}{C_1} \cdot \frac{\|\boldsymbol{\mu}\|_2^2 n}{kp} - \frac{C_2 + 1}{C_2} \cdot \frac{\|\boldsymbol{\mu}\|_2^2 \sqrt{n}}{kp} - \frac{C_3 n}{p} \cdot \frac{\|\boldsymbol{\mu}\|_2^2 n}{kp} - \frac{C_4 n}{p} \cdot \frac{\|\boldsymbol{\mu}\|_2^2 \sqrt{n}}{kp} - \frac{C_5 n\|\boldsymbol{\mu}\|_2}{\sqrt{kp}}.$$

Above, we use the fact that the terms $|h_{cc}^{(j)}|, |h_{jc}^{(j)}| \leq C\epsilon/(k^2\sqrt{n})$ are sufficiently small compared to 1. Consequently, the numerator is lower bounded by

$$\left(\frac{C_1 - 1}{C_1} \cdot \frac{\|\boldsymbol{\mu}\|_2^2 n}{kp} - \frac{C_2 + 1}{C_2} \cdot \frac{\|\boldsymbol{\mu}\|_2^2 \sqrt{n}}{kp} - \frac{C_3 n}{p} \cdot \frac{\|\boldsymbol{\mu}\|_2^2 n}{kp} - \frac{C_4 n}{p} \cdot \frac{\|\boldsymbol{\mu}\|_2^2 \sqrt{n}}{kp} - \frac{C_5 n\|\boldsymbol{\mu}\|_2}{\sqrt{kp}}\right)^2 \Big/ \det_j^2. \tag{59}$$

Second, we upper bound the denominator. For this, note that under the assumption of equal energy and equal priors on class means (Assumption 1), there exist constants $C_1, C_2 > 0$ such that $C_1 \leq \det_j / \det_{-j} \leq C_2$. (In fact, a very similar statement was proved in Eqn. (34) and used in the proof of Theorem 2). Moreover, Lemma 2 shows that the terms $s_{cc}^{(j)}$ and $s_{jj}^{(-j)}$ are of the same order, so it suffices to upper bound $\frac{s_{cc}^{(j)}}{\det_j}$. Again applying Lemma 2, we have

$$\frac{s_{cc}^{(j)}}{\det_j} \leq \frac{C_6}{\det_j} \cdot \frac{n}{kp} \tag{60}$$

on the event $\mathcal{E}_v$. Then, combining Equations (59) and (60) gives us

$$(57) \geq \frac{n}{C_0 kp} \cdot \frac{1}{\det_j} \left((1 - \frac{C_1}{\sqrt{n}} - \frac{C_2 n}{p})\|\boldsymbol{\mu}\|_2^2 - C_3\sqrt{k}\|\boldsymbol{\mu}\|_2\right)^2$$

$$\geq \frac{n}{C_0 kp} \cdot \frac{1}{\frac{C_4\|\boldsymbol{\mu}\|_2^2 n}{kp} + 2 + \frac{C_5 n^2\|\boldsymbol{\mu}\|_2^2}{kp^2}} \left(\left(1 - \frac{C_1}{\sqrt{n}} - \frac{C_2 n}{p}\right)\|\boldsymbol{\mu}\|_2^2 - C_3\sqrt{k}\|\boldsymbol{\mu}\|_2\right)^2$$

$$\geq \|\boldsymbol{\mu}\|_2^2 \frac{\left(\left(1 - \frac{C_1}{\sqrt{n}} - \frac{C_2 n}{p}\right)\|\boldsymbol{\mu}\|_2 - C_3\sqrt{k}\right)^2}{C_6\left(\|\boldsymbol{\mu}\|_2^2 + \frac{kp}{n}\right)}, \tag{61}$$

where the second inequality follows from the following upper bound on $\det_j$ on the event $\mathcal{E}_v$:

$$\det_j = (\|\boldsymbol{\mu}\|_2^2 - t_{cc}^{(j)})s_{cc}^{(j)} + (h_{cc}^{(j)} + 1)^2 \leq \|\boldsymbol{\mu}\|_2^2 s_{cc}^{(j)} + 2({h_{cc}^{(j)}}^2 + 1) \leq \frac{C_4\|\boldsymbol{\mu}\|_2^2 n}{kp} + 2 + \frac{C_5 n^2\|\boldsymbol{\mu}\|_2^2}{kp^2}.$$

### E.1.3 Completing the proof

Because of our assumption of equal energy on class means and equal priors, the analysis above can be applied to bound $\frac{((\widehat{\mathbf{w}}_c - \widehat{\mathbf{w}}_j)^T \boldsymbol{\mu}_c)^2}{(\widehat{\mathbf{w}}_c^T \widehat{\mathbf{w}}_c + \widehat{\mathbf{w}}_j^T \widehat{\mathbf{w}}_j)}$, for every $j \neq c$ and $c \in [k]$. We define the *unconditional* event

$$\mathcal{E}_{u2} := \left\{\frac{((\widehat{\mathbf{w}}_c - \widehat{\mathbf{w}}_j)^T \boldsymbol{\mu}_c)^2}{(\widehat{\mathbf{w}}_c^T \widehat{\mathbf{w}}_c + \widehat{\mathbf{w}}_j^T \widehat{\mathbf{w}}_j)} \text{ is lower bounded by (61) for every } j \neq c\right\}.$$

We have

$$\mathbb{P}(\mathcal{E}_{u2}^c) \leq \mathbb{P}(\mathcal{E}_{u2}^c | \mathcal{E}_v) + \mathbb{P}(\mathcal{E}_v^c)$$
$$\leq \frac{c_4}{n} + c_5 k(e^{-\frac{n}{c_6}} + e^{-\frac{n}{c_7 k^2}}) \leq \frac{c_4}{n} + c_8 k e^{-\frac{n}{c_7 k^2}}$$

for constants $c_i$'s $> 1$. Thus, the class-wise error $\mathbb{P}_{e|c}$ is upper bounded by

$$(k-1) \exp\left( -\|\boldsymbol{\mu}\|_2^2 \frac{\left( \left(1 - \frac{C_1}{\sqrt{n}} - \frac{C_2 n}{p}\right) \|\boldsymbol{\mu}\|_2 - C_3\sqrt{k} \right)^2}{C_4\left(\|\boldsymbol{\mu}\|_2^2 + \frac{kp}{n}\right)} \right)$$

with probability at least $1 - \frac{c_4}{n} - c_8 k e^{-\frac{n}{c_7 k^2}}$. This completes the proof. $\qquad\square$

### E.1.4 Proof of Equation (57)

Here, using the results of Section F.1, we show how to obtain Eqn. (57) from Eqn. (55). First, by [WT21, Appendix C.2] (with $\mathbf{y}$ replaced by $\mathbf{v}_m$), we have

$$\mathbf{v}_m (\mathbf{X}^T \mathbf{X})^{-1} \mathbf{v}_m = \frac{s_{mm}^{(-m)}}{\det_{-m}}, \quad \text{for all } m \in [k],$$

where $\det_{-m} = (\|\boldsymbol{\mu}\|_2^2 - t_{mm}^{(-m)}) s_{mm}^{(-m)} + (h_{mm}^{(-m)} + 1)^2$. Then [WT21, Equation (44)] gives

$$\|\boldsymbol{\mu}_c\|_2^2 \cdot \mathbf{v}_c (\mathbf{X}^T \mathbf{X})^{-1} \mathbf{v}_c + \mathbf{v}_c (\mathbf{X}^T \mathbf{X})^{-1} \mathbf{d}_c = \frac{\|\boldsymbol{\mu}_c\|_2^2 s_{cc}^{(j)} - s_{cc}^{(j)} t_{cc}^{(j)} + h_{cc}^{(j)^2} + h_{cc}^{(j)}}{\det_j},$$

where $\det_j = (\|\boldsymbol{\mu}\|_2^2 - t_{cc}^{(j)}) s_{cc}^{(j)} + (h_{cc}^{(j)} + 1)^2$. Note that $\det_j = \det_{-c}$ when $c = k$ and $j = k - 1$.

For $\mathbf{v}_c (\mathbf{X}^T \mathbf{X})^{-1} \mathbf{v}_j$ and $\mathbf{v}_j (\mathbf{X}^T \mathbf{X})^{-1} \mathbf{d}_c$, we can again express the $k$-th order quadratic forms in terms of $j$-th order quadratic forms as follows:

$$\mathbf{v}_c (\mathbf{X}^T \mathbf{X})^{-1} \mathbf{v}_j = \frac{s_{cj}^{(j)} + s_{cj}^{(j)} h_{cc}^{(j)} - s_{cc}^{(j)} h_{jc}^{(j)}}{\det_j},$$

$$\mathbf{v}_j (\mathbf{X}^T \mathbf{X})^{-1} \mathbf{d}_c = \frac{\|\boldsymbol{\mu}\|_2^2 s_{cc}^{(j)} h_{jc}^{(j)} - \|\boldsymbol{\mu}\|_2^2 s_{cj}^{(j)} h_{cc}^{(j)} + h_{cc}^{(j)} h_{jc}^{(j)} + h_{jc}^{(j)} - s_{cj}^{(j)} t_{cc}^{(j)}}{\det_j}.$$

Thus, we have

$$\|\boldsymbol{\mu}\|_2^2 \mathbf{v}_c (\mathbf{X}^T \mathbf{X})^{-1} \mathbf{v}_j + \mathbf{v}_j (\mathbf{X}^T \mathbf{X})^{-1} \mathbf{d}_c = \frac{\|\boldsymbol{\mu}_c\|_2^2 s_{jc}^{(j)} + h_{jc}^{(j)} + h_{jc}^{(j)} h_{cc}^{(j)} - s_{jc}^{(j)} t_{cc}^{(j)}}{\det_j}.$$

This completes the proof. $\qquad\square$

### E.1.5 Extensions to unorthogonal means

While we made the orthogonality assumption on class means (Assumption 2) for simplicity, our error analysis can conceivably be extended to the more general unorthogonal setting. We provide a brief discussion of this extension here. To upper bound the class-wise error $\mathbb{P}_{e|c}$, recall that we need to lower bound the quantity in Eqn. (55). As with the orthogonal case, we consider $c = k, j = k - 1$ without loss of generality. Recall that $\widehat{\mathbf{w}}_c = \mathbf{X}(\mathbf{X}^T \mathbf{X})^{-1} \mathbf{v}_c$ and $\mathbf{X} = \sum_{j=1}^{k} \boldsymbol{\mu}_j \mathbf{v}_j^T + \mathbf{Q}$. Thus

$$\widehat{\mathbf{w}}_c^T \boldsymbol{\mu}_c = \|\boldsymbol{\mu}\|_2^2 \mathbf{v}_c^T (\mathbf{X}^T \mathbf{X})^{-1} \mathbf{v}_c + \sum_{m \neq c} \boldsymbol{\mu}_m^T \boldsymbol{\mu}_c \mathbf{v}_m^T (\mathbf{X}^T \mathbf{X})^{-1} \mathbf{v}_c + \mathbf{v}_c^T (\mathbf{X}^T \mathbf{X})^{-1} \mathbf{d}_c \text{ and}$$

$$\widehat{\mathbf{w}}_j^T \boldsymbol{\mu}_c = \|\boldsymbol{\mu}\|_2^2 \mathbf{v}_j^T (\mathbf{X}^T \mathbf{X})^{-1} \mathbf{v}_c + \boldsymbol{\mu}_j^T \boldsymbol{\mu}_c \mathbf{v}_j^T (\mathbf{X}^T \mathbf{X})^{-1} \mathbf{v}_j + \sum_{m \neq c,j} \boldsymbol{\mu}_m^T \boldsymbol{\mu}_c \mathbf{v}_m^T (\mathbf{X}^T \mathbf{X})^{-1} \mathbf{v}_j + \mathbf{v}_j^T (\mathbf{X}^T \mathbf{X})^{-1} \mathbf{d}_c.$$

We have already obtained the bounds for $\|\boldsymbol{\mu}\|_2^2 \mathbf{v}_c^T (\mathbf{X}^T \mathbf{X})^{-1} \mathbf{v}_c + \mathbf{v}_c^T (\mathbf{X}^T \mathbf{X})^{-1} \mathbf{d}_c - \|\boldsymbol{\mu}\|_2^2 \mathbf{v}_c^T (\mathbf{X}^T \mathbf{X})^{-1} \mathbf{v}_j - \mathbf{v}_j^T (\mathbf{X}^T \mathbf{X})^{-1} \mathbf{d}_c$ in Appendix E.1. Moreover, under the equal energy and priors assumption, the $\mathbf{v}_j^T (\mathbf{X}^T \mathbf{X})^{-1} \mathbf{v}_j$ terms have the same bound for every $j \in [k]$. Similarly, the $\mathbf{v}_j^T (\mathbf{X}^T \mathbf{X})^{-1} \mathbf{v}_m$ terms also have the same bound for all $j \neq m \in [k]$. An upper bound on classification error can then be derived in terms of the inner products between the mean vectors. We leave the detailed derivation to the reader. Expressions are naturally more complicated.

## E.2 Proof of Corollary 2

We now prove the condition for benign overfitting provided in Corollary 2. Note that following Theorem 2, we assume that

$$p > C_1 k^3 n \log(kn) + n - 1 \quad \text{and} \quad p > C_2 k^{1.5} n^{1.5} \|\boldsymbol{\mu}\|_2. \tag{62}$$

We begin with the setting where $\|\boldsymbol{\mu}\|_2^2 > C \frac{kp}{n}$, for some $C > 1$. In this case, we get that Eqn. (61) is lower bounded by $\frac{1}{c}\left(\left(1 - \frac{C_3}{\sqrt{n}} - \frac{C_4 n}{p}\right) \|\boldsymbol{\mu}\|_2 - C_5\sqrt{k}\right)^2$, and we have

$$\left(\left(1 - \frac{C_3}{\sqrt{n}} - \frac{C_4 n}{p}\right) \|\boldsymbol{\mu}\|_2 - C_5\sqrt{k}\right)^2 > \|\boldsymbol{\mu}\|_2^2 - 2\|\boldsymbol{\mu}\|_2^2 \frac{C_3}{\sqrt{n}} - 2\|\boldsymbol{\mu}\|_2^2 \frac{C_4 n}{p} - 2C_5\sqrt{k}\|\boldsymbol{\mu}\|_2$$

$$> \left(1 - \frac{2C_3}{\sqrt{n}}\right) \frac{kp}{n} - 2\|\boldsymbol{\mu}\|_2^2 \frac{C_4 n}{p} - 2C_5\sqrt{k}\|\boldsymbol{\mu}\|_2. \tag{63}$$

Then Eqn. (62) gives

$$(63) > \left(1 - \frac{2C_3}{\sqrt{n}}\right) \frac{kp}{n} - \left(\frac{p}{k^{1.5}n^{1.5}}\right)^2 \frac{C_6 n}{p} - \frac{C_7\sqrt{k}p}{k^{1.5}n^{1.5}}$$

$$= \frac{kp}{n}\left(1 - \frac{2C_3}{\sqrt{n}} - \frac{C_6}{k^4 n} - \frac{C_7}{k^2\sqrt{n}}\right), \tag{64}$$

which goes to $+\infty$ as $\left(\frac{p}{n}\right) \to \infty$.

Next, we consider the case $\|\boldsymbol{\mu}\|_2^2 \le \frac{kp}{n}$. Moreover, we assume that $\|\boldsymbol{\mu}\|_2^4 = C_2\left(\frac{p}{n}\right)^\alpha$, for $\alpha > 1$. Then, Eqn. (61) is lower bounded by $\frac{n}{ckp}\|\boldsymbol{\mu}\|_2^4 \left(\left(1 - \frac{C_3}{\sqrt{n}} - \frac{C_4 n}{p}\right) - \frac{C_5\sqrt{k}}{\|\boldsymbol{\mu}\|_2}\right)^2$, and we get

$$\frac{n}{kp}\|\boldsymbol{\mu}\|_2^4\left(\left(1 - \frac{C_3}{\sqrt{n}} - \frac{C_4 n}{p}\right) - \frac{C_5\sqrt{k}}{\|\boldsymbol{\mu}\|_2}\right)^2 > \left(1 - \frac{2C_3}{\sqrt{n}}\right)\frac{n}{kp}\|\boldsymbol{\mu}\|_2^4 - \frac{C_6 n^2}{kp^2}\|\boldsymbol{\mu}\|_2^4 - \frac{C_7 n}{\sqrt{kp}}\|\boldsymbol{\mu}\|_2^3$$

$$\ge \left(1 - \frac{2C_3}{\sqrt{n}}\right)\frac{1}{k}\left(\frac{p}{n}\right)^{\alpha-1} - \frac{C_6}{k}\left(\frac{p}{n}\right)^{\alpha-2} - \frac{C_7}{\sqrt{k}}\left(\frac{p}{n}\right)^{0.75\alpha-1}, \tag{65}$$

where the last inequality uses Equations (62) and condition $\|\boldsymbol{\mu}\|_2^2 \le \frac{kp}{n}$. Consequently, the RHS of Eqn. (65) will go to $+\infty$ as $\left(\frac{p}{n}\right) \to \infty$, provided that $\alpha > 1$. Overall, it suffices to have

$$p > \max\left\{C_1 k^3 n \log(kn) + n - 1, C_2 k^{1.5} n^{1.5}\|\boldsymbol{\mu}\|_2, \frac{n\|\boldsymbol{\mu}\|_2^2}{k}\right\},$$

$$\text{and } \|\boldsymbol{\mu}\|_2^4 \ge C_8\left(\frac{p}{n}\right)^\alpha, \quad \text{for } \alpha \in (1, 2].$$

All of these inequalities hold provided that $\|\boldsymbol{\mu}\|_2 = \Theta(p^\beta)$ for $\beta \in (1/4, 1/2]$ for finite $k$ and $n$. This completes the proof. $\qquad\square$

## E.3 Error analysis for MLM

In this section, we present partial results on error analysis of the MNI classifier when data is generated by the MLM. Importantly, for this case we consider more general anisotropic structure in the covariance matrix $\boldsymbol{\Sigma}$, in accordance with typical benign overfitting analyses [BLLT20, MNS+20]. The analysis in this section builds non-trivially on analysis that was done for the binary case [MNS+20]. We start by carrying over the assumptions from that analysis, starting with the $s$-sparse assumption.

**Assumption 3** ($s$-sparse class means [MNS+20]). *We assume that all of the class means $\boldsymbol{\mu}_c, c \in [k]$ are $s$-sparse in the basis given by the eigenvectors of $\boldsymbol{\Sigma}$. In other words, we have*

$$\boldsymbol{V}^{-1}\boldsymbol{\mu}_{c,j} = 0 \text{ if } j > s.$$

In addition to [MNS+20], this $s$-sparse assumption is also made in corresponding works on regression [HMRT19, TB20] and shown to be necessary for consistency of MSE of the minimum-$\ell_2$-norm interpolation arising from bias. Next, we make a special assumption of bi-level structure in the covariance matrix for ease of statements of results, just as in [MNS+20].

**Assumption 4** (Bi-level ensemble [MNS$^+$20])**.** *We assume that the eigenvalues of the covariance matrix, given by $\boldsymbol{\lambda}$, have a **bilevel** structure. In particular, our bi-level ensemble is parameterized by $(n, m, q, r)$ where $m > 1$, $0 \leq r < 1$ and $0 < q < (p - r)$. We set parameters $p := n^m$, $s := n^r$ and $a = n^{-q}$. Then, the eigenvalues of the covariance matrix are given by*

$$\lambda_j = \begin{cases} \lambda_H := \frac{ap}{s}, \ 1 \leq j \leq s \\ \lambda_L := \frac{(1-a)p}{p-s}, \ \text{otherwise.} \end{cases}$$

*We will fix $(m, q, r)$ and study the classification error as a function of $n$. While the bi-level ensemble structure is not in principle needed for complete statements of results, it admits particularly clean characterizations of classification error rates as well as easily interpretable conditions for consistency. See [MNS$^+$20] for additional context on the bi-level ensemble and examples of its manifestation in high-dimensional machine learning models.*

Finally, we imbue the above assumptions with an equal energy and orthogonality assumption, as in the GMM case. These assumptions are specific to the multiclass task, and effective subsume Assumption 3.

**Assumption 5** (Equal energy and orthogonality)**.** *We assume that the class means are equal energy, i.e. $\|\boldsymbol{\mu}\|_2 = 1/\sqrt{\lambda_H}$ for all $c \in [k]$, and are orthogonal, i.e. $\boldsymbol{\mu}_i^\top \boldsymbol{\mu}_j = 0$ for all $i \neq j \in [k]$. Together with Assumptions 3 and 4, a simple coordinate transformation gives us*

$$\boldsymbol{\mu}_c = \frac{1}{\sqrt{\lambda_H}} e_{j_c} \ \text{for some } j_c \in [s], \ j_c \neq j_{c'} \ \text{for all } c \neq c' \in [k], \ \text{and}$$

$$\boldsymbol{\Sigma} = \boldsymbol{\Lambda}$$

*without loss of generality. The normalization by the factor $\frac{1}{\sqrt{\lambda_H}}$ is done to ensure that the signal strength is equal to 1, i.e. $\mathbb{E}[(\mathbf{x}^\top \boldsymbol{\mu}_c)^2] = 1$ for all $c \in [k]$.*

Under these assumptions, we state our main result for the total classification error of MLM. Our error bounds will be on the *excess* risk over and above the Bayes error rate incurred by the optimal classifier $\{\widehat{\mathbf{w}}_c = \boldsymbol{\mu}_c\}_{c \in [k]}$, which we denote by $\mathbb{P}_{e, \text{Bayes}}$.

**Theorem 6.** *Under Assumptions 4 with $q < 1 - r$ and 5, there are universal constants $U, L_1, L_2$ such that the total excess classification error of the MNI under the MLM model is given by*

$$\mathbb{P}_e - \mathbb{P}_{e, \text{Bayes}} \leq k^2 \left( \frac{1}{2} - \frac{1}{\pi} \tan^{-1}(\text{SNR}(n)) \right), \ \text{where}$$

$$\text{SNR}(n) \geq L_1 \sqrt{\log n} \cdot n^{\frac{\min\{(m-1),(1-r)\}}{2}}, \ 0 < q < (1 - r).$$

Keeping $k$ constant with respect to $n$, Theorem 6 implies benign overfitting in multiclass classification, i.e. $\mathbb{P}_e - \mathbb{P}_{e, \text{Bayes}} \to 0$ as $n \to \infty$, if $q < (1 - r)$. Whether we can obtain consistency for the larger regime $q < (1 - r) + \frac{(m-1)}{2}$, as in the binary case [MNS$^+$20], remains an open question: as we will see in the proof, multiclass analysis introduces several new terms that are more difficult to deal with than in the binary case.

We set up notation for important quantities in the analysis. Note that Assumption 5 directly implies that $\mu_{c, j_c} = 1$ for all $c \in [k]$. For each class $c \in [k]$, we define the *survival* and *contamination* terms as below:

$$\text{SU}_c(n) := \sqrt{\lambda_H} \widehat{w}_{c, j_c} \tag{66a}$$

$$\text{CN}_c(n) := \sqrt{\sum_{j \neq j_c} \lambda_j \widehat{w}_{c, j}^2} \tag{66b}$$

Intuitively, we would like $\text{SU}_c(n) \to 1$ and $\text{CN}_c(n) \to 0$; note that this would be exactly the case if $\widehat{\mathbf{w}}_c = \boldsymbol{\mu}_c$ for all $c \in [k]$. We state and prove our main lemma that characterizes the classification error in MLM as a function of survival and contamination.

**Lemma 12.** *The excess classification risk is bounded by*

$$\mathbb{P}_e - \mathbb{P}_{e, \text{Bayes}} \leq \sum_{c_1 < c_2} \left( \frac{1}{2} - \frac{1}{\pi} \tan^{-1} \left( \frac{\text{SU}_{c_1}(n) + \text{SU}_{c_2}(n) - \text{CN}_{c_1}(n) - \text{CN}_{c_2}(n)}{2(|\text{SU}_{c_1}(n) - \text{SU}_{c_2}(n)| + \text{CN}_{c_1}(n) + \text{CN}_{c_2}(n))} \right) \right) \tag{67}$$

*Proof.* We consider a fixed $\mathbf{x}$, and (following the notation in [TOS20]) the $k$-dimensional vectors

$$\mathbf{g} := \begin{bmatrix} \mathbf{x}^\top \widehat{\mathbf{w}}_1 & \mathbf{x}^\top \widehat{\mathbf{w}}_2 & \dots & \mathbf{x}^\top \widehat{\mathbf{w}}_k \end{bmatrix}$$

$$\mathbf{h} := \begin{bmatrix} \mathbf{x}^\top \boldsymbol{\mu}_1 & \mathbf{x}^\top \boldsymbol{\mu}_2 & \dots & \mathbf{x}^\top \boldsymbol{\mu}_k \end{bmatrix}$$

Further, we define the *multinomial logit* variable

$$Y(\mathbf{h}) = j \text{ w.p. } \frac{\exp\{h_j\}}{\sum_{m=1}^k \exp\{h_m\}}.$$

Recall that $\mathbb{P}_e = \mathbb{P}\left(\arg\max(\mathbf{g}) \neq Y(\mathbf{h})\right)$, where the probability is taken both over the fresh test sample $\mathbf{x}$ and the randomness in the multinomial logit variable. We note that for there to be a classification error conditioned on $\mathbf{x}$, *at least* one of the following two events needs to hold: a) $\arg\max(\mathbf{g}) \neq \arg\max(\mathbf{h})$, or b) $Y(\mathbf{h}) \neq \arg\max(\mathbf{h})$. To see this, note that if neither a) nor b) held, we would have $\arg\max(\mathbf{g}) = Y(\mathbf{h})$ and we would not have a classification error conditional on the covariate being $\mathbf{x}$. Thus, applying a union bound gives us

$$\mathbb{P}_e \leq \mathbb{P}_{e,0} + \mathbb{P}_{e,\text{Bayes}} \text{ where}$$

$$\mathbb{P}_{e,0} := \mathbb{P}\left(\arg\max(\mathbf{g}) \neq \arg\max(\mathbf{h})\right) \text{ and}$$

$$\mathbb{P}_{e,\text{Bayes}} := \mathbb{P}\left(\arg\max(\mathbf{h}) \neq Y(\mathbf{h})\right).$$

Thus, it suffices to provide an upper bound on $\mathbb{P}_{e,0}$ as defined. We note that for there to be an error of the form $\arg\max(\mathbf{g}) \neq \arg\max(\mathbf{h})$, there *needs* to exist indices $c_1, c_2 \in [k]$ (whose choice can depend on $\mathbf{x}$) such that $\mathbf{x}^\top \boldsymbol{\mu}_{c_1} \geq \mathbf{x}^\top \boldsymbol{\mu}_{c_2}$ but $\mathbf{x}^\top \widehat{\mathbf{w}}_{c_1} < \mathbf{x}^\top \widehat{\mathbf{w}}_{c_2}$. In other words, we have

$$\mathbb{P}_{e,0} \leq \mathbb{P}\left(\mathbf{x}^\top \boldsymbol{\mu}_{c_1} \geq \mathbf{x}^\top \boldsymbol{\mu}_{c_2} \text{ and } \mathbf{x}^\top \widehat{\mathbf{w}}_{c_1} < \mathbf{x}^\top \widehat{\mathbf{w}}_{c_2} \text{ for some } c_1 \neq c_2\right)$$

$$\leq \sum_{c_1 \neq c_2} \mathbb{P}\left(\mathbf{x}^\top \boldsymbol{\mu}_{c_1} \geq \mathbf{x}^\top \boldsymbol{\mu}_{c_2} \text{ and } \mathbf{x}^\top \widehat{\mathbf{w}}_{c_1} < \mathbf{x}^\top \widehat{\mathbf{w}}_{c_2}\right)$$

$$= \sum_{c_1 < c_2} \mathbb{P}\left(\mathbf{x}^\top \boldsymbol{\Delta}_{c_1,c_2} \cdot \mathbf{x}^\top \widehat{\boldsymbol{\Delta}}_{c_1,c_2} < 0\right)$$

where we define

$$\boldsymbol{\Delta}_{c_1,c_2} := \boldsymbol{\mu}_{c_1} - \boldsymbol{\mu}_{c_2} \text{ and}$$

$$\widehat{\boldsymbol{\Delta}}_{c_1,c_2} := \widehat{\mathbf{w}}_{c_1} - \widehat{\mathbf{w}}_{c_2}.$$

Noting that $\mathbf{x} \sim \mathcal{N}(\mathbf{0}, \boldsymbol{\Sigma})$, setting $\boldsymbol{E}_{c_1,c_2} := \boldsymbol{\Sigma}^{1/2}\boldsymbol{\Delta}_{c_1,c_2}$, $\widehat{\boldsymbol{E}}_{c_1,c_2} := \boldsymbol{\Sigma}^{1/2}\widehat{\boldsymbol{\Delta}}_{c_1,c_2}$, as well as using rotation invariance of the Gaussian distribution and Gaussian decomposition yields:

$$\mathbb{P}\left(\mathbf{x}^\top \boldsymbol{\Delta}_{c_1,c_2} \cdot \mathbf{x}^\top \widehat{\boldsymbol{\Delta}}_{c_1,c_2} < 0\right) \tag{68}$$

$$= \mathbb{P}_{\mathbf{g}\sim\mathcal{N}(\mathbf{0},\mathbf{I})}\left(\mathbf{g}^\top \boldsymbol{E}_{c_1,c_2} \cdot \mathbf{g}^\top \widehat{\boldsymbol{E}}_{c_1,c_2} < 0\right)$$

$$= \mathbb{P}_{\substack{G\sim\mathcal{N}(0,1)\\H\sim\mathcal{N}(0,1)}}\left(\|\boldsymbol{E}_{c_1,c_2}\|_2 G \cdot \left(\mathsf{SU}(\widehat{\boldsymbol{\Delta}}_{c_1,c_2}, \boldsymbol{\Delta}_{c_1,c_2})G + \mathsf{CN}(\widehat{\boldsymbol{\Delta}}_{c_1,c_2}, \boldsymbol{\Delta}_{c_1,c_2})H\right) < 0\right)$$

$$= \mathbb{P}_{\substack{G\sim\mathcal{N}(0,1)\\H\sim\mathcal{N}(0,1)}}\left(\left(\mathsf{SU}(\widehat{\boldsymbol{\Delta}}_{c_1,c_2}, \boldsymbol{\Delta}_{c_1,c_2})G^2 + \mathsf{CN}(\widehat{\boldsymbol{\Delta}}_{c_1,c_2}, \boldsymbol{\Delta}_{c_1,c_2})HG\right) < 0\right)$$

$$= \frac{1}{2} - \frac{1}{\pi}\tan^{-1}\left(\frac{\mathsf{SU}(\widehat{\boldsymbol{\Delta}}_{c_1,c_2}, \boldsymbol{\Delta}_{c_1,c_2})}{\mathsf{CN}(\widehat{\boldsymbol{\Delta}}_{c_1,c_2}, \boldsymbol{\Delta}_{c_1,c_2})}\right), \quad \text{where} \tag{69}$$

$$\mathsf{SU}(\widehat{\boldsymbol{\Delta}}_{c_1,c_2}, \boldsymbol{\Delta}_{c_1,c_2}) := \frac{\widehat{\boldsymbol{E}}_{c_1,c_2}^T \boldsymbol{E}_{c_1,c_2}}{\|\boldsymbol{E}_{c_1,c_2}\|_2} = \frac{\widehat{\boldsymbol{\Delta}}^\top \boldsymbol{\Sigma}\boldsymbol{\Delta}}{\|\boldsymbol{\Sigma}^{1/2}\boldsymbol{\Delta}\|_2} \text{ and}$$

$$\mathsf{CN}(\widehat{\boldsymbol{\Delta}}_{c_1,c_2}, \boldsymbol{\Delta}_{c_1,c_2}) := \sqrt{\|\widehat{\boldsymbol{E}}_{c_1,c_2}\|_2^2 - \frac{\left(\widehat{\boldsymbol{E}}_{c_1,c_2}^T \boldsymbol{E}_{c_1,c_2}\right)^2}{\|\boldsymbol{E}_{c_1,c_2}\|_2^2}}$$

$$= \sqrt{\left(\widehat{\boldsymbol{\Delta}} - \frac{\widehat{\boldsymbol{\Delta}}_{c_1,c_2}^\top \boldsymbol{\Sigma}\boldsymbol{\Delta}_{c_1,c_2}}{\|\boldsymbol{\Sigma}^{1/2}\boldsymbol{\Delta}_{c_1,c_2}\|_2^2} \boldsymbol{\Delta}_{c_1,c_2}\right)^\top \boldsymbol{\Sigma}\left(\widehat{\boldsymbol{\Delta}}_{c_1,c_2} - \frac{\widehat{\boldsymbol{\Delta}}_{c_1,c_2}^\top \boldsymbol{\Sigma}\boldsymbol{\Delta}_{c_1,c_2}}{\|\boldsymbol{\Sigma}^{1/2}\boldsymbol{\Delta}_{c_1,c_2}\|_2^2} \boldsymbol{\Delta}_{c_1,c_2}\right)}$$

denote the generalized survival and contamination terms respectively. For the equality in Equation (69), we used the fact that the ratio $H/G$ of two independent standard normals follows the standard Cauchy distribution as in the proof of Proposition 1 in [MNS$^+$20].

It remains to expand the terms $\mathsf{SU}(\widehat{\boldsymbol{\Delta}}_{c_1,c_2}, \boldsymbol{\Delta}_{c_1,c_2})$ and $\mathsf{CN}(\widehat{\boldsymbol{\Delta}}_{c_1,c_2}, \boldsymbol{\Delta}_{c_1,c_2})$. First, we observe that

$$
\begin{aligned}
\mathsf{SU}(\widehat{\boldsymbol{\Delta}}_{c_1,c_2}, \boldsymbol{\Delta}_{c_1,c_2}) &= \frac{(\widehat{\mathbf{w}}_{c_1} - \widehat{\mathbf{w}}_{c_2})^\top \boldsymbol{\Sigma}(\boldsymbol{\mu}_{c_1} - \boldsymbol{\mu}_{c_2})}{\|\boldsymbol{\Sigma}^{1/2}(\boldsymbol{\mu}_{c_1} - \boldsymbol{\mu}_{c_2})\|_2} \\
&= \frac{(\widehat{\mathbf{w}}_{c_1} - \widehat{\mathbf{w}}_{c_2})^\top \boldsymbol{\Sigma}(\boldsymbol{\mu}_{c_1} - \boldsymbol{\mu}_{c_2})}{\sqrt{2}} \\
&= \sqrt{\lambda_H}(\widehat{w}_{c_1,j_{c_1}} + \widehat{w}_{c_2,j_{c_2}} - \widehat{w}_{c_1,j_{c_2}} - \widehat{w}_{c_2,j_{c_1}}) \\
&= \mathsf{SU}_{c_1}(n) + \mathsf{SU}_{c_2}(n) - \sqrt{\lambda_H} \cdot \widehat{w}_{c_1,j_{c_2}} - \sqrt{\lambda_H} \cdot \widehat{w}_{c_2,j_{c_1}} \\
&\geq \mathsf{SU}_{c_1}(n) + \mathsf{SU}_{c_2}(n) - \mathsf{CN}_{c_1}(n) - \mathsf{CN}_{c_2}(n),
\end{aligned}
$$

where the last inequality follows because we have $\mathsf{CN}_{c_1}(n) := \sqrt{\sum_{j \neq j_{c_1}} \lambda_j \widehat{w}_j^2} \geq |\sqrt{\lambda_H}\widehat{w}_{j_{c_2}}|$. Similar reasoning holds for the term $\mathsf{CN}_{c_2}(n)$. Note that we have critically used the orthogonality assumption, which implies that $j_{c_1} \neq j_{c_2}$.

Second, we analyze the contamination term $\mathsf{CN}(\widehat{\boldsymbol{\Delta}}_{c_1,c_2}, \boldsymbol{\Delta}_{c_1,c_2})$. We denote $\widehat{\boldsymbol{\Delta}} := \widehat{\boldsymbol{\Delta}}_{c_1,c_2}$ and $\boldsymbol{\Delta} := \boldsymbol{\Delta}_{c_1,c_2}$ for shorthand. We have

$$
\mathsf{CN}(\widehat{\boldsymbol{\Delta}}_{c_1,c_2}, \boldsymbol{\Delta}_{c_1,c_2}) = \sqrt{\mathbb{E}[B(\mathbf{x})^2]} \quad \text{where} \quad B(\mathbf{x}) := \left( \widehat{\boldsymbol{\Delta}} - \frac{\widehat{\boldsymbol{\Delta}}^\top \boldsymbol{\Sigma} \boldsymbol{\Delta}}{\|\boldsymbol{\Sigma}^{1/2}\boldsymbol{\Delta}\|_2^2} \boldsymbol{\Delta} \right)^\top \mathbf{x}.
$$

We characterize the orthogonal term $\widehat{\boldsymbol{\Delta}} - \frac{\widehat{\boldsymbol{\Delta}}^\top \boldsymbol{\Sigma} \boldsymbol{\Delta}}{\|\boldsymbol{\Sigma}^{1/2}\boldsymbol{\Delta}\|_2^2} \boldsymbol{\Delta}$. By simple algebra, we get

$$
\left( \widehat{\boldsymbol{\Delta}} - \frac{\widehat{\boldsymbol{\Delta}}^\top \boldsymbol{\Sigma} \boldsymbol{\Delta}}{\|\boldsymbol{\Sigma}^{1/2}\boldsymbol{\Delta}\|_2^2} \boldsymbol{\Delta} \right)_j = \begin{cases} \frac{\widehat{w}_{c_1,j_{c_1}} - \widehat{w}_{c_2,j_{c_2}} + \widehat{w}_{c_1,j_{c_2}} - \widehat{w}_{c_2,j_{c_1}}}{2} & \text{if } j = j_{c_1} \\ \frac{\widehat{w}_{c_1,j_{c_1}} - \widehat{w}_{c_2,j_{c_2}} + \widehat{w}_{c_1,j_{c_2}} - \widehat{w}_{c_2,j_{c_1}}}{2} & \text{if } j = j_{c_2} \\ \widehat{w}_{c_1,j} - \widehat{w}_{c_2,j} & \text{otherwise.} \end{cases}
$$

This gives us

$$
\begin{aligned}
\boldsymbol{\Sigma}^{1/2} \left( \widehat{\boldsymbol{\Delta}} - \frac{\widehat{\boldsymbol{\Delta}}^\top \boldsymbol{\Sigma} \boldsymbol{\Delta}}{\|\boldsymbol{\Sigma}^{1/2}\boldsymbol{\Delta}\|_2^2} \boldsymbol{\Delta} \right) = {}& \frac{\sqrt{\lambda_H}\widehat{w}_{c_1,j_{c_1}} - \sqrt{\lambda_H}\widehat{w}_{c_2,j_{c_2}}}{2}(\hat{\boldsymbol{e}}_{j_{c_1}} + \hat{\boldsymbol{e}}_{j_{c_2}}) \\
& + \boldsymbol{\Sigma}^{1/2} \left( \sum_{j \neq j_{c_1}, j_{c_2}} \widehat{w}_{c_1,j}\hat{\boldsymbol{e}}_j \right) - \boldsymbol{\Sigma}^{1/2} \left( \sum_{j \neq j_{c_1}, j_{c_2}} \widehat{w}_{c_2,j}\hat{\boldsymbol{e}}_j \right) \\
& + \frac{\sqrt{\lambda_H}(\widehat{w}_{c_1,j_{c_2}} - \widehat{w}_{c_2,j_{c_1}})}{2}(\hat{\boldsymbol{e}}_{j_{c_1}} + \hat{\boldsymbol{e}}_{j_{c_2}}).
\end{aligned}
$$

Since $\sqrt{\mathbb{E}[B(\mathbf{x})^2]} = \|\boldsymbol{\Sigma}^{1/2}\left( \widehat{\boldsymbol{\Delta}} - \frac{\widehat{\boldsymbol{\Delta}}^\top \boldsymbol{\Sigma} \boldsymbol{\Delta}}{\|\boldsymbol{\Sigma}^{1/2}\boldsymbol{\Delta}\|_2^2} \boldsymbol{\Delta} \right)\|_2$, applying the triangle inequality and recalling the survival and contamination terms in Equations (66), then gives us

$$
\begin{aligned}
\mathsf{CN}(\widehat{\boldsymbol{\Delta}}_{c_1,c_2}, \boldsymbol{\Delta}_{c_1,c_2}) &\leq \frac{1}{\sqrt{2}}(\mathsf{SU}_{c_1}(n) - \mathsf{SU}_{c_2}(n)) + \left( 1 + \frac{1}{\sqrt{2}} \right)(\mathsf{CN}_{c_1}(n) + \mathsf{CN}_{c_2}(n)) \\
&\leq 2(\mathsf{SU}_{c_1}(n) - \mathsf{SU}_{c_2}(n) + \mathsf{CN}_{c_1}(n) + \mathsf{CN}_{c_2}(n)).
\end{aligned}
$$

This completes the proof. $\qquad\square$

Next, we provide characterizations of $\mathsf{SU}_c(n)$ and $\mathsf{CN}_c(n)$ in the multiclass case. These constitute extensions from Lemmas 11 and 13 [MNS$^+$20] to deal with two new aspects of the MLM: the multiclass setting, and label noise generated by the logistic model (note that [MNS$^+$20] considered only constant label noise whose magnitude does not depend on the covariate). We start with the characterization of survivals.

**Lemma 13** (extension of Lemma 11, [MNS$^+$20]). *There exist universal positive constants* $L_1, L_2, U_1, U_2$ *such that*

$$\mathsf{SU}^L(n) \leq \mathsf{SU}_c(n) \leq \mathsf{SU}^U(n), \ \ where$$

$$\mathsf{SU}^L(n) := \begin{cases} c_k(1 + L_1 n^{q-(1-r)})^{-1}, \ 0 < q < 1 - r \\ c_k L_2 n^{(1-r)-q}, \ q > 1 - r. \end{cases}$$

$$\mathsf{SU}^U(n) := \begin{cases} c_k(1 + U_1 n^{q-(1-r)})^{-1}, \ 0 < q < 1 - r \\ c_k U_2 n^{(1-r)-q}, \ q > 1 - r. \end{cases}$$

*Above,* $c_k > 0$ *is a fixed strictly positive constant that depends on* $k$ *but not on* $n$.

Next, we provide an upper-bound characterization of contamination.

**Lemma 14** (extension of Lemma 13, [MNS$^+$20]). *There exist universal positive constants* $U_3, U_4$ *such that*

$$\mathsf{CN}_c(n) \leq \mathsf{CN}^U(n) := \begin{cases} U_3 \sqrt{\log n} \cdot n^{-\frac{\min\{m-1, 1-r\}}{2}}, \ 0 < q < 1 - r \\ U_4 \sqrt{\log n} \cdot n^{-\frac{\min\{m-1, 2q+r-q\}}{2}}, \ q > 1 - r. \end{cases}$$

Taking Lemmas 13 and 14 as true for the moment, we get

$$\mathbb{P}_{e,0} \leq k^2 \left( \frac{1}{2} - \tan^{-1}\left( \frac{2\mathsf{SU}^L(n) - 2\mathsf{CN}^U(n)}{2|\mathsf{SU}^U(n) - \mathsf{SU}^L(n)| + 2\mathsf{CN}^U(n)} \right) \right).$$

The new term of particular interest is $|\mathsf{SU}^U(n) - \mathsf{SU}^L(n)|$. When $0 < q < 1 - r$, we have

$$|\mathsf{SU}^U(n) - \mathsf{SU}^L(n)| = \frac{c_k}{1 + L_1 n^{q-(1-r)}} - \frac{c_k}{1 + U_1 n^{q-(1-r)}}$$
$$\leq c_k(U_1 - L_1)n^{q-(1-r)}.$$

Then, the argument within $\tan^{-1}(\cdot)$ is lower bounded by

$$\frac{2c(1 + L_1 n^{q-(1-r)})^{-1} - 2U_3\sqrt{\log n}n^{-\frac{\min\{m-1,1-r\}}{2}}}{c(U_1 - L_1)n^{q-(1-r)} + 2U_3\sqrt{\log n}n^{-\frac{\min\{m-1,1-r\}}{2}}}$$
$$\geq L_3 n^{\frac{\min\{m-1,1-r\}}{2}},$$

where $L_3 > 0$ is some universal positive constant. This completes the proof for the case $0 < q < 1-r$. Further, this term goes to $\infty$ as $n \to \infty$, implying that $\mathbb{P}_e \to 0$ as $n \to \infty$ in this case.

It only remains to prove Lemmas 13 and 14, which we do below.

### E.3.1 Proof of Lemma 13

The proof follows similarly to the proof of Theorem 4 in Appendix D.3 and Lemma 11 in Appendix E of [MNS$^+$20], with two important extensions: one, to the multiclass model, and two, considering the logistic model for label noise. First, an identical series of steps to the proof of Theorem 4 gives us

$$\mathsf{SU}_c(n) = \frac{\lambda_{j_c} \cdot \mathbf{u}_{j_c}^\top \mathbf{A}_{-j_c}^{-1} \mathbf{z}_c}{1 + \lambda_{j_c} \cdot \mathbf{u}_{j_c}^\top \mathbf{A}_{-j_c}^{-1} \mathbf{u}_{j_c}}.$$

As with the proof of Theorem 4 in [MNS$^+$20], we can bound the quadratic forms around their expectations by using the Hanson-Wright inequality [RV$^+$13], thereby getting

$$\mathsf{SU}_c(n) \leq c_k \cdot \frac{\lambda_{j_c}\left( \frac{(n-s)}{c\widetilde{\lambda}_{s+1}r_s(\boldsymbol{\Sigma}_{-j_c})} - \frac{c_3 n^{3/4}}{\lambda_{s+1}r_s(\boldsymbol{\Sigma})} \right)}{1 + \lambda_{j_c}\left( \frac{cn}{\widetilde{\lambda}_{s+1}r_s(\boldsymbol{\Sigma}_{-j_c})} + \frac{c_4 n^{3/4}}{\lambda_{s+1}r_s(\boldsymbol{\Sigma})} \right)} \ \ \text{and}$$

$$\mathsf{SU}_c(n) \geq c_k \cdot \frac{\lambda_{j_c}\left( \frac{cn}{\widetilde{\lambda}_{s+1}r_s(\boldsymbol{\Sigma}_{-j_c})} + \frac{c_3 n^{3/4}}{\lambda_{s+1}r_s(\boldsymbol{\Sigma})} \right)}{1 + \lambda_{j_c}\left( \frac{(n-s)}{c\widetilde{\lambda}_{s+1}r_s(\boldsymbol{\Sigma}_{-j_c})} - \frac{c_4 n^{3/4}}{\lambda_{s+1}r_s(\boldsymbol{\Sigma})} \right)},$$

where we define $c_k := \mathbb{E}[u_{j_c,1}z_{c,1}]$. Plugging this into an identical argument as in the proof of Lemma 11 in Appendix E of [MNS$^+$20] gives us the statement of Lemma 13. It remains to show that $c_k$ is strictly positive (clearly, it will not depend on $n$ or $p$). To do this, we critically utilize the orthogonality and equal-weight Assumption 5 as well as the details of the MLM. We combine these to get

$$\mathbb{P}\left(z_{j_c,1} = \frac{k-1}{k}\Big|\{u_{j,1}\}_{j=1}^p\right) = \frac{\exp(u_{j_c,1})}{\sum_{c'\in[k]}\exp(u_{j_{c'},1})},$$

where $\{u_{j',1}\}_{c'\in[k]}$ are IID standard Gaussian. Thus, we get

$$\mathbb{E}[u_{j_c,1}z_{c,1}] = \mathbb{E}\left[u_{j_c,1}\left(\frac{k-1}{k}\cdot\frac{\exp(u_{j_c,1})}{\sum_{c'\in[k]}\exp(u_{j_{c'},1})}\right) - \frac{1}{k}\left(1 - \frac{\exp(u_{j_c,1})}{\sum_{c'\in[k]}\exp(u_{j_{c'},1})}\right)\right]$$

$$= \mathbb{E}\left[u_{j_c,1}\left(\frac{\exp(u_{j_c,1})}{\sum_{c'\in[k]}\exp(u_{j_{c'},1})}\right)\right].$$

Now, we overload notation and write $U_c := u_{j_c,1}$ for each $c \in [k]$. We also write $\mathbf{U} := [U_1 \quad \ldots \quad U_k]$ as shorthand. Note that $U_c$ i.i.d. $\sim \mathcal{N}(0,1)$. By symmetry, we have

$$c_k = \mathbb{E}\left[\frac{U_1 e^{U_1}}{\sum_{c=1}^k e^{U_c}}\right]$$

$$= \frac{1}{k}\mathbb{E}\left[\mathbf{U}^\top g(\mathbf{U})\right]$$

$$= \mathbb{E}\left[U_1 \cdot g(\mathbf{U})\right]$$

where $g_i(\mathbf{U}) := \frac{e^{U_i}}{\sum_{i'=1}^k e^{U_i'}}$. Then, applying Stein's lemma, we get

$$\mathbb{E}\left[U_1 \cdot g(\mathbf{U})\right] = \sum_{i=1}^n \mathbb{E}[U_1 U_i] \cdot \mathbb{E}\left[\frac{\partial g}{\partial U_i}\right]$$

$$= \mathbb{E}\left[\frac{\partial g}{\partial U_1}\right]$$

$$= \mathbb{E}\left[\frac{\sum_{i\neq 1} e^{U_1+U_i}}{(\sum_{i=1}^k e^{U_i})^2}\right] > 0.$$

The last step follows because the argument inside the expectation can never take value 0 and is always non-negative. This completes the proof. $\square$

### E.3.2 Proof of Lemma 14

This proof is a simple extension of the argument in [MNS$^+$20, Theorem 5]. First, we recall that

$$\widetilde{\mathbf{w}}_c := \mathbf{X}(\mathbf{X}^T\mathbf{X})^{-1}\mathbf{z}_c.$$

As a direct consequence (following the proof of [MNS$^+$20, Lemma 6]), we get

$$\mathsf{CN}_c(n) = \sqrt{\mathbf{z}_c^\top \mathbf{C}\mathbf{z}_c}, \text{ where}$$

$$\mathbf{C} := (\mathbf{X}^T\mathbf{X})^{-1}\left(\sum_{j=1,j\neq j_c}^d \lambda_j^2 \mathbf{u}_j\mathbf{u}_j^\top\right)(\mathbf{X}^T\mathbf{X})^{-1}$$

where $\{\mathbf{u}_j\}_{j=1}^p$ denotes the rows of data matrix $\mathbf{X}$ normalized by $\sqrt{\lambda_j}$. We now define $\mathbf{A} := \mathbf{X}^T\mathbf{X}$ for shorthand, and $\mathbf{A}_{-j_c} := \sum_{j=1,j\neq j_c}^d \lambda_j \mathbf{u}_j\mathbf{u}_j^\top$ as the "leave-one-out" matrix. Note that these

matrices are functionals of the training data covariates, and so are common across all classes. Further, again following the proof of [MNS$^+$20, Lemma 6], we get

$$\mathsf{CN}_c(n) = \sqrt{\widetilde{\mathbf{z}}_c^\top \widetilde{\mathbf{C}} \widetilde{\mathbf{z}}_c}, \text{ where}$$

$$\widetilde{\mathbf{z}}_c := \mathbf{z}_c - \mathsf{SU}_c(n)\mathbf{u}_c \text{ and}$$

$$\widetilde{\mathbf{C}} := (\mathbf{A}_{-j_c})^{-1} \left( \sum_{j=1, j \neq j_c}^{d} \lambda_j^2 \mathbf{u}_j \mathbf{u}_j^\top \right) (\mathbf{A}_{-j_c})^{-1}$$

Now, an identical argument to the proof of [MNS$^+$20, Lemma 29] gives us

$$\widetilde{\mathbf{z}}_c^\top \widetilde{\mathbf{C}} \widetilde{\mathbf{z}}_c \leq 2\mathbf{z}_c^\top \widetilde{\mathbf{C}} \mathbf{z}_c + 2\mathsf{SU}_c(n)^2 \cdot \mathbf{u}_{j_c}^\top \widetilde{\mathbf{C}} \mathbf{u}_{j_c}$$

$$\leq 4\mathrm{Tr}(\widetilde{\mathbf{C}}) \left( 1 + \frac{1}{c} \right) \cdot \log n,$$

where $c$ is some universal positive constant. The last inequality follows as a critical consequence of three facts:

1. We have $\mathsf{SU}_c(n) \leq 1$ almost surely.

2. Noting that $\mathbf{u}_{j_c}$ is isotropic Gaussian, we have $\mathbf{u}_{j_c}^\top \widetilde{\mathbf{C}} \mathbf{u}_{j_c} \leq 2\mathrm{Tr}(\widetilde{\mathbf{C}}) \left( 1 + \frac{1}{c} \right) \cdot \log n$ with high probability from an application of the Hanson-Wright inequality [RV$^+$13].

3. Noting that the Hanson-Wright inequality also applies to sub-Gaussian random vectors with uncorrelated components [RV$^+$13], we similarly apply it to $\mathbf{z}_c$ by noting that $z_{c,j}^2 \leq 1$ and $\mathbb{E}[z_{c,j} z_{c,j'}] = 0$ for all $j \neq j'$). The $z_{c,j}$'s being uncorrelated is an important consequence of the orthogonality Assumption 5 and independence of training data.

After this, an analysis identical to that of the binary case (contained in Appendices D.4.2 and Lemma 13 of [MNS$^+$20]) completes the proof of the result. $\qquad \square$

# F  Recursive formulas for higher-order quadratic forms

We first show how quadratic forms involving the $j$-th order Gram matrix $\mathbf{A}_j^{-1}$ can be expressed using quadratic forms involving the $(j-1)$-th order Gram matrix $\mathbf{A}_{j-1}^{-1}$. For concreteness, we consider $j = 1$; identical expressions hold for any $j > 1$ with the only change being in the superscripts. Recall from Appendix C that we can write

$$\mathbf{A}_1 = \mathbf{A}_0 + \begin{bmatrix} \|\boldsymbol{\mu}\|_2 \mathbf{v}_1 & \mathbf{Q}^T \boldsymbol{\mu}_1 & \mathbf{v}_1 \end{bmatrix} \begin{bmatrix} \|\boldsymbol{\mu}\|_2 \mathbf{v}_1^T \\ \mathbf{v}_1^T \\ \boldsymbol{\mu}_1^T \mathbf{Q} \end{bmatrix} = \mathbf{Q}^T \mathbf{Q} + \begin{bmatrix} \|\boldsymbol{\mu}\|_2 \mathbf{v}_1 & \mathbf{d}_1 & \mathbf{v}_1 \end{bmatrix} \begin{bmatrix} \|\boldsymbol{\mu}\|_2 \mathbf{v}_1^T \\ \mathbf{v}_1^T \\ \mathbf{d}_1^T \end{bmatrix}.$$

The first step is to derive an expression for $\mathbf{A}_1^{-1}$. By the Woodbury identity [HJ12], we get

$$\mathbf{A}_1^{-1} = \mathbf{A}_0^{-1} - \mathbf{A}_0^{-1} \begin{bmatrix} \|\boldsymbol{\mu}\|_2 \mathbf{v}_1 & \mathbf{d}_1 & \mathbf{v}_1 \end{bmatrix} \begin{bmatrix} \mathbf{I} + \begin{bmatrix} \|\boldsymbol{\mu}\|_2 \mathbf{v}_1^T \\ \mathbf{v}_1^T \\ \mathbf{d}_1^T \end{bmatrix} \mathbf{A}_0^{-1} \begin{bmatrix} \|\boldsymbol{\mu}\|_2 \mathbf{v}_1 & \mathbf{d}_1 & \mathbf{v}_1 \end{bmatrix} \end{bmatrix}^{-1} \begin{bmatrix} \|\boldsymbol{\mu}\|_2 \mathbf{v}_1^T \\ \mathbf{v}_1^T \\ \mathbf{d}_1^T \end{bmatrix} \mathbf{A}_0^{-1}.$$
$$(70)$$

We first compute the inverse of the $3 \times 3$ matrix $\mathbf{B} := \begin{bmatrix} \mathbf{I} + \begin{bmatrix} \|\boldsymbol{\mu}\|_2 \mathbf{v}_1^T \\ \mathbf{v}_1^T \\ \mathbf{d}_1^T \end{bmatrix} \mathbf{A}_0^{-1} \begin{bmatrix} \|\boldsymbol{\mu}\|_2 \mathbf{v}_1 & \mathbf{d}_1 & \mathbf{v}_1 \end{bmatrix} \end{bmatrix}$.

Recalling our definitions of the terms $s_{mj}^{(c)}, h_{mj}^{(c)}$ and $t_{mj}^{(c)}$ in Eqn. (28) in Appendix C, we have:

$$\mathbf{B} = \begin{bmatrix} 1 + \|\boldsymbol{\mu}\|_2^2 s_{11}^{(0)} & \|\boldsymbol{\mu}\|_2 h_{11}^{(0)} & \|\boldsymbol{\mu}\|_2 s_{11}^{(0)} \\ \|\boldsymbol{\mu}\|_2 s_{11}^{(0)} & 1 + h_{11}^{(0)} & s_{11}^{(0)} \\ \|\boldsymbol{\mu}\|_2 h_{11}^{(0)} & t_{11}^{(0)} & 1 + h_{11}^{(0)} \end{bmatrix}.$$

Recalling $\mathbf{B}^{-1} = \frac{1}{\det_0}\text{adj}(\mathbf{B})$, where $\det_0$ is the determinant of $\mathbf{B}$ and $\text{adj}(\mathbf{B})$ is the adjoint of $\mathbf{B}$, simple algebra gives us

$$\det_0 = s_{11}^{(0)}(\|\boldsymbol{\mu}\|_2^2 - t_{11}^{(0)}) + (h_{11}^{(0)} + 1)^2,$$

and

$$\text{adj}(\mathbf{B}) = \begin{bmatrix} (h_{11}^{(0)} + 1)^2 - s_{11}^{(0)}t_{11}^{(0)} & \|\boldsymbol{\mu}\|_2(s_{11}^{(0)}t_{11}^{(0)} - h_{11}^{(0)} - h_{11}^{(0)^2}) & -\|\boldsymbol{\mu}\|_2 s_{11}^{(0)} \\ -\|\boldsymbol{\mu}\|_2 s_{11}^{(0)} & h_{11}^{(0)} + 1 + \|\boldsymbol{\mu}\|_2^2 s_{11}^{(0)} & -s_{11}^{(0)} \\ \|\boldsymbol{\mu}\|_2(s_{11}^{(0)}t_{11}^{(0)} - h_{11}^{(0)} - h_{11}^{(0)^2}) & \|\boldsymbol{\mu}\|_2^2 h_{11}^{(0)^2} - t_{11}^{(0)}(1 + \|\boldsymbol{\mu}\|_2^2 s_{11}^{(0)}) & h_{11}^{(0)} + 1 + \|\boldsymbol{\mu}\|_2^2 s_{11}^{(0)} \end{bmatrix}.$$

We will now use these expressions to derive expressions for the 1-order quadratic forms that are used in Appendix C.4.

### F.1   Expressions for 1-st order quadratic forms

We now show how quadratic forms of order 1 can be expressed as a function of quadratic forms of order 0. All of the expressions are derived as a consequence of plugging in the expression for $\mathbf{B}^{-1}$ together with elementary matrix algebra.

First, we have

$$s_{mk}^{(1)} = \mathbf{v}_m^T\mathbf{A}_1^{-1}\mathbf{v}_k = \mathbf{v}_m^T\mathbf{A}_0^{-1}\mathbf{v}_k - \begin{bmatrix} \|\boldsymbol{\mu}\|_2 s_{m1}^{(0)} & h_{m1}^{(0)} & s_{m1}^{(0)} \end{bmatrix} \frac{\text{adj}(\mathbf{B})}{\det_0} \begin{bmatrix} \|\boldsymbol{\mu}\|_2 s_{k1}^{(0)} \\ s_{k1}^{(0)} \\ h_{k1}^{(0)} \end{bmatrix}$$

$$= s_{mk}^{(0)} - \frac{1}{\det_0}(\star)_s^{(0)}, \tag{71}$$

where we define

$$(\star)_s^{(0)} := (\|\boldsymbol{\mu}\|_2^2 - t_{11}^{(0)})s_{1k}^{(0)}s_{1m}^{(0)} + s_{1m}^{(0)}h_{k1}^{(0)}h_{11}^{(0)} + s_{1k}^{(0)}h_{m1}^{(0)}h_{11}^{(0)} - s_{11}^{(0)}h_{k1}^{(0)}h_{m1}^{(0)} + s_{1m}^{(0)}h_{k1}^{(0)} + s_{1k}^{(0)}h_{m1}^{(0)}.$$

Thus, for the case $m = k$ we have

$$s_{kk}^{(1)} = \mathbf{v}_k^T\mathbf{A}_1^{-1}\mathbf{v}_k = \mathbf{v}_k^T\mathbf{A}_0^{-1}\mathbf{v}_k - \begin{bmatrix} \|\boldsymbol{\mu}\|_2 s_{k1}^{(0)} & h_{k1}^{(0)} & s_{k1}^{(0)} \end{bmatrix} \frac{\text{adj}(\mathbf{B})}{\det_0} \begin{bmatrix} \|\boldsymbol{\mu}\|_2 s_{k1}^{(0)} \\ s_{k1}^{(0)} \\ h_{k1}^{(0)} \end{bmatrix}$$

$$= s_{kk}^{(0)} - \frac{1}{\det_0}\left((\|\boldsymbol{\mu}\|_2^2 - t_{11}^{(0)})s_{1k}^{(0)^2} + 2s_{1k}^{(0)}h_{k1}^{(0)}h_{11}^{(0)} - s_{11}^{(0)}h_{k1}^{(0)^2} + 2s_{1k}^{(0)}h_{k1}^{(0)}\right). \tag{72}$$

Next, we have

$$h_{mk}^{(1)} = \mathbf{v}_m^T\mathbf{A}_1^{-1}\mathbf{d}_k = \mathbf{v}_m^T\mathbf{A}_0^{-1}\mathbf{d}_k - \begin{bmatrix} \|\boldsymbol{\mu}\|_2 s_{m1}^{(0)} & h_{m1}^{(0)} & s_{m1}^{(0)} \end{bmatrix} \frac{\text{adj}(\mathbf{B})}{\det_0} \begin{bmatrix} \|\boldsymbol{\mu}\|_2 h_{1k}^{(0)} \\ h_{1k}^{(0)} \\ t_{1k}^{(0)} \end{bmatrix}$$

$$= h_{mk}^{(0)} - \frac{1}{\det_0}(\star)_h^{(0)}, \tag{73}$$

where we define

$$(\star)_h^{(0)} = (\|\boldsymbol{\mu}\|_2^2 - t_{11}^{(0)})s_{1m}^{(0)}h_{1k}^{(0)} + h_{m1}^{(0)}h_{1k}^{(0)}h_{11}^{(0)} + h_{m1}^{(0)}h_{1k}^{(0)} + s_{1m}^{(0)}t_{k1}^{(0)} + s_{1m}^{(0)}t_{k1}^{(0)}h_{11}^{(0)} - s_{11}^{(0)}t_{k1}^{(0)}h_{m1}^{(0)}.$$

Next, we have

$$t_{km}^{(1)} = \mathbf{d}_k^T\mathbf{A}_1^{-1}\mathbf{d}_m = \mathbf{d}_k^T\mathbf{A}_0^{-1}\mathbf{d}_m - \begin{bmatrix} \|\boldsymbol{\mu}\|_2 h_{1k}^{(0)} & t_{1k}^{(0)} & h_{1k}^{(0)} \end{bmatrix} \frac{\text{adj}(\mathbf{B})}{\det_0} \begin{bmatrix} \|\boldsymbol{\mu}\|_2 h_{1m}^{(0)} \\ h_{1m}^{(0)} \\ t_{1m}^{(0)} \end{bmatrix}$$

$$= t_{km}^{(0)} - \frac{1}{\det_0}(\star)_t^{(0)}, \tag{74}$$

where we define

$$(\star)_t^{(0)} = (\|\boldsymbol{\mu}\|_2^2 - t_{11}^{(0)})h_{1m}^{(0)}h_{1k}^{(0)} + t_{m1}^{(0)}h_{1k}^{(0)}h_{11}^{(0)} + t_{k1}^{(0)}h_{1m}^{(0)}h_{11}^{(0)} + t_{1m}^{(0)}h_{1k}^{(0)} + t_{1k}^{(0)}h_{1m}^{(0)} - s_{11}^{(0)}t_{1m}^{(0)}t_{1k}^{(0)}.$$

Thus, for the case $m = k$ we have

$$t_{kk}^{(1)} = \mathbf{d}_k^T \mathbf{A}_1^{-1} \mathbf{d}_k = \mathbf{d}_k^T \mathbf{A}_0^{-1} \mathbf{d}_k - \begin{bmatrix} \|\boldsymbol{\mu}\|_2 h_{1k}^{(0)} & t_{1k}^{(0)} & h_{1k}^{(0)} \end{bmatrix} \frac{\mathrm{adj}(\mathbf{B})}{\mathrm{det}_0} \begin{bmatrix} \|\boldsymbol{\mu}\|_2 h_{1k}^{(0)} \\ h_{1k}^{(0)} \\ t_{1k}^{(0)} \end{bmatrix}$$
$$= t_{kk}^{(0)} - \frac{1}{\mathrm{det}_0}\left( (\|\boldsymbol{\mu}\|_2^2 - t_{11}^{(0)})h_{1k}^{(0)2} + 2t_{1k}^{(0)}h_{1k}^{(0)}h_{11}^{(0)} - s_{11}^{(0)}t_{1k}^{(0)2} + 2t_{1k}^{(0)}h_{1k}^{(0)} \right). \tag{75}$$

Next, we have

$$f_{ki}^{(1)} = \mathbf{d}_k^T \mathbf{A}_1^{-1} \mathbf{e}_i = \mathbf{d}_k^T \mathbf{A}_0^{-1} \mathbf{e}_i - \begin{bmatrix} \|\boldsymbol{\mu}\|_2 h_{1k}^{(0)} & t_{1k}^{(0)} & h_{1k}^{(0)} \end{bmatrix} \frac{\mathrm{adj}(\mathbf{B})}{\mathrm{det}_0} \begin{bmatrix} \|\boldsymbol{\mu}\|_2 g_{1i}^{(0)} \\ g_{1i}^{(0)} \\ f_{1i}^{(0)} \end{bmatrix}$$
$$= f_{ki}^{(0)} - \frac{1}{\mathrm{det}_0}(\star)_f^{(0)}, \tag{76}$$

where we define

$$(\star)_f^{(0)} = (\|\boldsymbol{\mu}\|_2^2 - t_{11}^{(0)})h_{1k}^{(0)}g_{1i}^{(0)} + t_{1k}^{(0)}g_{1i}^{(0)} + t_{1k}^{(0)}h_{11}^{(0)}g_{1i}^{(0)} + h_{1k}^{(0)}f_{1i}^{(0)} + h_{1k}^{(0)}h_{11}^{(0)}f_{1i}^{(0)} - s_{11}^{(0)}t_{1k}^{(0)}f_{1i}^{(0)}.$$

Finally, we have

$$g_{ji}^{(1)} = \mathbf{v}_j^T \mathbf{A}_1^{-1} \mathbf{e}_i = \mathbf{v}_j^T \mathbf{A}_0^{-1} \mathbf{e}_i - \begin{bmatrix} \|\boldsymbol{\mu}\|_2 s_{j1}^{(0)} & h_{j1}^{(0)} & s_{j1}^{(0)} \end{bmatrix} \frac{\mathrm{adj}(\mathbf{B})}{\mathrm{det}_0} \begin{bmatrix} \|\boldsymbol{\mu}\|_2 g_{1i}^{(0)} \\ g_{1i}^{(0)} \\ f_{1i}^{(0)} \end{bmatrix}$$
$$= g_{ji}^{(0)} - \frac{1}{\mathrm{det}_0}(\star)_{gj}^{(0)}, \tag{77}$$

where we define

$$(\star)_{gj}^{(0)} = (\|\boldsymbol{\mu}\|_2^2 - t_{11}^{(0)})s_{1j}^{(0)}g_{1i}^{(0)} + g_{1i}^{(0)}h_{11}^{(0)}h_{j1}^{(0)} + g_{1i}^{(0)}h_{j1}^{(0)} + s_{1j}^{(0)}f_{1i}^{(0)} + s_{1j}^{(0)}h_{11}^{(0)}f_{1i}^{(0)} - s_{11}^{(0)}h_{j1}^{(0)}f_{1i}^{(0)}.$$

$\square$

# G   One-vs-all SVM

In this section, we derive conditions under which the OvA solutions $\mathbf{w}_{\mathrm{OvA},c}$ interpolate, i.e, all data points are support vectors in Eqn. (4).

## G.1   Gaussian mixture model

As in the case of the multiclass SVM, we assume equal priors on the class means and equal energy (Assumption 1).

**Theorem 7.** *Assume that the training set follows a multiclass GMM with noise covariance $\boldsymbol{\Sigma} = \mathbf{I}_p$ and Assumption 1 holds. Then, there exist constants $c_1, c_2, c_3 > 1$ and $C_1, C_2 > 1$ such that the solutions of the OvA-SVM and MNI are identical with probability at least $1 - \frac{c_1}{n} - c_2 k e^{-\frac{n}{c_3 k^2}}$ provided that*

$$p > C_1 kn \log(kn) + n - 1 \quad and \quad p > C_2 n^{1.5} \|\boldsymbol{\mu}\|_2. \tag{78}$$

We can compare Eqn. (78) with the corresponding condition for multiclass SVM in Theorem 2 (Eqn. (16)). Observe that the right-hand-side of Eqn. (78) above does not scale with $k$, while the right-hand-side of Eqn. (16) scales with $k$ as $k^3$. Otherwise, the scalings with $n$ and energy of class means $\|\boldsymbol{\mu}\|_2$ are identical. This discrepancy with respect to $k$-dependence arises because the multiclass SVM is equivalent to the OvA-SVM in Eqn. (24) with unequal margins $1/k$ and $(k-1)/k$ (as we showed in Thm. 1).

*Proof sketch.* Recall from Appendix C that we derived conditions under which the multiclass SVM interpolates the training data by studying the related symmetric OvA-type classifier defined in Eqn. (11). Thus, this proof is similar to the proof of Theorem 2 provided in Appendix C.2. The only difference is that the margins for the OvA-SVM are not $1/k$ and $(k-1)/k$, but 1 for all classes. Owing to the similarity between the arguments, we restrict ourselves to a proof sketch here.

Following Appendix C.2 and Eqn. (35), we consider $y_i = k$. We will derive conditions under which the condition

$$\left((1 + h_{kk}^{(-k)})g_{ki}^{(-k)} - s_{kk}^{(-k)}f_{ki}^{(-k)}\right) + C \sum_{j \neq k} \left((1 + h_{jj}^{(-j)})g_{ji}^{(-j)} - s_{jj}^{(-j)}f_{ji}^{(-j)}\right) > 0, \qquad (79)$$

holds with high probability for some $C > 1$. We define

$$\epsilon := \frac{n^{1.5}\|\boldsymbol{\mu}\|_2}{p} \leq \tau,$$

where $\tau$ is chosen to be a sufficiently small constant. Applying the same trick as in Lemma 2 (with the newly defined parameters $\epsilon$ and $\tau$) gives us with probability at least $1 - \frac{c_1}{n} - c_2 k e^{-\frac{n}{c_3 k^2}}$:

$$
\begin{aligned}
(79) &\geq \left(\left(1 - \frac{C_1 \epsilon}{\sqrt{k}\sqrt{n}}\right)\left(1 - \frac{1}{C_2}\right)\frac{1}{p} - \frac{C_3 \epsilon}{n} \cdot \frac{n}{kp}\right) - \frac{k}{C_4}\left(\left(1 + \frac{C_5 \epsilon}{\sqrt{k}\sqrt{n}}\right)\frac{1}{kp} - \frac{C_6 \epsilon}{n} \cdot \frac{n}{kp}\right) \\
&\geq \left(1 - \frac{1}{C_9} - \frac{C_{10}\epsilon}{\sqrt{k}\sqrt{n}} - \frac{C_{11}\epsilon}{k} - C_{12}\epsilon\right)\frac{1}{p} \\
&\geq \frac{1}{p}\left(1 - \frac{1}{C_9} - C_0\tau\right), \qquad (80)
\end{aligned}
$$

for some constants $C_i$'s $> 1$. We used the fact that $|g_{ji}^{(0)}| \leq (1/C)(1/(kp))$ for $j \neq y_i$ with probability at least $1 - \frac{c_1}{n} - c_2 k e^{-\frac{n}{c_3 k^2}}$ provided that $p > C_1 kn \log(kn) + n - 1$, which is the first sufficient condition in the theorem statement. $\qquad \square$

### G.2 Multinomial logistic model

Recall that we defined the data covariance matrix $\boldsymbol{\Sigma} = \sum_{i=1}^p \lambda_i \mathbf{v}_i \mathbf{v}_i^T = \boldsymbol{V}\boldsymbol{\Lambda}\boldsymbol{V}^T$ and its spectrum $\boldsymbol{\lambda} = [\lambda_1 \quad \cdots \quad \lambda_p]$. We also defined the effective dimensions $d_2 := \frac{\|\boldsymbol{\lambda}\|_1^2}{\|\boldsymbol{\lambda}\|_2^2}$ and $d_\infty := \frac{\|\boldsymbol{\lambda}\|_1}{\|\boldsymbol{\lambda}\|_\infty}$.

The following result provides sufficient conditions under which the OvA SVM and MNI classifier have the same solution with high probability under the MLM.

**Theorem 8.** *Assume that the training set follows a multiclass MLM. There exist constants $c$ and $C_1, C_2 > 1$ such that, if the following conditions hold:*

$$d_\infty > C_1 n \log(kn) \quad \text{and} \quad d_2 > C_2(\log(kn) + n), \qquad (81)$$

*the solutions of the OvA-SVM and MNI are identical with probability at least $(1 - \frac{c}{n})$. In the special case of isotropic covariance, the same result holds provided that*

$$p > 10n \log(\sqrt{k}n) + n - 1, \qquad (82)$$

Comparing this result to the corresponding results in Theorems 3 and 4, we observe that $k$ now only appears in the $\log$ function (as a result of $k$ union bounds). Thus, the unequal $1/k$ and $(k-1)/k$ margins that appear in the multiclass-SVM make interpolation harder than with the OvA-SVM, just as in the GMM case.

*Proof sketch.* For the OvA SVM classifier, we need to solve $k$ binary max-margin classification problems, hence the proof follows directly from [MNS$^+$20, Theorem 1] and [HMX21, Theorem 1] by applying $k$ union bounds. We omit the details for brevity. $\qquad \square$

## H  One-vs-one SVM

In this section, we first derive conditions under which the OvO solutions interpolate, i.e, all data points are support vectors. We then provide an upper bound on the classification error of the OvO solution.

In OvO classification, we solve $k(k-1)/2$ binary classification problems, e.g., for classes pair $(c, j)$, we solve

$$\mathbf{w}_{\text{OvO},(c,j)} := \arg\min_{\mathbf{w}} \|\mathbf{w}\|_2 \quad \text{sub. to } \mathbf{w}^T\mathbf{x}_i \geq 1, \text{ if } \mathbf{y}_i = c; \quad \mathbf{w}^T\mathbf{x}_i \leq -1 \text{ if } \mathbf{y}_i = j, \ \forall i \in [n]. \tag{83}$$

Then we apply these $k(k-1)/2$ classifiers to a fresh sample and the class that got the highest $+1$ voting gets predicted.

We now present conditions under which every data point becomes a support vector over these $k(k-1)/2$ problems. We again assume equal priors on the class means and equal energy (Assumption 1).

**Theorem 9.** *Assume that the training set follows a multiclass GMM with noise covariance $\boldsymbol{\Sigma} = \mathbf{I}_p$ and Assumption 1 holds. Then, there exist constants $c_1, c_2, c_3 > 1$ and $C_1, C_2 > 1$ such that the solutions of the OvA-SVM and MNI are identical with probability at least $1 - \frac{c_1}{n} - c_2 k e^{-\frac{n}{c_3 k^2}}$ provided that*

$$p > C_1 n \log(kn) + (2n/k) - 1 \quad \text{and} \quad p > C_2 n^{1.5} \|\boldsymbol{\mu}\|_2. \tag{84}$$

*Proof sketch.* Note that the margins of OvO SVM are 1 and $-1$, hence the proof is similar to the proof of Theorem 7. Recall that in OvO SVM, we solve $k(k-1)/2$ binary problems and each problems has sample size $2n/k$ with high probability. Therefore, compared to OvA SVM which solves $k$ problems each with sample size $n$, OvO SVM needs less overparameterization to achieve interpolation. Thus the first condition in Eqn. (78) reduces to $p > C_1 n \log(kn) + (2n/k) - 1$.  □

We now derive the classification risk for OvO SVM classifiers. Recall that OvO classification solves $k(k-1)/2$ binary subproblems. Specifically, for each pair of classes, say $(i,j) \in [k] \times [k]$, we train a classifier $\mathbf{w}_{ij} \in \mathbb{R}^p$ and the corresponding decision rule for a fresh sample $\mathbf{x} \in \mathbb{R}^p$ is $\hat{y}_{ij} = \text{sign}(\mathbf{x}^T \hat{\mathbf{w}}_{ij})$. Overall, each class $i \in [k]$ gets a voting score $s_i = \sum_{j \neq i} \mathbf{1}_{\hat{y}_{ij}=+1}$. Thus, the final decision is given by majority rule that *decides the class with the highest score*, i.e., $\arg\max_{i \in [k]} s_i$. Having described the classification process, the total classification error $\mathbb{P}_e$ for balanced classes is given by the conditional error $\mathbb{P}_{e|c}$ given the fresh sample belongs to class $c$. Without loss of generality, we assume $c = 1$. Formally, $\mathbb{P}_e = \mathbb{P}_{e|1} = \mathbb{P}_{e|1}(s_1 < s_2 \text{ or } s_1 < s_3 \text{ or } \cdots \text{ or } s_1 < s_k)$. Under the equal prior and energy assumption, by symmetry and union bound, the conditional classification risk given that true class is 1 can be upper bounded as:

$$\mathbb{P}_{e|1}(s_1 < s_2 \text{ or } s_1 < s_3 \text{ or } \cdots \text{ or } s_1 < s_k)$$
$$\leq \mathbb{P}_{e|1}(s_1 < k-1) = \mathbb{P}_{e|1}(\exists j \text{ s.t. } \hat{y}_{1j} \neq 1) \leq (k-1)\mathbb{P}_{e|1}(\hat{y}_{12} \neq 1).$$

Therefore, it suffices to bound $\mathbb{P}_{e|1}(y_{12} \neq 1)$. We can directly apply Theorem 5 with changing $k$ to 2 and $n$ to $2n/k$.

**Theorem 10.** *Let Assumption 2 and the condition in Eqn. (84) hold. Further assume constants $C_1, C_2, C_3 > 1$ such that $\left(1 - C_1\sqrt{\frac{k}{n}} - \frac{C_2 n}{kp}\right)\|\boldsymbol{\mu}\|_2 > C_3$. Then, there exist additional constants $c_1, c_2, c_3$ and $C_4 > 1$ such that the OvO SVM solutions satisfies:*

$$\mathbb{P}_{e|c} \leq (k-1) \exp\left(-\|\boldsymbol{\mu}\|_2^2 \frac{\left(\left(1 - C_1\sqrt{\frac{k}{n}} - \frac{C_2 n}{kp}\right)\|\boldsymbol{\mu}\|_2 - C_3\right)^2}{C_4\left(\|\boldsymbol{\mu}\|_2^2 + \frac{kp}{n}\right)}\right) \tag{85}$$

*with probability at least $1 - \frac{c_1}{n} - c_2 k e^{-\frac{n}{c_3 k^2}}$, for every $c \in [k]$. Moreover, the same bound holds for the total classification error $\mathbb{P}_e$.*