# OpenReview forum: "Benign Overfitting in Multiclass Classification: All Roads Lead to Interpolation"
_NeurIPS.cc/2021/Conference — NeurIPS 2021 Poster_

### Official Review · Reviewer_YfRx · 2021-07-14

**Rating:** 7
**Confidence:** 4

**Summary:**

In this paper, the authors study benign overfitting in multiclass linear classification. They show the equivalence between one-vs-all SVM and multiclass SVM with either cross entropy or squared loss under a sufficient condition. Furthermore they show that when overparametrization occurs, the sufficient condition holds with high probability.

**Limitations And Societal Impact:**

I like this paper overall. Here are my main concerns
* The authors assume linearly separable data in the their work. However, that is not true in most cases. I would like to know about their thoughts regarding what would happen in such a case.
* Building on the previous question, I am wondering if it is possible to show some experimental results on non linearly separable data?
* Most of the classifiers use DNNs this days. It has been shown that the generalization for DNNs are very different from that of other classifiers. I am wondering if it possible to extend the results from this paper to DNNs. I would like to know the authors thoughts on this.

**Main Review:**

The authors explore the literature of benign overfitting for multiclass classification. For this purpose they use SVMs.  They consider the space of linearly separable data. In particular they show that under a deterministic sufficient condition, one-vs-all  SVM, is equivalent to multiclass SVM classifier using both squared loss and cross entropy loss. This is the first case to show that these two loss functions can be equivalent.

Furthermore, they show that for GMM or MLM, the aforementioned sufficient condition is satisfied with high probability under overparametrization based on the number of classes and quantities related to the data covariance. They also then provide experimental results to back up this claim.

Moreover, they provide error bounds for data generated under these conditions. This shows that benign overfitting happens irrespective of the loss function chosen.

After Rebuttal

Increased score after reading author response and other reviews. Please find the details in the comments.

**Time Spent Reviewing:**

7

---

> ### Author Response · Authors · 2021-08-10
> **Response to reviewer YfRx**
>
> We are glad to see that you like our work and thank you for the helpful comments.
>
> 1) __Implications for non-separable setting:__ While it is true that several traditional ML settings have non-separable data, the modern ML regime almost exclusively considers heavily overparameterized models in the sense of p > n (see, e.g. Neyshabur et al (2015), Zhang et al (2017), Belkin et al (2018), Nakkiran et al (2019), Papyan et al (2020) for discussions on overparameterized DNNs). Our results and findings exclusively apply to this overparameterized regime, in which data is trivially separable. This is because when p > n, an interpolating estimator exists almost surely on random data, and such an interpolating estimator clearly separates the data. Moreover, as Neyshabur et al (2015), Zhang et al (2017), Nakkiran et al (2019), Papyan et al (2020) demonstrate, modern DNNs are trained to zero CE loss, meaning that they in-fact perfectly separate even noisy training data. Our results apply in the overparameterized regime that is directly relevant to these settings.
>
>       Having said that, we also include here a few remarks regarding the reviewer’s question about what is known in the non-separable regime. In a recent line of work, (Taheri et al. 2020,2021) sharply characterized the generalization error of empirical risk minimization with general loss functions for non-separable binary gaussian mixture and logistic data. Using these, they found for square-loss that it is (i) optimal among all convex losses for gaussian mixture data, and (ii) approximately optimal (see Sec. 4.1 in [Taheri et al.’20]) for logistic data with gaussian regressors. Please see also (Mai et al 2019, Kini & Thrampoulidis 2020) for related results. To the best of our knowledge, these results are the only ones to accurately quantify suboptimality gaps of CE/square loss in the non-separable setting, which has otherwise been mostly investigated empirically (see e.g. [Rif02,RK04]). Yet, these results are still limited to (i) the proportional high-dimensional regime where p and n grow linearly, and, (ii) binary data. Multiclass extensions (which are significantly more technically challenging) have only been recently studied in [TOS20], but are limited to square loss. Our work (together with contemporaneous work by [Loureiro et al. 2021]) is the first to directly study the CE loss and its relation to the square loss. We are happy to use the extra page allowed to us after acceptance to include these further literature remarks.
>
> 2) __Implications for deep neural networks:__ We view the overparameterized linear model studied in our paper as an important first step in understanding these behaviors for real-world DNNs. This approach is in line with a wealth of recent works that focus on linear models, for which they show that they are able to capture numerous empirical/theoretical phenomena surrounding deep learning. At a high-level, this abstraction happens in the following order: deep nets->neural tangent kernel->random features->linear models with general covariance. Specifically, linear models have been shown to imitate empirical observations such as double descent (e.g. Belkin et al.'19, Hastie et al.’19, Belkin et al.’20, Bartlett et al.’20, Montanari et al.’19, Deng et al.’19, Mignacco’19, Aubin et al.’20,Muthukumar et al. ‘20) and implicit bias (e.g. Soudry et al.’18, Nacson et al.’19,  Papyan et al.’20, Ji & Telgarsky ‘18,’20, Belkin et al. ’21). Here, we show for the first time for linear models that effective overparameterization implies multiclass SVM = MNI. Following the hierarchy of abstractions listed above the next natural steps would be extending our results to random features and kernel SVMs. We believe our techniques and insights can be built upon to provide these results (as has been done previously for other problems starting from techniques first established for linear models). Independently, it is interesting to note that the interpolation structure described in Theorem 1, i.e., $w^Tx_i = -1/k$ or $(k-1)/k$, is related to the new _empirical_ findings of neural collapse in training of DNNs (Papyan et al. ’20). We believe that there may be a deeper connection between these observations that is potentially of interest to the community and worth exploring in the future.

---

> > ### Comment · Reviewer_YfRx · 2021-09-01
> > **Thanks for Your Response**
> >
> > I want to thank the authors for their response to my questions.
> >
> > I agree with the first part but not fully. Overparametrization works sufficiently differently in DNNs when compared to SVMs. But it is also true that the same principals of linear separability still applies for SVMs too.
> >
> > Regarding the second response regarding implications for DNNs, I do not think the findings will be easily applicable to the DNNs. This is because how generalization works differently for DNNs when compared to SVMs. However, I agree with the proposed next steps here.
> >
> > After reading the response and the author's comments. I have decided to increase my score.

---

> > > ### Author Response · Authors · 2021-09-01
> > > **Thank you for your response!**
> > >
> > > Thank you very much for your reply and follow-up comments. We acknowledge the points made. Also, we are glad that your assessment of our work improved!

---

### Official Review · Reviewer_Cuyv · 2021-07-16

**Rating:** 7
**Confidence:** 3

**Summary:**

The paper studies the problem of multiclass classification in the overparameterized regimes. The work derives a sufficient condition under which multiple multiclass SVM classifiers converge to the min-norm interpolating (MNI) solution, extending a recent result ([MNS+ 20]) beyond the binary classification setting. The paper then shows that for certain configurations of the Gaussian mixture model and multinomial logistic model, this condition holds under high enough effective overparameterization. Finally, the paper derives non-asymptotic bounds on the error of the MNI classifier with multinomial logistic data and discusses sufficient conditions for the multiclass SVM to satisfy benign overfitting.


**Main Review:**

**Quality and Clarity**:

- The paper is that it is very well-written and well organized. The ideas are presented clearly with intuitive explanations, and the proofs are rigorous. The authors are careful in evaluating the strengths/weaknesses of their work. The work seems to be built upon established literature and the review of related works is informative.

- The technical parts of the paper are of very high quality


**Originality**:

- The central part of the paper is built upon existing results for binary classification. All main messages of the work are affirmative answers to the main ideas outlined by [MNS+ 20] and [HMX20], i.e., in overparameterized regimes, all losses lead to the min-norm interpolating solution, and all data points are (in a sense) support vectors.

- On the other hand, it is worth noting that several parts of the extension to multiclass are non-trivial (with varying degrees) and some require significantly new technical treatments:
   - The most important point of novelty of the paper is the derivation of Condition (8), which is a generalization of Equation (22) of [MNS+ 20]. For multiclass settings, the complementary slackness condition does not directly imply interpolation (as in the binary case) and the condition needs to be designed to ensure that a set of symmetric constraints also hold. Both the formulation and the technical proof of Theorem 1 are new and meaningful.
   - The results of Theorem 2, 3, 4, 5, albeit are of high quality and are non-trivial, mostly follow the framework of [MNS+ 20] and [HMX20] with technical adjustments.


**Significance**:

- As stated previously, all main messages of the work are affirmative answers to the main ideas outlined by [MNS+ 20] and [HMX20], thus the result themselves do not make as strong of an impact on the way people think about overparameterized regimes.

- The technical contributions of the paper are highly specialized to highly specific models of data (Gaussian mixture model with equal energy, Multinomial logistic model with orthogonal means). This lack of generality is likely to hinder the use of the work in more general settings.


**Time Spent Reviewing:**

3

---

> ### Author Response · Authors · 2021-08-10
> **Response to reviewer Cuyv**
>
> Thank you for your time spent on our submission and for recognizing our contribution.
>
> For the benefit of other readers and for transparency, we would like to take the opportunity to elaborate on some of the insightful comments made by the reviewer.
>
> 1) __Significance of results in overparameterization literature:__ Although the main motivation comes from previous works [MNS+ 20], [HMX20] and [WT20], our work is the first one that derives similar results in the multiclass case. As you have already noted, this does not constitute a simple extension technically. However, it is also conceptually subtle: the prior works [MNS+20] and [HMX20] main contribution was to show that “all training points are support vectors”; which trivially implies that SVM = MNI for the binary case. In the multiclass case, the meaning of “all training points are support vectors” is no longer clear. If one were to interpret this condition as meeting all the inequality constraints in the multiclass SVM with equality (Equation (3) in submission), this by itself does not imply SVM = MNI. Our main conceptual contribution is to prove the existence of additional symmetric structure in the multiclass SVM solution that implies that the equality constraints are met in a very specific way, i.e. $w^Tx_i = -1/k$ or $(k-1)/k$ depending on the value of the label -- this is what implies SVM = MNI. In addition to this proof technique being entirely new, we note that there is no precedent for such special structure having been shown (mathematically) on a solution to a multiclass classification task.
>
> 2) __Generality of framework:__ We would like to bring to your attention several extensions that will easily follow from our work, some of which have been already explicitly derived in this submission. The most important high level point is that the strongest assumptions are needed for the error analysis of MNI, but our result of SVM = MNI actually holds under minimal assumptions, as detailed below:
>
> - We do not require orthogonality of the mean vectors to prove our classification error bound for the GMM case; this is only for ease of exposition. We show this extension to generalization of unorthogonal means in the Supplementary Material, Lines 1137-1147. We are happy to state this as a result in the main text if it is helpful for the reader. We acknowledge that extending the analysis in MLM to the case of unorthogonal means does remain an important open direction. However, please note that these assumptions are only required for the error analysis, not for the results on SVM = MNI.
>
> - The equal-energy/prior assumption (for GMM) is indeed important for the error analysis. Although we think the current proof technique still works, the discussion on very unbalanced prior or energy case is non-trivial and could reveal more interesting findings; thus we leave this as one of the future works.
>
> - The Gaussianity (or mixture-of-Gaussians) assumption on covariates is most essential for sharp error analysis; this is owing to particular challenges in analyzing the multiclass classification error (see [TOS20]). We do not believe Gaussianity is as essential for the first main contribution of our paper, which is to show that SVM = MNI with high probability. Theorem 1 constitutes a deterministic equivalence condition that can be applied, in principle, to any data distribution. The consequence for MLM (Theorem 3) already only requires sub-Gaussian covariates (and as in [HMX20], would hold for more general designs). A mixture of sub-Gaussians (with iid components) would suffice for Theorem 2. While such an assumption of iid components is still strong, this assumption appears in some form or another in almost all benign overfitting analyses and removing it would be highly non-trivial and of independent mathematical interest.
>
> We discuss extensions in brief in Lines 273-275 of the main submission and we plan to further elaborate on these in the revised version of the paper (using the extra page allowed to us).
>
> Thank you again for your time and for your positive feedback.

---

> ### Author Response · Authors · 2021-09-03
> **Thank you**
>
> Dear reviewer,
>
> As the discussion period is reaching its end, we wanted to thank you again for your time spent on your submission.
>
> We appreciate your positive feedback!
>
> Best,
> Authors

---

### Official Review · Reviewer_k5ri · 2021-07-17

**Rating:** 5
**Confidence:** 4

**Summary:**

“Benign overfitting” is a recently revealed phenomenon that high-dimensional models achieving zero training loss can still have good generalization. Existing works studying benign overfitting are mostly focused on the regression setting or the binary classification setting. In this paper, the authors establish benign overfitting guarantees for multi-class classification problems. It is shown that empirical risk minimization with cross-entropy loss is equivalent to the min-norm interpolating (MNI) solutions and the one-vs-all SVM classifier, and then derive error bounds on the accuracy of the MNI classifier.

**Limitations And Societal Impact:**

The authors have adequately addressed the limitations and potential negative societal impact of their work.

**Main Review:**

I think the analysis in this paper is interesting. Although I did not fully check all the proofs, the results in this paper seem to be sound. However, I have the following concerns about this paper.

- I am not sure if the results in this paper are novel enough. In fact, the difference between this paper and previous works, i.e., multi-class classification versus binary classification, may be quite incremental. The proof technique in this paper seems to be quite similar to previous works on binary classification as well. Moreover, it is in fact not very clear whether there is any advantage of using the multi-class cross-entropy loss. It seems that if one solves multiple binary classification problems separately, and derive error bounds using the results in [WT20], the result could be better: the (k-1) factor in the bound of Theorem 5 won’t change, and the k in the bound will simply be improved to a constant. For these reasons, I feel that the contribution of this paper may be limited to the extension to the multinomial logistic model.

- It is also not very convincing whether the conditions in Theorem 2 are tight in k. The condition p = \Omega(k^3 n) requires a very large dimension p, and since n is already the total number of training data (not the number of data per class), the dependency of p on k is quite bad. In the experiments in Figure 2, the authors only checked the tightness of the second condition in (16).


**Time Spent Reviewing:**

4 hours

---

> ### Author Response · Authors · 2021-08-10
> **Response to reviewer k5ri**
>
> Thank you for your time spent on our submission. However, as we explain below, we feel that our main contributions have not been considered and evaluated in full. Below, we respond to your concerns point-by-point. We would be grateful to address any further feedback/questions during the discussion phase.
>
> __1) “I am not sure if the results in this paper are novel enough. In fact, the difference between this paper and prior works, i.e. multiclass classification versus binary classification, might be quite incremental.”__
>
> Our work is the first to discuss the equivalence between SVM and MNI classifiers and the generalization to MNI for multiclass classification. In particular, the reviewer appears to have overlooked a main point of novelty of our work in Theorem 1 (Equations (6) to (10)), which provides a sufficient condition for the SVM and MNI solutions to coincide. As mentioned in Lines 219-224, and as also acknowledged by Reviewer Cuyv, the complementary slackness condition in multiclass case does not directly imply interpolation, as in the binary case. As a consequence, completely new complementary slackness conditions for multiclass SVM need to be carefully designed to achieve interpolation. Instead, we achieved showing interpolation only after discovering that a set of symmetric constraints also hold, i.e., $w^Tx_i = -1/k$ or $(k-1)/k$. This observation is also entirely new: such symmetric structure has not been previously shown in a multiclass SVM problem.
>
> __2) “Moreover, it is in fact not clear whether there is any advantage of using the multi class cross entropy loss. It seems that if one solves multiple binary classification problems separately, and derive error bounds using the results in [WT20], the result could be better: the (k-1) factor in the bound of Theorem 5 won’t change, and the k in the bound will simply be improved to a constant. For these reasons, I feel that the contribution of this paper may be limited to the extension to the multinomial logistic model.”__
>
> There are two fundamental issues with the reviewer’s remark:
>
> (i) the practically relevant problem to study is that of multiclass SVM as we do in this paper. This is because multiclass SVM is tightly linked to the commonly used CE loss. Instead, the reviewer’s suggested multiple binary classification problems are not related to any commonly used loss function.
>
> (ii) even if one decided to study the solution of multiple binary classification problems, it is not correct that the results follow directly using results in [WT20].
>
> We elaborate on these points below.
>
> (i) The reviewer suggests studying the solution of multiple binary SVM problems. This can correspond to either the one-vs-one (OvO) SVM (solving $k(k-1)/2$ problems) or the one-vs-all (OvA) SVM (solving $k$ problems). Instead, we study the solution of multiclass SVM. As we argue below, we do not believe the reviewer’s suggestion is as relevant for practice nor is it as illustrative for the main question of comparing CE to square loss. First, unlike the multiclass SVM, there is no implicit bias connection of the OvO or the OvA SVM classifier to any commonly used loss function with gradient descent. Multiclass SVM, on the other hand, naturally arises by solving CE loss. By the same reason, study of OvO or OvA SVM is not informative about the connection of CE loss to squared loss, which is our other main result (Theorem 1). All in all, multiclass CE and square loss is what people use in practice. All questions in the literature have been about comparison of the two and this is the topic of study in our paper. In fact, we actually do show some results for one-vs-all SVM. But since (as explained above), this is not as  interesting compared to the multiclass SVM analysis, we have deferred the results in Section G in SM. Also, as we detail next, the proof in [WT20] cannot be used here either because in multiclass classification data models, the data matrix $X$ has multiple mean components and so we need to use the new proof technique developed in this work.
>
> (ii) The reasoning that error bounds follow directly using results in [WT20] is incorrect and does not work. The proofs of Theorems 2 and 5 are significantly more involved compared to the binary case in [WT20]. To see this, note that in the multiclass case the inverse gram matrix $(XX^T)^{-1}$ involves multiple distinct class means unlike in the binary case. In particular, the technique in [WT20] cannot be directly applied here. Instead, we developed a novel recursive argument as highlighted in Lines 290-297 and further detailed in Section C in the Supplementary Material (SM). Specifically, for the multiclass problem, the gram matrix becomes $(Q + \sum \mu_i v_i^T)(Q + \sum \mu_i v_i^T)^T$. We start from bounding the quadratic forms involving $(Q + \mu_1 v_1^T)(Q + \mu_1 v_1^T)^T$, then use these results to bound the quadratic forms of $(Q + \mu_1 v_1^T + \mu_2 v_2^T)(Q + \mu_1 v_1^T + \mu_2 v_2^T)^T$ and repeat this procedure to get the bounds for quadratic forms of $(Q + \sum \mu_i v_i^T)(Q + \sum \mu_i v_i^T)^T$. Even the first step of this recursive argument, i.e., to bound quadratic forms of $(Q + \mu_1 v_1^T)(Q + \mu_1 v_1^T)^T$, does not directly follow from [WT20]. As an illustration of this point, note in SM Lemma 5, that the $s_{ij}$ terms should be small enough compared to the $s_{ii}$ terms. Instead, in [WT20], there is only one such “$s$” term involved in the derivations (see Lemma 7), not two. Our new idea here to show the desired in Lemma 5 was to carefully use the fact that $v_i$’s are orthogonal by definition (see Lemma 5 in SM).
>
> We conclude the response to the reviewer point, by reiterating  (see our response in point (1) for more details) our paper’s novelty: aside Theorem 5 and the results for the multinomial logit model, Theorems 1 and 2 are completely new results establishing the equivalence of the multiclass SVM and the MNI. Putting this together with our above justification for the non-triviality of Theorem 5, we respectfully disagree with the reviewer’s assessment that the contributions of our paper are limited to the extension to the multinomial logistic model.
>
> __3) “It is also not very convincing whether the conditions in Theorem 2 are tight in k.”__
>
> Regarding the tightness of the result in Theorem 2, we already pointed out in Lines 305-307 that the bounds could be tighter in their dependence on $k$. This is an interesting direction that we leave to future work, but we do not believe it constitutes a major weakness of our work as this is the first result of its type for the multiclass SVM and the dependence is still polynomial in $k$.
>
> Again, we would be grateful to address any further questions during the discussion phase.

---

> > ### Comment · Reviewer_k5ri · 2021-08-22
> > **Thanks for your response**
> >
> > Thanks for your detailed and comprehensive response. I think my concerns about the novelty of the paper have been addressed well. After reading the response and comparing the proofs in this paper and previous works, I agree that the multitask case is more complicated and nontrivial even given the binary case results.
> >
> > I would like to clarify some of my comments in the original review. I agree that studying the solution of multiclass SVM is important and is different from studying multiple binary SVM problems. I made this comparison just for a sanity check, and to get some understanding of the possible implications of the results. If my understanding is correct, the multiclass SVM result in this paper also implies a risk bound of applying $k(k-1)/2$ one-vs-one SVMs: start from the bound in Theorem 5 in this paper, we can (1) keep the first (k-1) factor, (2) replace all the other k's with 2, and (3) replace n with 2n/k. If we do so, it seems that we actually obtain a better bound, because for binary classification we only need $|| \mu ||$ to be larger than a constant, instead of $k$.
> >
> > I think this may be an interesting result that worths some discussion, as it shows that solving and aggregating multiple binary classifiers may be better than solving all classification problems together. Is it the nature of the Gaussian mixture model?

---

> > > ### Author Response · Authors · 2021-08-23
> > > **Thanks for acknowledging our response and OvO vs MNI**
> > >
> > > We are happy to hear that you found our response helpful and that it has clarified your main concerns. Thank you for taking the time to read and respond to our comments and thank you for carefully reading our results.
> > >
> > > Next, we respond to your question above. Thank you for your comment: it makes for a nice discussion that will be added in our paper. We will do so in the Conclusions section making use of the additional one page provided to us if the paper is accepted.
> > >
> > > We start by clarifying the one-vs-one (OvO) classifier (brought up by the reviewer) and the MNI classifier (studied in our paper; see Equation (5)), which reveals fundamental differences in their structure. In particular, the decision rules and parameterizations of OvO and MNI classifiers are quite different, as detailed below.
> > >
> > > *OvO classifiers:* OvO classification solves $k(k-1)/2$ binary subproblems. Specifically, for each pair of classes, say $(i,j)\in [k]\times [k]$, we train a classifier $w_{ij}\in\mathbb{R}^p$ and the corresponding decision rule for a fresh sample $x\in\mathbb{R}^p$ is $\psi_{ij} = \text{sign}(x^Tw_{ij})$. Overall, each class $i\in[k]$ gets a voting *score* $s_i = \sum_{j \ne i}1[\psi_{ij} = +1]$. Thus, the final decision is given by majority rule that *decides the class with the highest score*, i.e. $\arg\max_{i \in [k]}s_i$. Having described the classification process, the error probability for balanced classes is given by the conditional error probability given the fresh sample belongs to (say) class 1. Formally, $\text{P(error)}=\text{P(error given the correct class is 1)} = \text{P}(s_1 < s_2 \ \text{or} \ s_1 < s_3 \ \text{or} \cdots \text{or} \ s_1 < s_k \ \text{given the correct class is 1}).$ Under the equal prior and energy assumption, by symmetry and union bound, we have $\text{P(error)} \le (k-1)\text{P}(s_1 < s_2 \ \text{given the correct class is 1})$.
> > >
> > > *MNI classifier:*  The MNI classifier (see Equation (5)) trains k classifiers, say $w_1, \ldots, w_k$ corresponding to the k-classes. After training the decision of which class the fresh sample $x$ belongs is given by an argmax rule that *decides the class with the highest logit*, i.e. $\arg\max_{i\in[k]} x^Tw_i$. Thus, by union bound the classification error is upper bounded as  $\text{P(error given the correct class is 1)} \le (k-1)\text{P}(x^T{w}_1 < x^T{w}_2 \ \text{given the correct class is 1})$. The main contribution of Section 4 is upper bounding the right-hand side above for a multiclass GMM model (see Section E in the SM for the proof). (As the reviewer noted and we mentioned previously, this is different compared to bounding the corresponding term for binary SVM).
> > >
> > > In conclusion, in order to bound error of the OvO classifier we need to bound (conditional probabilities) $\text{P}(s_1 < s_2)$ comparing *scores* $s_1,s_2$ against each other. On the other hand, the MNI classifier requires us to bound  $\text{P}(x^T{w}_1 < x^T{w}_2)$ comparing *logits* against each other. Our proof technique handles the latter and gives rise to Theorem 5. Moreover, the scores in the OvO case are given in terms of $k(k-1)/2$ classifiers, while in MNI there are only $k$ classifiers. A direct link between the two probabilities above seems non-trivial to derive. In its current form, Theorem 5 does *not* imply anything about the error of OvO classification.
> > >
> > > We agree that characterizing the performance of OvO classifiers might be of its own independent interest. However, this is beyond the scope and main focus of our paper, which is establishing equivalence of cross-entropy (CE) to LS and deriving benign overfitting conditions for CE. We thank the reviewer for their comment as this makes for a nice discussion to be added in the section concerning future work.
> > >
> > > We end with a short comment regarding a different potential interpretation of the reviewer’s comment that might also be of interest. While, as explained above, Theorem 5 and its current proof do not imply an error bound for OvO classifiers, it is reasonable to ask what happens when applied to a binary setting of only two classes. In that case, by substituting $k=2$ in our bound of Theorem 5, we derive an error bound that coincides with the error bound of Theorem 4.3 by [WT20]. Thus, as a sanity check, our result reduces to that of [WT20] in the binary case.
> > >
> > > Thank you again for your time and for the opportunity to further discuss those points. Please let us know if anything else remains unclear. We are always happy to respond to additional questions.

---

> > > > ### Comment · Reviewer_k5ri · 2021-08-25
> > > > **Thanks for your reply**
> > > >
> > > > Thank you for your response. Given that my concerns about the novelty of this paper have been addressed in the authors' first response, I am willing to increase my score.
> > > >
> > > > However, I am still not yet convinced by the authors’ follow-up comments that Theorem 5 and its current proof do not imply an error bound for OvO classifiers. According to the previous discussion, isn’t the following derivation correct?
> > > >
> > > > $ P(s_1 < s_2 \text{ or } s_1 < s_3 \text{ or } \cdots s_1 < s_k \text{ given the correct class is }1) $
> > > > $\leq P( s_1 \neq k-1  \text{ given the correct class is }1 ) $
> > > > $= P(\text{There exists }j, \psi_{1j} \text{ does not predict 1, given the correct class is }1)$
> > > > $ \leq (k-1) P(\psi_{12} \text{ does not predict 1, given the correct class is }1)$
> > > >
> > > > The first inequality is probably loose, as it relaxes the original condition on $s_i$ into a much weaker condition. However, this ensures that the result can be combined with Theorem 5 to give a bound for the OvO classifier.
> > > >
> > > > If the above derivation is correct, then it seems that Theorem 5 can indeed imply an error bound for OvO classifiers. And notably, this probably loose bound still seems to be better than the bound for the MNI classifier.
> > > >
> > > > Overall, I think the paper has good potential, but some additional discussions on the results and their relation to binary classification can help make the conclusions more convincing.

---

> > > > > ### Author Response · Authors · 2021-08-25
> > > > > **Thank you for your suggestion**
> > > > >
> > > > > Again, thank you for your time – we greatly appreciate your feedback. We are also happy that our responses have clarified concerns regarding novelty and the main focus of the paper.
> > > > >
> > > > > Regarding your comment on the OvO classifier, your argument is indeed correct. Thus, as you correctly noted, the paper’s bound does yield a bound for the error of the OvO classifier, which is interesting in its own right. As discussed above, we are happy to include a discussion on the OvO classifier in the main body of the paper, and we will also add a section in the SM with the derivation that you suggested.
> > > > >
> > > > > To conclude with the bigger picture (also for the potential benefit of other readers): the focus of our work is on comparing the cross-entropy loss (CE) to squared loss (LS) under overparameterization. By implicit bias theory, this translates to the question of comparing the multiclass-SVM to the MNI classifiers. For GMM and MLM data, this is answered by our Theorems 1, 2, 3 and 4.  As a byproduct of proving the key property in Equation (10) of Theorem 1, we discovered (see Lines 225-230) that when Equation (8) holds, then multiclass SVM has the same solution as the OvA-type classifier in Equation (11). This prompted us to also discuss the OvA-classifier in the paper (and include a supplementary Section G with more results on it). Now, the reviewer correctly noticed that our Theorem 5 actually also implies an error bound for the OvO classifier. This gives us a nice reason to also discuss the OvO classifier in the paper, which we are happy to do in the camera-ready version.

---

> > > > > > ### Comment · Reviewer_k5ri · 2021-08-30
> > > > > > **Thanks for your reply**
> > > > > >
> > > > > > I appreciate your discussion. During the discussion, my original concern on the novelty of Theorem 1 and Theorem 2 is resolved. However, our discussion also reveals some unnatural or counter-intuitive results regarding the number of classes k. And as is pointed out in the original review, some of the rates regarding k are not necessarily tight. As a paper focusing on multi-class classification, I feel that this is a weakness of this paper that needs improvement. For these reasons, I have changed my score from 4 to 5.

---

> > > > > > > ### Author Response · Authors · 2021-08-30
> > > > > > > **The only relevant problem to study here is that of CE minimization, thus multi-class SVM. We are disappointed that the reviewer is overlooking our main contributions.**
> > > > > > >
> > > > > > > We appreciate your time. However we respectfully disagree on your judgment of our paper. As we had mentioned originally, we feel that you have entirely overlooked our main contributions and the technical contributions in proving Theorems 1, 2, 3 and 4. The same is true with you overlooking the fact that **the only relevant problem to study here is that of CE minimization, thus multi-class SVM**. OvO classifier is simply not an interesting object for the purpose of our study. We feel that we have sufficiently highlighted throughout the text (see Lines 303-309 and Sections C.1, C.2 in the SM) and in our  responses to you above the non-trivial technical challenges, compared to binary studies, arising from the *multiclass* gaussian mixture model, which makes the result of Theorem 5 non trivial (unlike a bound for OvO, which is direct follow up from the binary case as you said). Regarding tightness with respect to k, is something that we do discuss in the paper: see Figures 2(a) and 2(b), in which our simulation results suggest that our equivalence conditions might be (nearly) tight.
> > > > > > >
> > > > > > > Again, to the best of our knowledge, this is the first paper to:
> > > > > > >
> > > > > > > (i) show and prove equivalence of multi-class SVM to LS revealing mechanisms and a geometry that are entirely new compared to the binary case;
> > > > > > >
> > > > > > > (ii) derive classification error bounds for multi-class SVM that prove it can be close to Bayes error under sufficient overparameterization.
> > > > > > >
> > > > > > > Thank you.

---

### Official Review · Reviewer_BFAo · 2021-07-25

**Rating:** 5
**Confidence:** 3

**Summary:**

The authors aim to establish the equivalence of multiple loss functions in training and to analyse the conditions for benign overfitting in the rigime of multiclass classification. The paper proposes a crucial condition under which the solutions of multiclass SVM, one-vs-all SVM and min-norm interpolating (MNI) classifier are equivalent. It is proven that this condition holds in high probability in both Gaussian mixture model and multinomial logistic model. Finally, the paper gives the genalization bound of the MNI classifier based on its equivalence with SVM, as well as the condition for benign overfitting.

**Limitations And Societal Impact:**

This part does not apply to this paper.

**Main Review:**

## Originality
Although the paper applies some techniques of former work, it is the first to propose a sufficient condition for the equivalence of multiclass SVM and the MNI classifier and the generalization of the MNI classifier.
## Quality
The analysis in the main body of the paper is strong and convincing. However, the authors claim in the introduction part that **square loss is equivalent or even superior to cross-entropy loss**. The main body only proves the equivalence of the loss based on margin and the square loss. Cross-entropy loss, which is deserves more concern as a common choice in deep learning, is not sufficiently discussed.
## Clarity
The paper is well organized from the equivalence of SVM and MNI to the generalization of MNI and benign overfitting. Readers may find it easy to follow the flow.
## Significance
This paper studies benign overfitting for the task of multiclass classification, which is an important topic that has not been touched upon. The paper would be of much more significance if cross-entropy loss were carefully studied.

**Time Spent Reviewing:**

12

---

> ### Author Response · Authors · 2021-08-10
> **Response to reviewer BFAo**
>
> Thank you for your time spent on our submission  and for recognizing the originality of our work.
>
> We would like to clarify that our work indeed has direct implications for the cross-entropy (CE) loss. This is a consequence of the line of work on _implicit bias_ of gradient methods. In particular, Theorem 7 in Soudry et al. (2018) shows that if we train classifier $W$ using CE with gradient descent of constant step size, when the data is separable, the normalized iterates $W^t$ converge to the solution of the _multiclass SVM_. Hence, our proven equivalence between the multiclass SVM and MNI solutions guarantees that _gradient descent run with CE loss or squared loss will yield solutions with the exact same generalization performance_. We have formally explained these in Lines 182-187 in our submission, where we introduced multiclass SVM. We are nevertheless happy to further emphasize this point in the paper’s Introduction.

---

> > ### Author Response · Authors · 2021-08-27
> > **We would appreciate an update. Thank you.**
> >
> > Dear reviewer,
> >
> > Thank you again for your time.
> >
> > We wanted to make sure that our response above has clarified your concern regarding CE loss. We would appreciate it if you let us know if there are any other questions/concerns that you might have and we are happy to respond.

---

> > > ### Comment · Reviewer_BFAo · 2021-09-01
> > > **Thanks for your response**
> > >
> > > From my point of view, the question of equivalence between multiclass SVM and CE loss has been solved according to Theorem 7 in Soudry et al. (2018).
> > >
> > > I noticed that the same result has been used in Muthukumar et al. (2020). One cannot find a global minimizer of CE loss, and I am just wondering whether there will be a different result if the training process is charaterized in a different way.
> > >
> > > In general, the reasoning of the article is complete if the equivalence of multiclass SVM and CE loss is emphasized.

---

> > > > ### Author Response · Authors · 2021-09-01
> > > > **Thank you for the acknowledgement -- Please let us know of any remaining questions**
> > > >
> > > > First, we would like to thank you for your response.
> > > >
> > > > We are following up because we want to make sure that we are not misinterpreting your response and we do not leave any further questions/concerns unanswered.
> > > >
> > > > **Re: "From my point of view, the question of equivalence between multiclass SVM and CE loss has been solved according to Theorem 7 in Soudry et al. (2018)."**
> > > >
> > > > Agreed (as we also mention in Lines 89-90, Lines 185-187 and in our previous response to the reviewer).
> > > >
> > > > **Re: I noticed that the same result has been used in Muthukumar et al. (2020).**
> > > >
> > > > The implicit bias result by Soudry et al (2018) is indeed very useful. Specifically in the literature focusing on generalization (rather than optimization) this result has been used by several recent works translating the question “what is the generalization error of the solution found by gradient descent on CE?” to the question “what is the generalization error of the SVM solution?”. For example, the result has been used in Montanari et al. (2019), Deng et al. (2019), Muthukumar et al. (2020). However, note that these past works use this result in the **binary case**. To the best of our knowledge, we are the first to investigate analytically the same question but for the **multiclass case**.
> > > >
> > > > As highlighted throughout the paper and in our responses to other reviewers, the multiclass case is more intricate (and arguably more interesting) as there are different ways that one can define support vector machines (e.g. multiclass SVM, OvA SVM). From an “implicit bias” consideration (Soudry et al. 2018), the “correct” object of study is mutliclass SVM, which is indeed the focus of our paper.
> > > >
> > > > We are happy to further highlight the above in the Introduction of the camera-ready version.
> > > >
> > > > **Re: One cannot find a global minimizer of CE loss, and I am just wondering whether there will be a different result if the training process is charaterized in a different way.**
> > > >
> > > > This is correct. When data are linearly separable (which is the case under sufficient overparameterization) “traveling” in the direction of *any* linear separator will make the CE loss approach zero. This is easiest to see in the binary case. Formally, let $w$ be any linear separator such that $\forall i\in[n]$ $:~y_i (x_i^T w)\ge 1$, and consider $w_\alpha=\alpha w, \alpha>0$ in the direction of $w$. Then, $L(w_\alpha)=\sum_{i\in[n]}\log\left(1+e^{-y_i(x_i^Tw_\alpha)}\right)\leq\sum_{i\in[n]}\log\left(1+e^{-\alpha}\right)$, which gives that $\lim_{\alpha\to\infty}L(w_\alpha)=0$.
> > > > This is why the implicit bias result of Soudry et al (2018) is very interesting and non-trivial: There can be many linear separators and it is not clear which one gradient descent prefers. It turns out that it prefers the separator with the maximum margin (aka the solution to hard-margin SVM).
> > > >
> > > > Having said that, which solution is preferred indeed depends not only on the loss, but also on the CE. Most works focus on gradient descent akin its relation to the most commonly used SGD. But more recently there have indeed been works studying implicit bias of adaptive methods (eg AdaGrad/Adam); see for example here: https://papers.nips.cc/paper/2019/hash/3335881e06d4d23091389226225e17c7-Abstract.html and https://arxiv.org/pdf/2012.06244.pdf
> > > >
> > > > **Re: In general, the reasoning of the article is complete if the equivalence of multiclass SVM and CE loss is emphasized.**
> > > >
> > > > Thank you for acknowledging that. We are happy to see that your concern has been addressed. Also, we are happy to further emphasize the implicit bias property and include the above discussions in the revision using the additional one page provided to us if paper is accepted. Thank you for your suggestion.
> > > >
> > > > Please let us know if there are any remaining open questions from your end.

---

> > > > > ### Comment · Reviewer_BFAo · 2021-09-04
> > > > > **Problem of equivalence between SVM and CE loss**
> > > > >
> > > > > I re-checked with Theorem 7 in Soudry et al. (2018) and discovered potential problems of this paper. I am still doubtful whether the paper could demonstrate the equivalence of multiclass SVM and CE loss. The authors pointed to Theorem 7 in Soudry et al (2018) for such an implicit bias result. However, it is not clear how Theorem 7 in Soudry et al (2018) can be used in this paper. The total classification error on lines 177 and 178 is related to multiple $w_k$. In order to apply Theorem 7 in Soudry et al (2018), it seems that we need to replace $w_k$ with $w_k(t) / \\| w_k(t) \\|$ and take $t\rightarrow \infty$. There is no evidence that $\\| w_k(t) \\|$ for different k are the same. In other words, in multiclass classification, the classification error depends on not only the direction of the classifiers but also the norms. So I think this part is not demonstrated rigorously.

---

> > > > > > ### Author Response · Authors · 2021-09-05
> > > > > > **The equivalence holds since the decision of GD (on CE loss) is in the limit the same as the decision of multiclass SVM**
> > > > > >
> > > > > > Dear reviewer,
> > > > > >
> > > > > > Thank you for your careful reading. Let us elaborate on this to clear any doubts.
> > > > > >
> > > > > > In their Theorem 7, Soudry et al. (2018), indeed prove that the solutions obtained by training with CE loss using GD behaves as $w_j(t) = w_{SVM,j} \log(t) + \rho_j(t)$, where $w_{SVM,j}$ is the solution of the multiclass-SVM and residual $\rho_j(t)$ is bounded. We thus have as $t \to \infty$, $w_j(t)/\log(t) \to w_{SVM,j}$ for all $j\in[k]$. Note that this normalization of log(t) *does not* depend on the class index $j$. This implies that the classification error of both solutions is identical as we demonstrate formally below.
> > > > > >
> > > > > > Notice that the predicted label is decided based on a “winner takes it all strategy”. Thus, for a fresh sample $x$ and $k$ classes:
> > > > > >
> > > > > > $\bullet$ For GD at iteration $t$, the decision is $y_{GD}(t)$ $= \arg\max_{j\in[k]} w_j(t)^T x$.
> > > > > >
> > > > > > $\bullet$ For multiclass SVM, the decision is $y_{multi-SVM} = \arg\max_{j\in[k]} w_{SVM,j}^T x$.
> > > > > >
> > > > > > Now consider the limit of $t\to\infty$:
> > > > > >
> > > > > > $$
> > > > > > \lim_{t\to\infty} y_{GD}(t)
> > > > > > = \lim_{t\to\infty} \arg\max_j w_j(t)^T x
> > > > > > = \lim_{t\to\infty} \arg\max_j { (w_j(t)^T x) / \log(t) }
> > > > > > = \arg\max_j \lim_{t\to\infty} (w_j(t)/\log(t))^Tx
> > > > > > = \arg\max_j (w_{SVM,j})^T x
> > > > > > = y_{multi-SVM}
> > > > > > $$
> > > > > >
> > > > > > Above, the second equality follows since nothing changes in the decision if we divide everything by the same $\log(t)$ factor. The third equality follows because the function $\max_{j\in[k]} x_j$ is continuous. The fourth inequality follows from Theorem 7 in Soudry et al. (2018).
> > > > > >
> > > > > > Thus, the decision of GD (on CE loss) is in the limit the same as the decision of multiclass SVM.
> > > > > >
> > > > > > We are happy to include the above reasoning in our paper as per your suggestion around lines 182-187, which initially stated the result on $w_j(t)/||w_j(t)||$ as a consequence of Theorem 7 in Soudry et al. (2018).

---

> ### Author Response · Authors · 2021-09-02
> **On your rating**
>
> Dear reviewer,
>
> We wanted to reach out again since the discussion phase will be finalizing soon. After your initial positive comments and after acknowledging that our response clarified your concern regarding CE loss, it appears to us that you are in favor of our work.
>
> We would greatly appreciate it if this is reflected in your final rating.
>
> Otherwise, we are happy to hear from you regarding any remaining concerns and attempt to clarify.
>
> We thank you again for your time. We understand that the reviews/discussions have required a lot of effort from all of you.
>
> Thank you,
> Authors

---

> ### Author Response · Authors · 2021-09-11
> **Is your concern resolved?**
>
> Dear Reviewer,
>
> We appreciate your time spent on our submission and engaging in the discussion below.
>
> We believe that we have answered your question regarding implicit bias of CE and its connection to multiclass-SVM (established by Soudry et al. 2018), which motivates our investigations. As per your suggestion, we will include our clarifying response to you in Section 2 of the revision, where we formally introduce our setting.
>
> For your convenience we would also like to repeat here our paper's two main novelties:
> 1) We prove for the first time that under large effective overparameterization the risks of mutli-SVM, OvA-SVM and MNI are all the same, i.e. R(multiclass-SVM)=R(MNI)=R(OvA-SVM),
> 2) We derive the first high-dimensional non-asymptotic risk bounds for the MNI classifier for multiclass GMM and MLM data
>
> We thank you again for your attention. We would appreciate it if you let us know of any remaining concerns or otherwise inform us of your final decision regarding our submission.

---

### Decision · Program_Chairs · 2021-09-27

**Decision:**

Accept (Poster)

**Comment:**

This paper proves the equivalence between multi-class logistic regression and multi-class SVM, based on which it proves the benign overfitting for multi-class interpolator. The main concern from one of the reviewers is that the results derived for multi-class benign overfitting are worse than using the standard technique (i.e., converting $k$-class classification to $k(k-1)/2$ 1 vs. 1 binary classification problems.) The authors have made a great effort to address the reviewers’ questions and concerns. After the author response and reviewer discussion, the paper gathers enough support from the reviewers. Thus, I recommend acceptance.